# Segment Any Point Cloud Sequences by Distilling Vision Foundation Models

**Youquan Liu**[1,*] **Lingdong Kong**[1,2,*] **Jun Cen**[3] **Runnan Chen**[4] **Wenwei Zhang**[1,5]
**Liang Pan**[5] **Kai Chen**[1] **Ziwei Liu**[5,✉]
[1]Shanghai AI Laboratory [2]National University of Singapore
[3]The Hong Kong University of Science and Technology [4]The University of Hong Kong
[5]S-Lab, Nanyang Technological University
{liuyouquan,konglingdong,zhangwenwei,chenkai}@pjlab.org.cn
jcenaa@connect.ust.hk {liang.pan,ziwei.liu}@ntu.edu.sg

## Abstract

Recent advancements in vision foundation models (VFMs) have opened up new possibilities for versatile and efficient visual perception. In this work, we introduce ***Seal***, a novel framework that harnesses VFMs for segmenting diverse automotive point cloud sequences. Seal exhibits three appealing properties: *i)* ***Scalability:*** VFMs are directly distilled into point clouds, obviating the need for annotations in either 2D or 3D during pretraining. *ii)* ***Consistency:*** Spatial and temporal relationships are enforced at both the camera-to-LiDAR and point-to-segment regularization stages, facilitating cross-modal representation learning. *iii)* ***Generalizability:*** Seal enables knowledge transfer in an off-the-shelf manner to downstream tasks involving diverse point clouds, including those from real/synthetic, low/high-resolution, large/small-scale, and clean/corrupted datasets. Extensive experiments conducted on eleven different point cloud datasets showcase the effectiveness and superiority of Seal. Notably, Seal achieves a remarkable $45.0\%$ mIoU on nuScenes after linear probing, surpassing random initialization by $36.9\%$ mIoU and outperforming prior arts by $6.1\%$ mIoU. Moreover, Seal demonstrates significant performance gains over existing methods across 20 different few-shot fine-tuning tasks on all eleven tested point cloud datasets. The code is available at this link[2].

## 1 Introduction

Inspired by the achievements of large language models (LLMs) [87, 35, 113, 73, 18], a wave of vision foundation models (VFMs), such as SAM [50], X-Decoder [122], and SEEM [123], has emerged. These VFMs are revolutionizing the field of computer vision by facilitating the acquisition of pixel-level semantics with greater ease. However, limited studies have been conducted on developing VFMs for the 3D domain. To bridge this gap, it holds great promise to explore the adaptation or extension of existing 2D VFMs for 3D perception tasks.

As an important 3D perception task, accurately segmenting the surrounding points captured by onboard LiDAR sensors is crucial for ensuring the safe operation of autonomous vehicles [92, 94, 81]. However, existing point cloud segmentation models rely heavily on large annotated datasets for training, which poses challenges due to the labor-intensive nature of point cloud labeling [2, 95]. To address this issue, recent works have explored semi- [55, 59] or weakly-supervised [95, 60] approaches to alleviate the annotation burden. While these approaches show promise, the trained point segmentors tend to perform well only on data within their distribution, primarily due to

[*]Youquan and Lingdong contributed equally to this work. ✉ Ziwei serves as the corresponding author.
[2]GitHub Repo: https://github.com/youquanl/Segment-Any-Point-Cloud.

37th Conference on Neural Information Processing Systems (NeurIPS 2023).

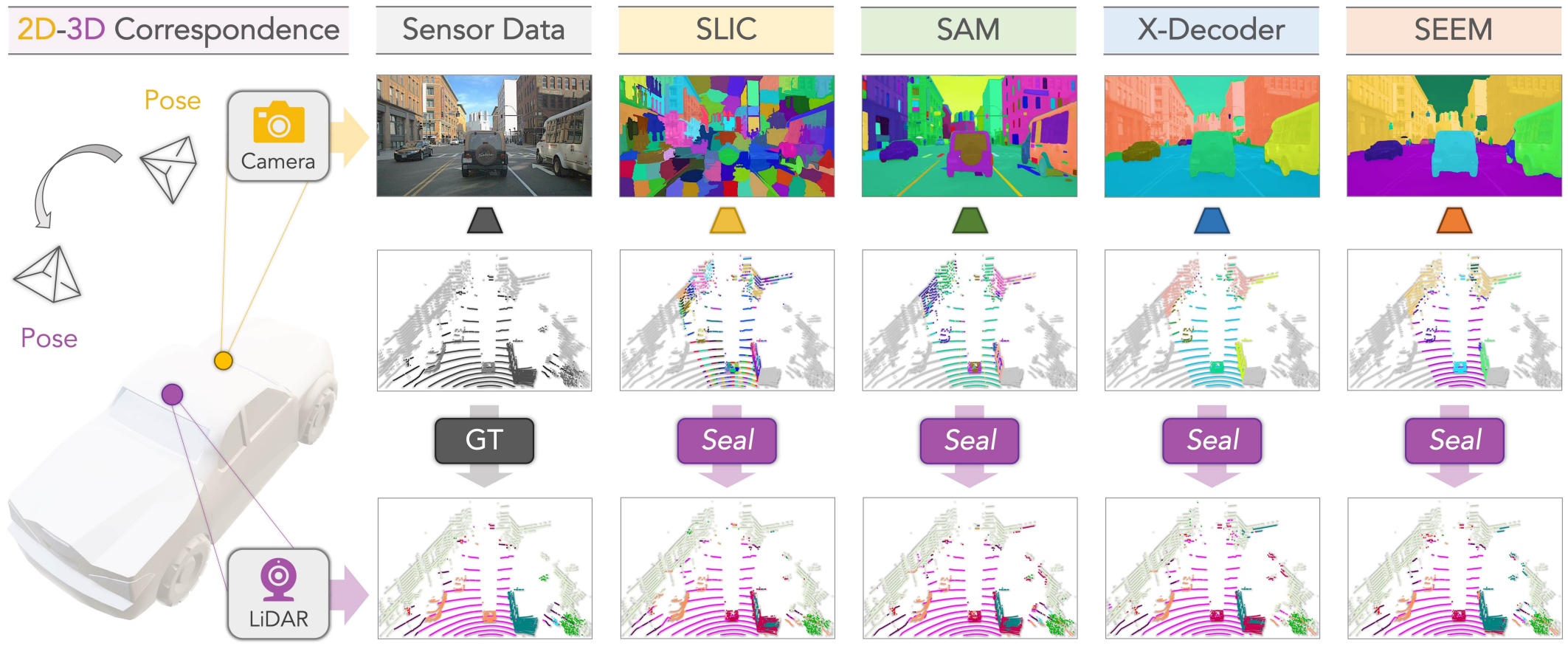

Figure 1: The proposed ***Seal*** distills semantic awareness on cameras views from VFMs to the point cloud via superpixel-driven contrastive learning. **[1st row]** Semantic superpixels generated by SLIC [1] and recent VFMs [50, 122, 123], where each color represents one segment. **[2nd row]** Semantic superpoints grouped by projecting superpixels to 3D via camera-LiDAR correspondence. **[3rd row]** Visualizations of the linear probing results of our framework driven by SLIC and different VFMs.

significant configuration differences (such as beam number, camera angle, emit rate) among different sensors [101, 28, 92]. This limitation inevitably hampers the scalability of point cloud segmentation.

To address the aforementioned challenges, we aim to develop a framework that can learn informative features while addressing the following objectives: *i)* Utilizing raw point clouds as input, thereby eliminating the need for semi or weak labels and reducing annotation costs. *ii)* Leveraging spatial and temporal cues inherent in driving scenes to enhance representation learning. *iii)* Ensuring generalizability to diverse downstream point clouds, beyond those used in the pretraining phase. Drawing inspiration from recent advancements in cross-modal representation learning [50, 122, 111, 123] and building upon the success of VFMs [50, 122, 111, 123], we propose a methodology that distills semantically-rich knowledge from VFMs to support self-supervised representation learning on challenging automotive point clouds. Our core idea is to leverage the 2D-3D correspondence in-between LiDAR and camera sensors and construct high-quality contrastive samples for cross-modal representation learning. As shown in Fig. 1, VFMs are capable of generating semantic superpixel groups from the camera views[3] and provide off-the-shelf semantic coherence for distinct objects and backgrounds in the 3D scene. Such consistency can be distilled from 2D to 3D and further yields promising performance on downstream tasks, which we will revisit more formally in later sections.

Compared to previous framework [85, 66], our VFM-assisted contrastive learning has three notable advantages: *i)* The semantically-aware region partition mitigates the severe "self-conflict" problem in driving scene contrastive learning. *ii)* The high-quality contrastive samples gradually form a more coherent optimization landscape, yielding a faster convergence rate than previous configurations. *iii)* A huge reduction in the number of superpixels also extenuates the overhead during pretraining.

Since a perfect calibration between LiDAR and cameras is often hard to meet, we further design a superpoint temporal consistency regularization to mitigate the case that there are potential errors from the sensor synchronization aspect. This objective directly resorts to the accurate geometric information from the point cloud and thus improves the overall resilience and reliability. As will be shown in the following sections, our framework (dubbed ***Seal***) is capable of **se**gmentation **a**ny point c**l**oud sequences across a wide range of downstream task on different automotive point cloud datasets.

To summarize, this work has the following key contributions:

- To the best of our knowledge, this study represents the first attempt at utilizing 2D vision foundation models for self-supervised representation learning on large-scale 3D point clouds.

[3]We show the front view for simplicity; more common setups have surrounding views from multiple cameras.

- We introduce ***Seal***, a scalable, consistent, and generalizable framework designed to capture semantic-aware spatial and temporal consistency, enabling the extraction of informative features from automotive point cloud sequences.
- Our approach demonstrates clear superiority over previous state-of-the-art (SoTA) methods in both linear probing and fine-tuning for downstream tasks across 11 different point cloud datasets with diverse data configurations.

## 2 Related Work

**Vision Foundation Models**. Recent excitement about building powerful visual perception systems based on massive amounts of training data [80, 50] or advanced self-supervised learning techniques [8, 74] is revolutionizing the community. The segment anything model (SAM) [50] sparks a new trend of general-purpose image segmentation which exhibits promising zero-shot transfer capability on diverse downstream tasks. Concurrent works to SAM, such as X-Decoder [122], OpenSeeD [111], SegGPT [99], and SEEM [123], also shed lights on the direct use of VFMs for handling different kinds of image-related tasks. In this work, we extend this aspect by exploring the potential of VFMs for point cloud segmentation. We design a new framework taking into consideration the off-the-shelf semantic awareness of VFMs to construct better spatial and temporal cues for representation learning.

**Point Cloud Segmentation**. Densely perceiving the 3D surroundings is important for autonomous vehicles [94, 41]. Various point cloud segmenters have been proposed, including methods based on raw points [42, 90, 43, 114, 78], range view [69, 106, 93, 118, 20, 16, 52], bird's eye view [115, 119, 9], voxels [19, 120, 37, 36], and multi-view fusion [62, 89, 107, 121, 17, 79]. Despite the promising results achieved, existing 3D segmentation models rely on large sets of annotated data for training, which hinders the scalability [28]. Recent efforts seek semi [55, 56, 59], weak [95, 40, 86, 63, 60], and active [64, 44, 117] supervisions or domain adaptation techniques [48, 47, 54, 76, 82, 68] to ease the annotation cost. In this work, we resort to self-supervised learning by distilling foundation models using camera-to-LiDAR associations, where no annotation is required during the pretraining stage.

**3D Representation Learning**. Stemmed from the image vision community, most 3D self-supervised learning methods focused on object-centric point clouds [84, 77, 83, 14, 97, 45, 110, 29, 91] or indoor scenes [38, 15, 22, 108, 10, 39, 57, 58, 109] by either pretext task learning [21, 70, 71, 112, 30], contrastive learning [11, 32, 12, 34, 27, 13, 46, 98, 104] or mask modeling [105, 31, 24, 61], where the scale and diversity are much lower than the outdoor driving scenes [67, 6]. PointContrast [103], DepthContrast [116], and SegContrast [72] are prior attempts aimed to establish contrastive objectives on point clouds. Recently, Sautier *et al.* [85] proposed the first 2D-to-3D representation distillation method called SLidR for cross-modal self-supervised learning on large-scale point clouds and exhibits promising performance. The follow-up work [66] further improves this pipeline with a semantically tolerant contrastive constraint and a class-balancing loss. Our framework also stems from the SLidR paradigm. Differently, we propose to leverage VFMs to establish the cross-modal contrastive objective which better tackles this challenging representation learning task. We also design a superpoint temporal consistency regularization to further enhance feature learning.

## 3 Seal: A Scalable, Consistent, and Generalizable Framework

In this section, we first revisit the 2D-to-3D representation distillation [85] as preliminaries (Sec. 3.1). We then elaborate on the technical details of our framework, which include the VFM-assisted spatial contrastive learning (Sec. 3.2) and the superpoint temporal consistency regularization (Sec. 3.3).

### 3.1 Preliminaries

Given a point cloud $\mathcal{P}^t = \{\mathbf{p}_i^t, \mathbf{e}_i^t | i = 1, .., N\}$ consists of $N$ points collected by a LiDAR acquisition at timestamp $t$, where $\mathbf{p}_i \in \mathbb{R}^3$ denotes the point coordinates and $\mathbf{e}_i \in \mathbb{R}^L$ is the feature embedding (intensity, elongation, *etc.*), our goal is to transfer the knowledge from an image set $\mathcal{I}^t = \{\{\mathbf{I}_1^t, ..., \mathbf{I}_j^t\} | j = 1, ..., V\}$ captured by $V$ synchronized cameras at $t$ to point cloud $\mathcal{P}^t$, where $\mathbf{I} \in \mathbb{R}^{3 \times H \times W}$ is a single image with height $H$ and width $W$. Prior works [85, 66] achieve this goal by first aggregating image regions that are similar in the RGB space into a superpixel set $\Phi_{\mathcal{S}} = \{\{\mathbf{s}_1, ..., \mathbf{s}_m\} | m = 1, ..., M\}$, via the unsupervised SLIC [1] algorithm. The corresponding

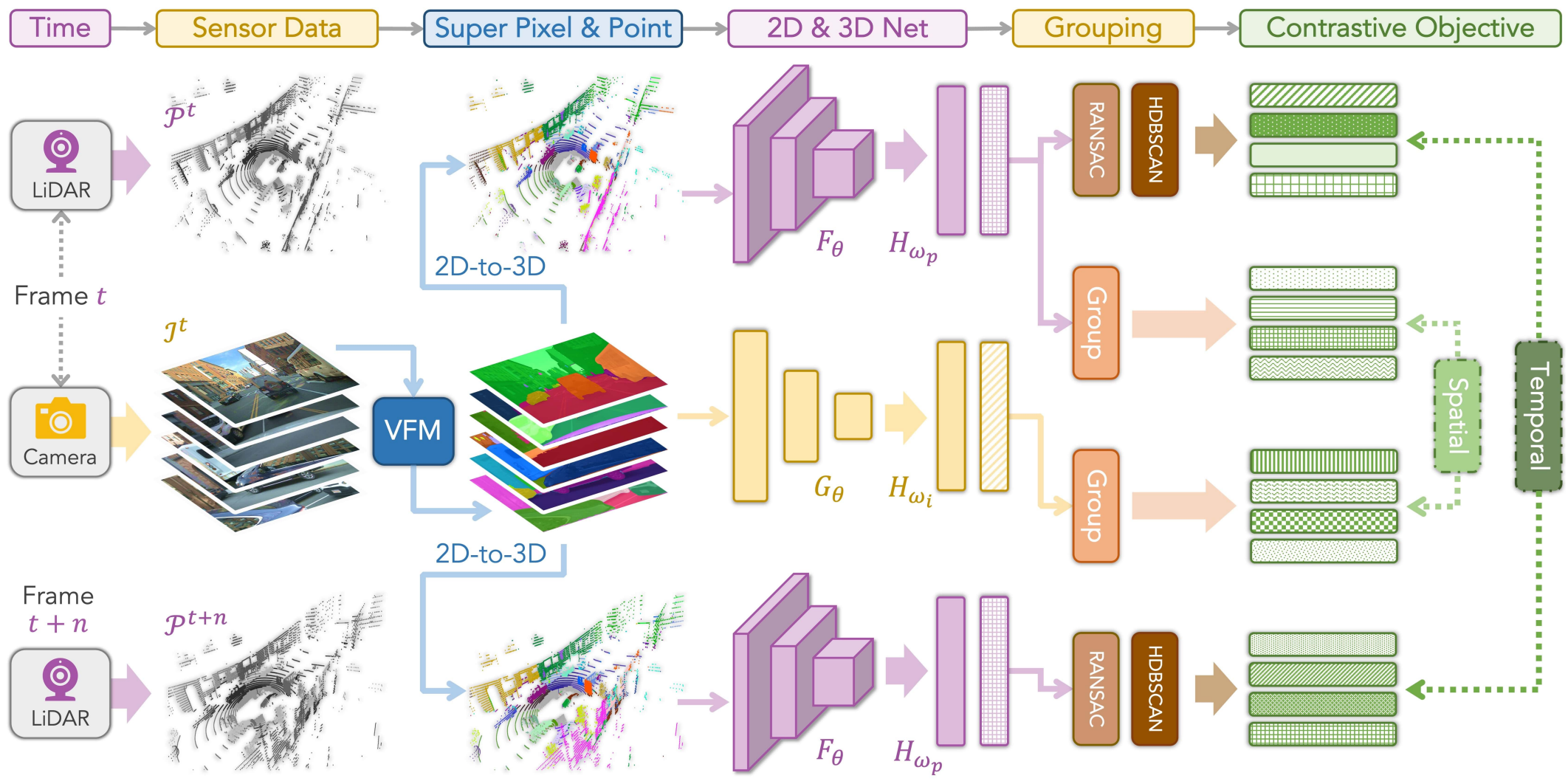

Figure 2: Overview of the ***Seal*** framework. We generate, for each {LiDAR, camera} pair $\{\mathcal{P}^t, \mathcal{I}^t\}$ at timestamp $t$ and another LiDAR frame $\mathcal{P}^{t+n}$ at timestamp $t+n$, the semantic superpixel and superpoint by vision foundation models (VFMs). Two pertaining objectives are then formed, including spatial contrastive learning between paired LiDAR and camera features (Sec. 3.2) and temporal consistency regularization between point segments at different timestamps (Sec. 3.3).

superpoint set $\Phi_{\mathcal{O}} = \{\{\mathbf{o}_1, ..., \mathbf{o}_m\} | m = 1, ..., M\}$ can be obtained by leveraging known sensor calibration parameters to establish correspondence between the points and image pixels. Specifically, since the LiDAR and cameras usually operate at different frequencies [7, 26], we transform each LiDAR point cloud $\mathbf{p_i} = (x_i, y_i, z_i)$ at timestamp $t_l$ to the pixel $\hat{\mathbf{p}}_i = (u_i, v_i) \in \mathbb{R}^2$ in the image plane at timestamp $t_c$ via the coordinate transformation. The point-to-pixel mapping is as follows:

$$[u_i, v_i, 1]^{\mathbf{T}} = \frac{1}{z_i} \times \Gamma_K \times \Gamma_{\text{camera}\leftarrow\text{ego}_{t_c}} \times \Gamma_{\text{ego}_{t_c}\leftarrow\text{global}} \times \Gamma_{\text{global}\leftarrow\text{ego}_{t_l}} \times \Gamma_{\text{ego}_{t_l}\leftarrow\text{lidar}} \times [x_i, y_i, z_i, 1]^{\mathbf{T}}, \tag{1}$$

where symbol $\Gamma_K$ denotes the camera intrinsic matrix. $\Gamma_{\text{camera}\leftarrow\text{ego}_{t_c}}$, $\Gamma_{\text{ego}_{t_c}\leftarrow\text{global}}$, $\Gamma_{\text{global}\leftarrow\text{ego}_{t_l}}$, and $\Gamma_{\text{ego}_{t_l}\leftarrow\text{lidar}}$ are the extrinsic matrices for the transformations of ego to camera at $t_c$, global to ego at $t_c$, ego to global at $t_l$, and LiDAR to ego at $t_l$, respectively.

### 3.2 Semantic Superpixel Spatial Consistency

**Superpixel Generation**. Prior works resort to SLIC [1] to group visually similar regions in the image into superpixels. This method, however, tends to over-segment semantically coherent areas (as shown in Fig. 1) and inevitably leads to several difficulties for contrastive learning. One of the main impediments is the so-called "self-conflict", where superpixels belonging to the same semantics become negative samples [96]. Although [66] proposed a semantically-tolerant loss to ease this problem, the lack of high-level semantic understanding still intensifies the implicit hardness-aware property of the contrastive loss. We tackle this challenge by generating semantic superpixels with VFMs. As shown in Fig. 1 and Fig. 4, these VFMs provide semantically-rich superpixels and yield much better representation learning effects in-between near and far points in the LiDAR point cloud.

**VFM-Assisted Contrastive Learning**. Let $F_{\theta_p} : \mathbb{R}^{N\times(3+L)} \rightarrow \mathbb{R}^{N\times C}$ be a 3D encoder with trainable parameters $\theta_p$, that takes a LiDAR point cloud as input and outputs a $C$-dimensional per-point feature. Let $G_{\theta_i} : \mathbb{R}^{H\times W\times 3} \rightarrow \mathbb{R}^{\frac{H}{s}\times\frac{W}{s}\times C}$ be an image encoder with parameters $\theta_i$, which is initialized from a set of 2D self-supervised pretrained parameters. The goal of our VFM-assisted contrastive learning is to transfer the knowledge from the pretrained 2D network to the 3D network via contrastive loss at the semantic superpixel level. To compute this VFM-assisted contrastive loss, we first build trainable projection heads $H_{\omega_p}$ and $H_{\omega_i}$ which map the 3D point features and 2D image features into the same $D$-dimensional embedding space. The point projection head $H_{\omega_p} : \mathbb{R}^{N\times C} \rightarrow \mathbb{R}^{N\times D}$ is a linear layer with $\ell_2$-normalization. The image projection head

$H_{\omega_i}: \mathbb{R}^{\frac{H}{s} \times \frac{W}{s} \times C} \rightarrow \mathbb{R}^{\frac{H}{s} \times \frac{W}{s} \times D}$ is a convolution layer with a kernel size of 1, followed by a fixed bilinear interpolation layer with a ratio of 4 in the spatial dimension, and outputs with $\ell_2$-normalization.

We distill the knowledge from the 2D network into the 3D network which is in favor of a solution where a semantic superpoint feature has a strong correlation with its corresponding semantic superpixel feature than any other features. Concretely, the superpixels $\Phi_{\mathcal{S}}$ are used to group pixel embedding and point embedding features. Then, an average pooling function is applied to each grouped point and pixel embedding features, to extract the superpixel embedding features $\mathbf{Q} \in \mathbb{R}^{M \times D}$ and superpoint embedding features $\mathbf{K} \in \mathbb{R}^{M \times D}$. The VFM-assisted contrastive loss is formulated as:

$$\mathcal{L}^{vfm} = \mathcal{L}\left(\mathbf{Q}, \mathbf{K}\right) = -\frac{1}{M}\sum_{i=1}^{M} \log \left[ \frac{e^{(\langle \mathbf{q}_i, \mathbf{k}_i \rangle / \tau)}}{\sum_{j \neq i} e^{(\langle \mathbf{q}_i, \mathbf{k}_j \rangle / \tau)} + e^{(\langle \mathbf{q}_i, \mathbf{k}_i \rangle / \tau)}} \right], \quad (2)$$

where $\langle \mathbf{q}_i, \mathbf{k}_j \rangle$ denotes the scalar product between superpoint embedding features and superpixel embedding features to measure the similarity. $\tau$ is the temperature term.

**Role in Our Framework**. Our VFM-assisted contrastive objective exhibits superiority over previous methods from three aspects: *i)* The semantically-rich superpixels provided by VFMs mitigate the "self-conflict" problem in existing approaches. *ii)* As we will show in the following sections, the high-quality contrastive samples from VFMs gradually form a much more coherent optimization landscape and yield a faster convergence rate than the unsupervised superpixel generation method. *iii)* Using superpixels generated by VFMs also helps our framework run faster than previous works, since the embedding length of $\mathbf{Q}$ and $\mathbf{K}$ has been reduced from a few hundred (SLIC [1]) to dozens (ours).

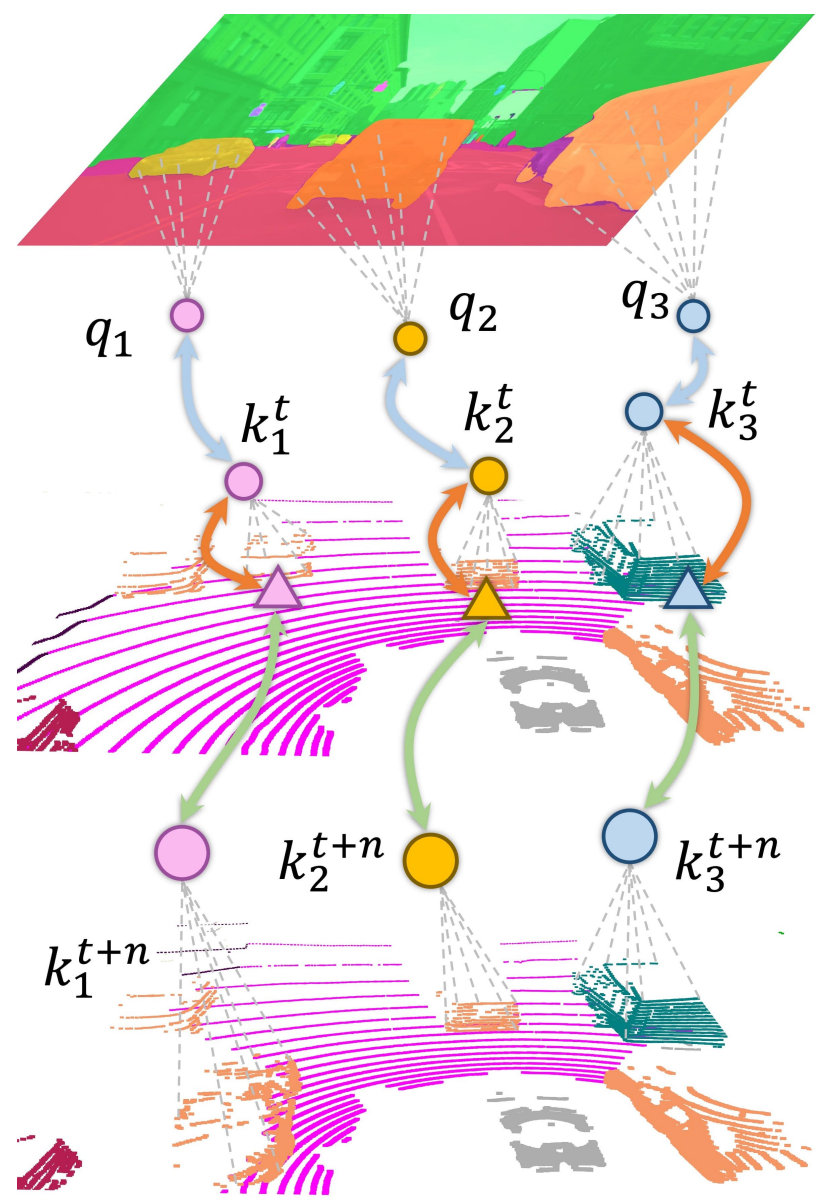

Figure 3: The positive feature correspondences in the contrastive learning objective in our contrastive learning framework. The *circles* and *triangles* represent the instance-level and the point-level features, respectively.

### 3.3 Semantic Superpoint Temporal Consistency

The assumption of having perfectly synchronized LiDAR and camera data might become too ideal and cannot always be fulfilled in actual deployment, which limits the scalability. In this work, we resort to accurate geometric information from point clouds to further relieve this constraint.

**Implicit Geometric Clustering**. To group coarse instance segments in a LiDAR frame, we first partition non-ground plane points $\mathcal{G}^t$ by eliminating the ground plane of a point cloud $\mathcal{P}^t$ at timestamp $t$ in an unsupervised manner via RANSAC [25]. Then, we group $\mathcal{G}^t$ to yield a set of $M_k$ segments $\mathcal{K}^t = \{\mathcal{K}_1^t, ..., \mathcal{K}_{M_k}^t\}$ with the help of HDBSCAN [23]. To map different segment views at different timestamps, we transform those LiDAR frames across different timestamps to the global frame and aggregate them with concatenations. The aggregated point cloud is denoted as $\widetilde{\mathcal{P}} = \{\widetilde{\mathcal{P}}^t, ..., \widetilde{\mathcal{P}}^{t+n}\}$. Similarly, we generate non-ground plane $\widetilde{\mathcal{G}} = \{\widetilde{\mathcal{G}}^t, ..., \widetilde{\mathcal{G}}^{t+n}\}$ from $\widetilde{\mathcal{P}}$ via RANSAC [25]. In the same manner as the single scan, we group $\widetilde{\mathcal{G}}$ to obtain $M_k$ segments $\widetilde{\mathcal{K}} = \{\widetilde{\mathcal{K}}_1, ..., \widetilde{\mathcal{K}}_{M_k}\}$. To generate the segment masks for all $n+1$ scans at $n$ consecutive timestamps, *i.e.*, $\widetilde{\mathcal{K}} = \{\widetilde{\mathcal{K}}^t, ..., \widetilde{\mathcal{K}}^{t+n}\}$, we maintain the point index mapping from the aggregated point cloud $\widetilde{\mathcal{P}}$ to the $n+1$ individual scans.

**Superpoint Temporal Consistency**. We leverage the clustered segments to compute the temporal consistency loss among related semantic superpoints. Here we assume $n = 1$ (*i.e.* the next frame) without loss of generalizability. Specifically, given a sampled temporal pair $\widetilde{\mathcal{P}}^t$ and $\widetilde{\mathcal{P}}^{t+1}$ and their corresponding segments $\widetilde{\mathcal{K}}^t$ and $\widetilde{\mathcal{K}}^{t+1}$, we compute the point-wise features $\hat{\mathcal{F}}^t \in \mathbb{R}^{N \times D}$ and $\hat{\mathcal{F}}^{t+1} \in \mathbb{R}^{N \times D}$ from the point projection head $H_{\omega_p}$. As for the target embedding, we split the point features $\hat{\mathcal{F}}^t$ and $\hat{\mathcal{F}}^{t+1}$ into $M_k$ groups by segments $\widetilde{\mathcal{K}}^t$ and $\widetilde{\mathcal{K}}^{t+1}$. Then, we apply an average pooling operation on $\hat{\mathcal{F}}^{t+1}$ to get $M_k$ target mean feature vectors $\mathcal{F}^{t+1} = \{\mathcal{F}_1^{t+1}, \mathcal{F}_2^{t+1}, ..., \mathcal{F}_{M_k}^{t+1}\}$, where $\mathcal{F}_{M_k}^{t+1} \in \mathbb{R}^{1 \times D}$. Let the split point feature $\hat{\mathcal{F}}^t$ be $\mathcal{F}^t = \{\mathcal{F}_1^t, \mathcal{F}_2^t, ..., \mathcal{F}_{M_k}^t\}$, where $\mathcal{F}_{M_k}^t \in \mathbb{R}^{k \times D}$ and

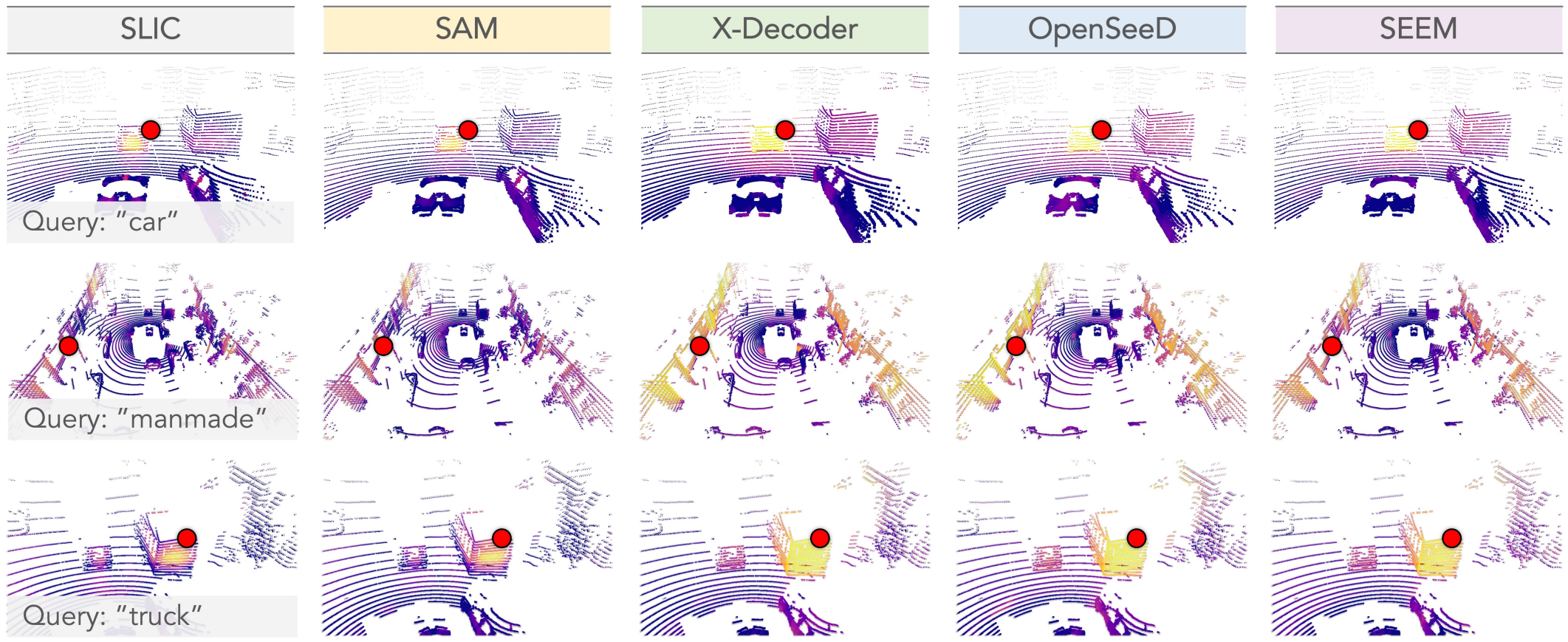

Figure 4: The **cosine similarity** between a query point (denoted as the **red dot**) and the feature learned with SLIC [1] and different VFMs [50, 122, 111, 123]. The queried semantic classes from top to bottom examples are: "car", "manmade", and "truck". The color goes from **violet** to **yellow** denoting **low** and **high** similarity scores, respectively. Best viewed in color.

$k$ is the number of points in the corresponding segment. We compute the temporal consistency loss $\mathcal{L}^{t\rightarrow t+1}$ to minimize the differences between the point features in the current frame (timestamp $t$) and the corresponding segment mean features from the next frame (timestamp $t+1$) as follows:

$$\mathcal{L}^{t\rightarrow t+1} = -\frac{1}{M_k}\sum_{i=1}^{M_k}\log\left[\frac{e^{(\langle \mathbf{f}_i^t,\mathbf{f}_i^{t+1}\rangle/\tau)}}{\sum_{j\neq i} e^{(\langle \mathbf{f}_i^t,\mathbf{f}_j^{t+1}\rangle/\tau)} + e^{(\langle \mathbf{f}_i^t,\mathbf{f}_i^{t+1}\rangle/\tau)}}\right]. \tag{3}$$

Since the target embedding for all points within a segment in the current frame serves as the mean segment representation from the next frame, this loss will force points from a segment to converge to a mean representation while separating from other segments, implicitly clustering together points from the same instance. Fig. 3 provides the positive feature correspondence in our contrastive learning framework. Furthermore, we swap $\hat{\mathcal{F}}^t$ when generating the target mean embedding features to form a symmetric representation. In this way, the correspondence is encouraged from both $t\rightarrow t+1$ and $t+1\rightarrow t$, which leads to the following optimization objective: $\mathcal{L}^{tmp} = \mathcal{L}^{t\rightarrow t+1} + \mathcal{L}^{t+1\rightarrow t}$.

**Point to Segment Regularization**. To pull close the LiDAR points belonging to the same instance at timestamp $t$, we minimize the distance between the point feature $\mathcal{F}^t$ and the corresponding mean cluster feature $\mathcal{C}^t$. To implement this, we leverage a max-pooling function to pool $\mathcal{F}^t$ according to the segments to obtain $\mathcal{C}^t = \{\mathcal{C}_1^t, \mathcal{C}_2^t, ..., \mathcal{C}_{M_k}^t\}$, where $\mathcal{C}_{M_k}^t \in \mathbb{R}^{1\times D}$. The point-to-segment regularization is thus achieved via the following loss function:

$$\mathcal{L}^{p2s} = -\frac{1}{M_k N_k}\sum_{i=1}^{M_k}\sum_{a=1}^{N_k}\log\left[\frac{e^{(\langle \mathbf{c}_i^t,\mathbf{f}_{i,a}^t\rangle/\tau)}}{\sum_{j\neq i} e^{(\langle \mathbf{c}_i^t,\mathbf{f}_{j,a}^t\rangle/\tau)} + e^{(\langle \mathbf{c}_i^t,\mathbf{f}_{i,a}^t\rangle/\tau)}}\right], \tag{4}$$

where $N_k$ represents the number of points within the corresponding segment. The final optimization objective is to minimize the aforementioned semantic spatial consistency loss $\mathcal{L}^{vfm}$, temporal consistency loss $\mathcal{L}^{tmp}$, and the point-to-segment regularization loss $\mathcal{L}^{p2s}$.

**Role in Our Framework**. Our semantic superpoint temporal consistency resorts to the accurate geometric information from the point cloud and exploits the different views of an instance across different timestamps to learn a temporally consistent representation. Considering the worst-case scenario that the 2D-3D correspondence between the LiDAR and camera sensors becomes unreliable, this geometric constraint can still effectively mitigate the potential errors that occur in inaccurate cross-sensor calibration and synchronization. Besides, our point-to-segment regularization mechanism can serve to aggregate the spatial information thus contributing to the effect of better-distinguishing instances in the LiDAR-acquired scene, *e.g.*, "car" and "truck". As we will show in the following sections, our experimental results are able to verify the effectiveness and superiority of the proposed consistency regularization objectives, even under certain degrees of perturbation in-between sensors.

Table 1: Comparisons of different pretraining methods pretrained on *nuScenes* [26] and fine-tuned on *nuScenes* [7], *SemanticKITTI* [3], *Waymo Open* [88], and *Synth4D* [82]. **LP** denotes linear probing with frozen backbones. Symbol † denotes fine-tuning with the LaserMix augmentation [55]. Symbol ‡ denotes fine-tuning with semi-supervised learning. All mIoU scores are given in percentage (%).

| Method & Year | nuScenes LP | nuScenes 1% | nuScenes 5% | nuScenes 10% | nuScenes 25% | nuScenes Full | KITTI 1% | Waymo 1% | Synth4D 1% |
|---|---|---|---|---|---|---|---|---|---|
| Random | 8.10 | 30.30 | 47.84 | 56.15 | 65.48 | 74.66 | 39.50 | 39.41 | 20.22 |
| PointContrast [ECCV'20] [103] | 21.90 | 32.50 | - | - | - | - | 41.10 | - | - |
| DepthContrast [ICCV'21] [116] | 22.10 | 31.70 | - | - | - | - | 41.50 | - | - |
| PPKT [arXiv'21] [65] | 35.90 | 37.80 | 53.74 | 60.25 | 67.14 | 74.52 | 44.00 | 47.60 | 61.10 |
| SLidR [CVPR'22] [85] | 38.80 | 38.30 | 52.49 | 59.84 | 66.91 | 74.79 | 44.60 | 47.12 | 63.10 |
| ST-SLidR [CVPR'23] [66] | 40.48 | 40.75 | 54.69 | 60.75 | 67.70 | 75.14 | 44.72 | 44.93 | - |
| **Seal (Ours)** | **44.95** | **45.84** | **55.64** | **62.97** | **68.41** | **75.60** | **46.63** | **49.34** | **64.50** |
| **Seal † (Ours)** | - | 48.41 | 57.84 | 65.52 | 70.80 | 77.13 | - | - | - |
| **Seal ‡ (Ours)** | - | 49.53 | 58.64 | 66.78 | 72.31 | 78.28 | - | - | - |

# 4 Experiments

## 4.1 Settings

**Data**. We verify the effectiveness of our approach on ***eleven*** different point cloud datasets. [1]***nuScenes*** [7, 26], [2]***SemanticKITTI*** [3], and [3]***Waymo Open*** [88] contain large-scale LiDAR scans collected from real-world driving scenes; while the former adopted a Velodyne HDL32E, data from the latter two datasets are acquired by 64-beam LiDAR sensors. [4]***ScribbleKITTI*** [95] shares the data with [3] but are weakly annotated with line scribbles. [5]***RELLIS-3D*** [49] is a multimodal dataset collected in an off-road campus environment. [6]***SemanticPOSS*** [75] is a small-scale set with an emphasis on dynamic instances. [7]***SemanticSTF*** [102] consist of LiDAR scans from adverse weather conditions. [8]***SynLiDAR*** [100], [9]***Synth4D*** [82], and [10]***DAPS-3D*** [51] are synthetic datasets collected from simulators. We also conduct extensive robustness evaluations on the [11]***nuScenes-C*** dataset proposed in the Robo3D benchmark [53], a comprehensive collection of eight out-of-distribution corruptions that occur in driving scenarios. More details on these datasets are in the Appendix.

**Implementation Details**. We use MinkUNet [19] as the 3D backbone which takes cylindrical voxels of size 0.10m as inputs. Similar to [85, 66], our 2D backbone is a ResNet-50 [33] pretrained with MoCoV2 [12]. We pretrain our segmentation network for 50 epochs on two GPUs with a batch size of 32, using SGD with momentum and a cosine annealing scheduler. For fine-tuning, we follow the exact same data split, augmentation, and evaluation protocol as SLidR [85] on *nuScenes* and *SemanticKITTI*, and adopt a similar procedure on other datasets. The training objective is to minimize a combination of the cross-entropy loss and the Lovász-Softmax loss [4]. We compare the results of prior arts from their official reporting [85, 66]. Since PPKT [65] and SLidR [85] only conducted experiments on *nuScenes* and *SemanticKITTI*, we reproduce their best-possible performance on the other nine datasets using public code. For additional details, please refer to the Appendix.

**Metrics**. We follow the conventional reporting of Intersection-over-Union (IoU) on each semantic class and the mean IoU (mIoU) across all classes. For robustness probing, we follow the Robo3D protocol [53] and report the mean Corruption Error (mCE) and mean Resilience Rate (mRR) scores calculated by using the $\text{MinkUNet}_{18}$ (torchsparse) implemented by Tang *et al.* [89] as the baseline.

## 4.2 Comparative Study

**Comparison to State-of-the-Arts**. We compare ***Seal*** with random initialization and five existing pretraining approaches under both linear probing (LP) protocol and few-shot fine-tuning settings on *nuScenes* [26] in Table 1. We observe that the pretraining strategy can effectively improve the accuracy of downstream tasks, especially when the fine-tuning budget is very limited (*e.g.* 1%, 5%, and 10%). Our framework achieves 44.95% mIoU under the LP setup – a 4.47% mIoU lead over to the prior art ST-SLidR [66] and a 6.15% mIoU boost compared to the baseline SLidR [85]. What is more, ***Seal*** achieves the best scores so far across all downstream fine-tuning tasks, which verifies the superiority of VFM-assisted contrastive learning and spatial-temporal consistency regularization. We also show that recent out-of-context augmentation [55] could enhance the feature learning during fine-tuning, which establishes a new state of the art on the challenging *nuScenes* benchmark.

Table 2: Comparisons of different pretraining methods pretrained on *nuScenes* [26] and fine-tuned on different downstream point cloud datasets. All mIoU scores are given in percentage (%).

| Method | ScribbleKITTI | | RELLIS-3D | | SemanticPOSS | | SemanticSTF | | SynLiDAR | | DAPS-3D | |
|---|---|---|---|---|---|---|---|---|---|---|---|---|
| | 1% | 10% | 1% | 10% | Half | Full | Half | Full | 1% | 10% | Half | Full |
| Random | 23.81 | 47.60 | 38.46 | 53.60 | 46.26 | 54.12 | 48.03 | 48.15 | 19.89 | 44.74 | 74.32 | 79.38 |
| PPKT [65] | 36.50 | 51.67 | 49.71 | 54.33 | 50.18 | 56.00 | 50.92 | 54.69 | 37.57 | 46.48 | 78.90 | 84.00 |
| SLidR [85] | 39.60 | 50.45 | 49.75 | 54.57 | 51.56 | 55.36 | 52.01 | 54.35 | 42.05 | 47.84 | 81.00 | 85.40 |
| **Seal (Ours)** | **40.64** | **52.77** | **51.09** | **55.03** | **53.26** | **56.89** | **53.46** | **55.36** | **43.58** | **49.26** | **81.88** | **85.90** |

Table 3: Robustness evaluations under eight out-of-distribution corruptions in the *nuScenes-C* dataset from the Robo3D benchmark [53]. All mCE, mRR, and mIoU scores are given in percentage (%).

| | Initial | Backbone | mCE ↓ | mRR ↑ | Fog | Wet | Snow | Move | Beam | Cross | Echo | Sensor |
|---|---|---|---|---|---|---|---|---|---|---|---|---|
| LP | PPKT [65] | MinkUNet | 183.44 | **78.15** | 30.65 | 35.42 | 28.12 | 29.21 | 32.82 | 19.52 | 28.01 | 20.71 |
| LP | SLidR [85] | MinkUNet | 179.38 | 77.18 | 34.88 | 38.09 | **32.64** | 26.44 | 33.73 | **20.81** | 31.54 | 21.44 |
| LP | **Seal (Ours)** | MinkUNet | **166.18** | 75.38 | **37.33** | **42.77** | 29.93 | **37.73** | **40.32** | 20.31 | **37.73** | **24.94** |
| Full | Random | PolarNet | 115.09 | 76.34 | 58.23 | 69.91 | 64.82 | 44.60 | 61.91 | 40.77 | 53.64 | 42.01 |
| Full | Random | CENet | 112.79 | 76.04 | 67.01 | 69.87 | 61.64 | 58.31 | 49.97 | **60.89** | 53.31 | 24.78 |
| Full | Random | WaffleIron | 106.73 | 72.78 | 56.07 | 73.93 | 49.59 | 59.46 | 65.19 | 33.12 | **61.51** | 44.01 |
| Full | Random | Cylinder3D | 105.56 | 78.08 | 61.42 | 71.02 | 58.40 | 56.02 | 64.15 | 45.36 | 59.97 | 43.03 |
| Full | Random | SPVCNN | 106.65 | 74.70 | 59.01 | 72.46 | 41.08 | 58.36 | 65.36 | 36.83 | 62.29 | **49.21** |
| Full | Random | MinkUNet | 112.20 | 72.57 | 62.96 | 70.65 | 55.48 | 51.71 | 62.01 | 31.56 | 59.64 | 39.41 |
| Full | PPKT [65] | MinkUNet | 105.64 | 76.06 | 64.01 | 72.18 | 59.08 | 57.17 | 63.88 | 36.34 | 60.59 | 39.57 |
| Full | SLidR [85] | MinkUNet | 106.08 | 75.99 | 65.41 | 72.31 | 56.01 | 56.07 | 62.87 | 41.94 | 61.16 | 38.90 |
| Full | **Seal (Ours)** | MinkUNet | **92.63** | **83.08** | **72.66** | **74.31** | **66.22** | **66.14** | **65.96** | 57.44 | 59.87 | 39.85 |

**Downstream Generalization**. To further verify the capability of ***Seal*** on segmenting *any* automotive point clouds, we conduct extensive experiments on eleven different datasets and show the results in Table 1 and Table 2. Note that each of these datasets has a unique data collection protocol and a diverse data distribution – ranging from diverse sensor types and data acquisition environments to scales and fidelity – making this evaluation a comprehensive one. The results show that our framework constantly outperforms prior arts across all downstream tasks on all eleven datasets, which concretely supports the effectiveness and superiority of our proposed approach.

**Semi-Supervised Learning**. Recent research [55, 59] reveals that combining both labeled and unlabeled data for semi-supervised point cloud segmentation can significantly boost the performance on downstream tasks. We follow these works to implement such a learning paradigm where a momentum-updated teacher model is adopted to assign pseudo-labels for the unlabeled data. The results from the last row of Table 1 show that ***Seal*** is capable of providing reliable supervision signals for data-efficient learning. Notably, our framework with partial annotation is able to surpass some recent fully-supervised learning methods. More results on this aspect are included in the Appendix.

**Robustness Probing**. It is often of great importance to assess the quality of learned representations on out-of-distribution data, especially for cases that occur in the real-world environment. We resort to the recently established *nuScenes-C* in the Robo3D benchmark [53] for such robustness evaluations. From Table 3 we observe that the self-supervised learning methods [65, 85] in general achieve better robustness than their baseline [19]. ***Seal*** achieves the best robustness under almost all corruption types and exhibits superiority over other recent segmentation backbones with different LiDAR representations, such as range view [16], BEV [115], raw points [78], and multi-view fusion [89].

**Qualitative Assessment**. We visualize the predictions of each pertaining method pretrained and fine-tuned on *nuScenes* [26] in Fig. 5. A clear observation is the substantial enhancement offered by all pretraining methods when juxtaposed with a baseline of random initialization. Diving deeper into the comparison among the trio of tested techniques, ***Seal*** stands out, delivering superlative segmentation outcomes, especially in the intricate terrains of driving scenarios. This is credited to the strong spatial and temporal consistency learning that ***Seal*** tailored to excite during the pretraining. We have cataloged additional examples in the Appendix for more detailed visual comparisons.

### 4.3 Ablation Study

**Foundation Model Comparisons**. We provide the first study on adapting VFMs for large-scale point cloud representation learning and show the results in Table 4. We observe that different VFMs exhibit

Table 4: Ablation study on pretraining frameworks (ours *vs.* SLidR [85]) and the knowledge transfer effects from different vision foundation models. All mIoU scores are given in percentage (%).

| Method | Superpixel | nuScenes LP | nuScenes 1% | nuScenes 5% | nuScenes 10% | nuScenes 25% | nuScenes Full | KITTI 1% | Waymo 1% | Synth4D 1% |
|---|---|---|---|---|---|---|---|---|---|---|
| Random | - | 8.10 | 30.30 | 47.84 | 56.15 | 65.48 | 74.66 | 39.50 | 39.41 | 20.22 |
| SLidR | SLIC [1] | 38.80 | 38.30 | 52.49 | 59.84 | 66.91 | 74.79 | 44.60 | 47.12 | 63.10 |
| | SAM [50] | 41.49 | 43.67 | **55.97** | **61.74** | **68.85** | **75.40** | 43.35 | 48.64 | 63.15 |
| | X-Decoder [122] | 41.71 | 43.02 | 54.24 | 61.32 | 67.35 | 75.11 | 45.70 | 48.73 | 63.21 |
| | OpenSeeD [111] | 42.61 | 43.82 | 54.17 | 61.03 | 67.30 | 74.85 | **45.88** | 48.64 | **63.31** |
| | SEEM [123] | **43.00** | **44.02** | 53.03 | 60.84 | 67.38 | 75.21 | 45.72 | **48.75** | 63.13 |
| **Seal** | SLIC [1] | 40.89 | 39.77 | 53.33 | 61.58 | 67.78 | 75.32 | 45.75 | 47.74 | 63.37 |
| | SAM [50] | 43.94 | 45.09 | **56.95** | 62.35 | **69.08** | **75.92** | 46.53 | 49.00 | 63.76 |
| | X-Decoder [122] | 42.64 | 44.31 | 55.18 | 62.03 | 68.24 | 75.56 | 46.02 | 49.11 | 64.21 |
| | OpenSeeD [111] | 44.67 | 44.74 | 55.13 | 62.36 | 69.00 | 75.64 | 46.13 | 48.98 | 64.29 |
| | SEEM [123] | **44.95** | **45.84** | 55.64 | **62.97** | 68.41 | 75.60 | **46.63** | **49.34** | **64.50** |

Table 5: Ablation study of each component pretrained on *nuScenes* [26] and fine-tuned on *nuScenes* [26], *SemanticKITTI* [3], and *Waymo Open* [88]. **C2L:** Camera-to-LiDAR distillation. **VFM:** Vision foundation models. **STC:** Superpoint temporal consistency. **P2S:** Point-to-segment regularization.

| # | C2L | VFM | STC | P2S | nuScenes LP | nuScenes 1% | nuScenes 5% | nuScenes 10% | nuScenes 25% | nuScenes Full | KITTI 1% | Waymo 1% |
|---|---|---|---|---|---|---|---|---|---|---|---|---|
| (1) | ✓ | | | | 38.80 | 38.30 | 52.49 | 59.84 | 66.91 | 74.79 | 44.60 | 47.12 |
| (2) | ✓ | | ✓ | | 40.45 | 41.62 | 54.67 | 60.48 | 67.61 | 75.30 | 45.38 | 48.08 |
| (3) | ✓ | ✓ | | | 43.00 | 44.02 | 53.03 | 60.84 | 67.38 | 75.21 | 45.72 | 48.75 |
| (4) | ✓ | ✓ | ✓ | | 44.01 | 44.78 | 55.36 | 61.99 | 67.70 | 75.00 | 46.49 | 49.15 |
| (5) | ✓ | ✓ | | ✓ | 43.35 | 44.25 | 53.69 | 61.11 | 67.42 | 75.44 | 46.07 | 48.82 |
| (6) | ✓ | ✓ | ✓ | ✓ | **44.95** | **45.84** | **55.64** | **62.97** | **68.41** | **75.60** | **46.63** | **49.34** |

diverse abilities in encouraging contrastive objectives. All VFMs show larger gains than SLIC [1] with both frameworks, while SEEM [123] in general performs the best. Notably, SAM [50] tends to generate more fine-grained superpixels and yields better results when fine-tuning with more annotated data. We conjecture that SAM [50] generally provides more negative samples than the other three VFMs which might be conducive to the superpixel-driven contrastive learning. On all setups, ***Seal*** constantly surpasses SLidR [85] by large margins, which verifies the effectiveness of our framework.

**Cosine Similarity**. We visualize three examples of feature similarity across different VFMs in Fig. 4. We observe that our contrastive objective has already facilitated distinction representations before fine-tuning. The semantically-rich VFMs such as X-Decoder [122], OpenSeeD [111], and SEEM [123] offers overt feature cues for recognizing objects and backgrounds; while the unsupervised (SLIC [1]) or too fine-grained (SAM [50]) region partition methods only provide limited semantic awareness. Such behaviors have been reflected in the linear probing and downstream fine-tuning performance (see Table 4), where SEEM [123] tends to offer better consistency regularization impacts during the cross-sensor representation learning.

**Component Analysis**. Table 5 shows the ablation results of each component in the ***Seal*** framework. Specifically, direct integration of VFMs (row #3) or temporal consistency learning (row #2) brings $4.20\%$ and $1.65\%$ mIoU gains in LP, respectively, while a combination of them (row #4) leads to a $5.21\%$ mIoU gain. The point-to-segment regularization (row #5) alone also provides a considerable performance boost of around $4.55\%$ mIoU. Finally, an integration of all proposed components (row #6) yields our best-performing model, which is $6.15\%$ mIoU better than the prior art [85] in LP and also outperforms on all in-distribution and -out-of-distribution downstream setups.

**Sensor Misalignment**. An accurate calibration is crucial for establishing correct correspondences between LiDAR and cameras. In most cases, those sensors should be well-calibrated for an autonomous car. It is unusual if the calibration is completely unknown, but it is possible to be imprecise due to a lack of maintenance. Hence, we conduct the following experiments to validate the robustness of our method. For each point coordinate $\mathbf{p}_i = (x_i, y_i, z_i)$ in a LiDAR point cloud, its corresponding pixel $\hat{\mathbf{p}}_i = (u_i, v_i)$ in the camera view can be found via Eq. 1. To simulate the misalignment between LiDAR and cameras, we insert random noises into the camera extrinsic matrix $\Gamma_K$, with a relative proportion of $1\%$, $5\%$, and $10\%$. Table 6 shows the results of PPKT [65], SLidR [85], and ***Seal*** under such noise perturbations. We observe that the possible calibration errors between LiDAR

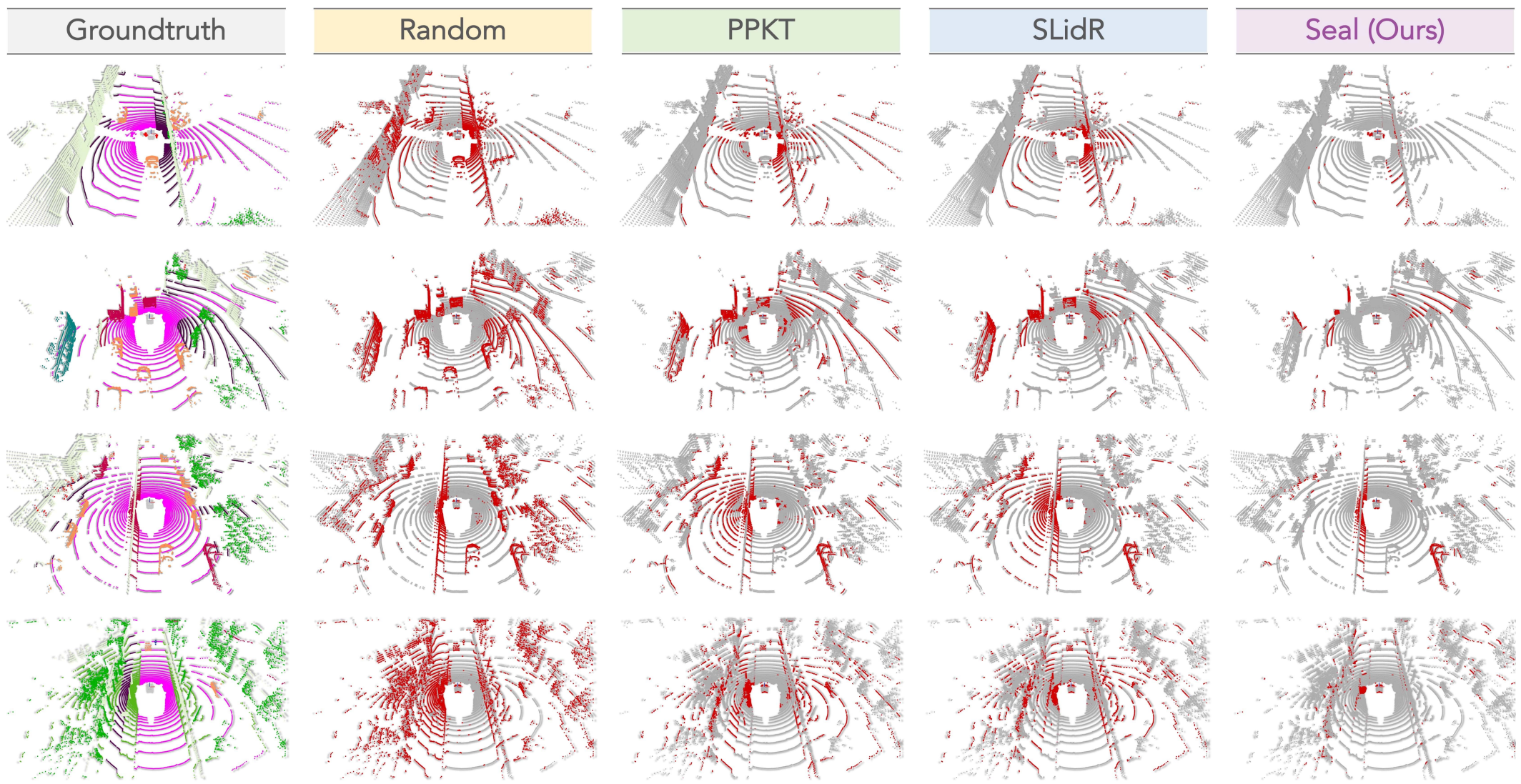

Figure 5: The **qualitative results** of different point cloud pretraining approaches pretrained on the raw data of *nuScenes* [26] and fine-tuned with 1% labeled data. To highlight the differences, the **correct** / **incorrect** predictions are painted in **gray** / **red**, respectively. Best viewed in color.

Table 6: Ablation study on the possible misalignment between the LiDAR and camera sensors. The perturbation is randomly generated and inserted. All mIoU scores are given in percentage (%).

| Method | 1% Misalignment | | | | 5% Misalignment | | | | 10% Misalignment | | | |
|---|---|---|---|---|---|---|---|---|---|---|---|---|
| | 1% | 5% | 10% | 25% | 1% | 5% | 10% | 25% | 1% | 5% | 10% | 25% |
| PPKT [65] | 34.94 | 51.11 | 58.54 | 65.01 | 33.69 | 51.40 | 58.00 | 64.11 | 33.35 | 50.98 | 57.84 | 63.52 |
| SLidR [85] | 37.92 | 53.08 | 59.89 | 66.90 | 38.00 | 52.36 | 60.01 | 64.10 | 37.30 | 51.11 | 58.50 | 64.50 |
| **Seal (Ours)** | **45.23** | **55.71** | **62.62** | **68.13** | **45.66** | **55.42** | **62.77** | **68.01** | **44.80** | **54.45** | **61.80** | **68.29** |

and cameras will cause performance degradation for different pretraining approaches (compared to Table 1). The performance degradation for PPKT [65] is especially prominent; we conjecture that this is because the point-wise consistency regularization of PPKT [65] relies heavily on the calibration accuracy and encounters problems under misalignment. Both SLidR [85] and ***Seal*** exhibit certain robustness; we believe the superpixel-level consistency is less sensitive to calibration perturbations. It is worth mentioning that ***Seal*** can maintain good performance under calibration error, since: *i)* Our VFM-assisted representation learning tends to be more robust; and *ii)* We enforce superpoint temporal consistency during the pertaining which does not rely on the 2D-3D correspondence.

## 5 Concluding Remark

In this study, we presented ***Seal***, a versatile self-supervised learning framework capable of segmenting *any* automotive point clouds by encouraging spatial and temporal consistency during the representation learning stage. Additionally, our work pioneers the utilization of VFMs to enhance 3D scene understanding. Extensive experimental results on 20 downstream tasks across eleven different point cloud datasets verified the effectiveness and superiority of our framework. We aspire for this research to catalyze further integration of large-scale 2D and 3D representation learning endeavors, which could shed light on the development of robust and annotation-efficient perception models.

**Potential Limitations**. Although our proposed framework holistically improved the point cloud segmentation performance across a wide range of downstream tasks, there are still some limitations that could hinder the scalability. *i)* Our model operates under the assumption of impeccably calibrated and synchronized LiDAR and cameras, which might not always hold true in real-world scenarios. *ii)* We only pretrain the networks on a single set of point clouds with unified setups; while aggregating more abundant data from different datasets for pretraining would further improve the generalizability of this framework. These limitations present intriguing avenues for future investigations.

**Acknowledgements**. This study is supported by the Ministry of Education, Singapore, under its MOE AcRF Tier 2 (MOE-T2EP20221-0012), NTU NAP, and under the RIE2020 Industry Alignment Fund – Industry Collaboration Projects (IAF-ICP) Funding Initiative, as well as cash and in-kind contribution from the industry partner(s). This study is also supported by the National Key R&D Program of China (No. 2022ZD0161600). We thank Tai Wang for the insightful review and discussion.

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

# Appendix

In this appendix, we supplement the following materials to support the findings and observations in the main body of this paper:

- Section A elaborates on additional implementation details to facilitate reproduction.
- Section B provides the complete quantitative results of our experiments.
- Section C includes more qualitative results to allow better visual comparisons.
- Section D acknowledges the public resources used during the course of this work.

## A Additional Implementation Detail

### A.1 Datasets

In this work, we conduct extensive experiments from a wide range of point cloud datasets to verify the effectiveness of the proposed ***Seal*** framework. A summary of the detailed configurations and emphases of these datasets is shown in Table A.

- [1]**nuScenes** [26]: The nuScenes[4] dataset offers a substantial number of samples collected by the LiDAR, RADAR, camera, and IMU sensors from Boston and Singapore, allowing machine learning models to learn useful multi-modal features effectively. For the point cloud segmentation task, it consists of $1000$ scenes of a total number of $1.1$B annotated points, collected by a Velodyne HDL32E LiDAR sensor. It also includes image data from six cameras, which are synchronized with the LiDAR sensor. In this work, we use the LiDAR point clouds and image data from nuScenes for model pretraining. We also conduct detailed fine-tuning experiments to validate the effectiveness of representation learning. More details of this dataset can be found at `https://www.nuscenes.org/nuscenes`.
- [2]**SemanticKITTI** [3]: The SemanticKITTI dataset is a comprehensive dataset designed for semantic scene understanding of LiDAR sequences. This dataset aims to advance the development of algorithms and models for autonomous driving and scene understanding using LiDAR data. It provides 22 densely labeled point cloud sequences that cover urban street scenes, which are captured by a Velodyne HDL-64E LiDAR sensor. In this work, we use the LiDAR point clouds from SemanticKITTI as a downstream task to validate the generalizability of pertaining methods. More details of this dataset can be found at `http://semantic-kitti.org`.
- [3]**Waymo Open** [88]: The Waymo Open dataset is a large-scale collection of real-world autonomous driving data. The 3D semantic segmentation subset contains $1150$ scenes, split into $798$ training, $202$ validation, and $150$ testing scenes. This subset contains $23691$ training scans, $5976$ validation scans, and $2982$ testing scans, respectively, with semantic segmentation labels from $23$ classes. The data are captured by five LiDAR sensors: one mid-range LiDAR sensor truncated to a maximum of 75 meters, and four short-range LiDAR sensors truncated to a maximum of 20 meters. In this work, we use the LiDAR point clouds from Waymo Open as a downstream task to validate the generalizability of pertaining methods. More details of this dataset can be found at `https://waymo.com/open`.
- [4]**ScribbleKITTI** [95]: The ScribbleKITTI dataset is a recent variant of the SemanticKITTI dataset, with weak supervisions annotated by line scribbles. It shares the exact same amount of training samples with SemanticKITTI, *i.e.*, $19130$ scans collected by a Velodyne HDL-64E LiDAR sensor, where the total number of valid semantic labels is $8.06\%$ compared to the fully-supervised version. Annotating the LiDAR point cloud in such a way corresponds to roughly a 90% time-saving. In this work, we use the LiDAR point clouds from ScribbleKITTI as a downstream task to validate the generalizability of pertaining methods. More details of this dataset can be found at `https://github.com/ouenal/scribblekitti`.

[4]Here we refer to the *lidarseg* subset of the nuScenes dataset. Know more details about this dataset at the official webpage: `https://www.nuscenes.org/nuscenes`.

Table A: The sensor configuration and data statistics for the ***eleven*** datasets used in our experiments.

| Dataset | Illustration | Sensor Setup | Statistics | Type |
|---|---|---|---|---|
| nuScenes [26] | 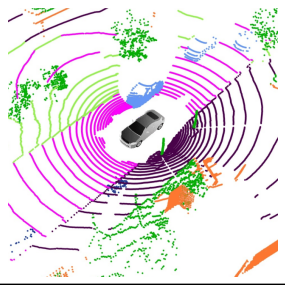 | 1× LiDAR (32-beam)
6× RGB Camera
5× RADAR
1× IMU & GPS | 16 semantic classes
29130 training samples
6019 validation samples
6008 testing samples | Real-world
Low-resolution point cloud
Dense annotation
Multi-modality |
| SemanticKITTI [3] | 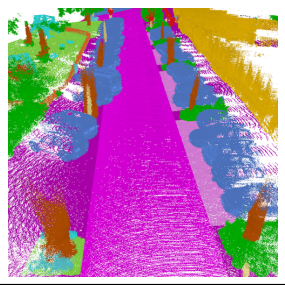 | 1× LiDAR (64-beam)
1× Stereo Camera
1× IMU & GPS | 19 semantic classes
19130 training samples
4071 validation samples
20351 testing samples | Real-world
High-resolution point cloud
Dense annotation
Multi-modality |
| Waymo Open [88] | 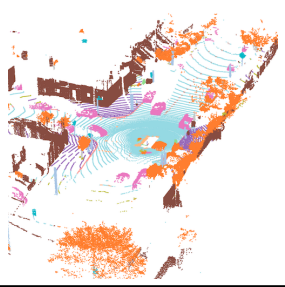 | 1× LiDAR (64-beam)
5× RGB Camera
1× IMU & GPS | 23 semantic classes
23691 training samples
5976 validation samples
2982 testing samples | Real-world
High-resolution point cloud
Dense annotation
Multi-modality |
| ScribbleKITTI [95] | 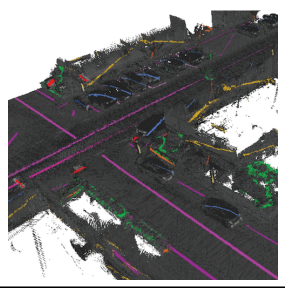 | 1× LiDAR (64-beam)
1× Stereo Camera
1× IMU & GPS | 19 semantic classes
19130 training samples
4071 validation samples
20351 testing samples | Real-world
High-resolution point cloud
Sparse annotation
Weakly-supervised learning |
| RELLIS-3D [49] | 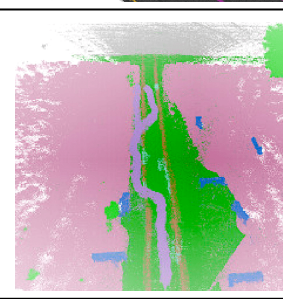 | 1× LiDAR (64-beam)
1× LiDAR (32-beam)
1× Stereo Camera
1× IMU & GPS | 20 semantic classes
7800 training samples
2413 validation samples
3343 testing samples | Real-world
High-resolution point cloud
Dense annotation
Multi-modality |
| SemanticPOSS [75] | 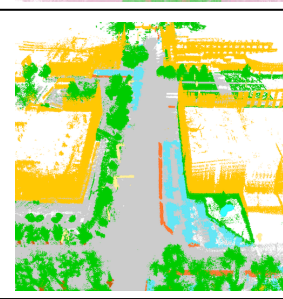 | 1× LiDAR (40-beam)
1× RGB Camera
1× IMU & GPS | 14 semantic classes
2488 training samples
500 validation samples | Real-world
High-resolution point cloud
Dense annotation
Dynamic instance |
| SemanticSTF [102] | 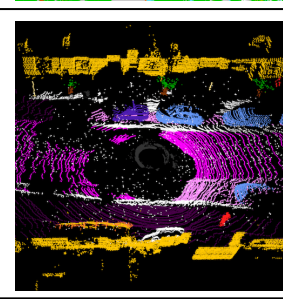 | 1× LiDAR (64-beam)
1× LiDAR (32-beam)
1× Stereo Camera
1× RADAR | 21 semantic classes
1326 training samples
250 validation samples
500 testing samples | Real-world
High-resolution point cloud
Dense annotation
Adverse weather |
| SynLiDAR [100] | 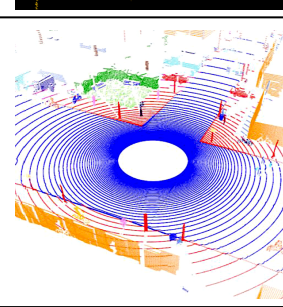 | 1× LiDAR (64-beam)
1× simulation suite | 32 semantic classes
198396 total samples | Synthetic
High-resolution point cloud
Dense annotation
Transfer learning |
| Synth4D [82] | 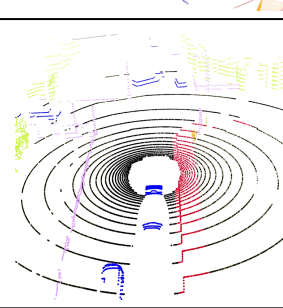 | 1× LiDAR (64-beam)
1× LiDAR (32-beam)
1× simulation suite | 22 semantic classes
10000 training samples
10000 validation samples | Synthetic
Low-resolution point cloud
Dense annotation
Transfer learning |
| DAPS-3D [51] | 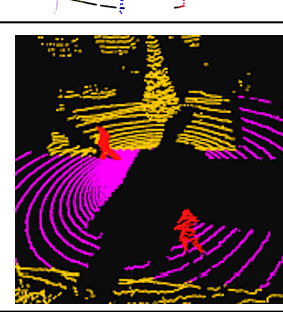 | 3× LiDAR (64-beam)
1× simulation suite | 4 semantic classes
19061 training samples
3995 validation samples | Semi-synthetic
High-resolution point cloud
Dense annotation
Transfer learning |
| nuScenes-C [53] | 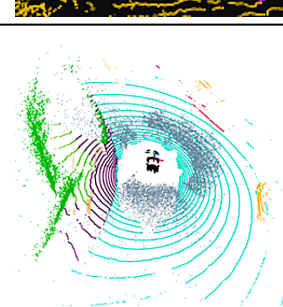 | 1× LiDAR (32-beam)
6× RGB Camera
5× RADAR
1× IMU & GPS | 16 semantic classes
144456 validation samples | Synthetic
Low-resolution point cloud
Dense annotation
Robustness |

- [5]**RELLIS-3D** [49]: The RELLIS-3D dataset is a multimodal dataset collected in an off-road environment from the Rellis Campus of Texas A&M University. It consists of 13556 LiDAR scans from 5 traversal sequences. The point-wise annotations are initialized by using the camera-LiDAR calibration to project the more than 6000 image annotations onto the point clouds. In this work, we use the LiDAR point clouds from RELLIS-3D as a downstream task to validate the generalizability of pertaining methods. More details of this dataset can be found at `http://www.unmannedlab.org/research/RELLIS-3D`.

- [6]**SemanticPOSS** [75]: The SemanticPOSS dataset is a relatively small-scale point cloud dataset with an emphasis on dynamic instances. It includes 2988 scans collected by a Hesai Pandora LiDAR sensor, which is a 40-channel LiDAR with 0.33 degree vertical resolution, a forward-facing color camera, 4 wide-angle mono cameras covering 360 degrees around the ego-car. The data in this dataset was collected from the campus of Peking University. In this work, we use the LiDAR point clouds from SemanticPOSS as a downstream task to validate the generalizability of pertaining methods. More details of this dataset can be found at `https://www.poss.pku.edu.cn/semanticposs`.

- [7]**SemanticSTF** [102]: The SemanticSTF dataset is a small-scale collection of 2076 scans, where the data are borrowed from the STF dataset [5]. The scans are collected by a Velodyne HDL64 S3D LiDAR sensor and covered various adverse weather conditions, including 694 snowy, 637 dense-foggy, 631 light-foggy, and 114 rainy scans. The whole dataset is split into three sets: 1326 scans for training, 250 scans for validation, and 500 scans for testing. All three splits have similar proportions of scans from different weather conditions. In this work, we use the LiDAR point clouds from SemanticSTF as a downstream task to validate the generalizability of pertaining methods. More details of this dataset can be found at `https://github.com/xiaoaoran/SemanticSTF`.

- [8]**SynLiDAR** [100]: The SynLiDAR dataset contains synthetic point clouds captured from constructed virtual scenes using the Unreal Engine 4 simulator. In total, this dataset contains 13 LiDAR point cloud sequences with 198396 scans. As stated, the virtual scenes in SynLiDAR are constituted by physically accurate object models that are produced by expert modelers with the 3D-Max software. In this work, we use the LiDAR point clouds from SynLiDAR as a downstream task to validate the generalizability of pertaining methods. More details of this dataset can be found at `https://github.com/xiaoaoran/SynLiDAR`.

- [9]**Synth4D** [82]: The Synth4D dataset includes two subsets with point clouds captured by simulated Velodyne LiDAR sensors using the CARLA simulator. We use the Synth4D-nuScenes subset in our experiments. It is composed of around 20000 labeled point clouds captured by a virtual vehicle navigating in town, highway, rural area, and city. The label mappings are mapped to that of the nuScenes dataset. In this work, we use the LiDAR point clouds from Synth4D-nuScenes as a downstream task to validate the generalizability of pertaining methods. More details of this dataset can be found at `https://github.com/saltoricristiano/gipso-sfouda`.

- [10]**DAPS-3D** [51]: The DAPS-3D dataset consists of two subsets: DAPS-1 and DAPS-2; while the former is a semi-synthetic one with a larger scale, the latter is recorded during a real field trip of the cleaning robot to the territory of the VDNH Park in Moscow. Both subsets are with scans collected by a 64-line Ouster OS0 LiDAR sensor. We use the DAPS-1 subset in our experiments, which contains 11 LiDAR sequences with more than 23000 labeled point clouds. In this work, we use the LiDAR point clouds from DAPS-1 as a downstream task to validate the generalizability of pertaining methods. More details of this dataset can be found at `https://github.com/subake/DAPS3D`.

- [11]**nuScenes-C** [53]: The nuScenes-C dataset is one of the corruption sets in the Robo3D benchmark, which is a comprehensive benchmark heading toward probing the robustness of 3D detectors and segmentors under out-of-distribution scenarios against natural corruptions that occur in real-world environments. A total number of eight corruption types, stemming from severe weather conditions, external disturbances, and internal sensor failure, are considered, including ‘fog’, ‘wet ground’, ‘snow’, ‘motion blur’, ‘beam missing’, ‘crosstalk’, ‘incomplete echo’, and ‘cross-sensor’ scenarios. These corruptions are simulated with rules constrained by physical principles or engineering experiences. In this work, we use the LiDAR point clouds from nuScenes-C as a downstream task to validate the robustness of

Table B: The **statistics of superpixels** (for front-view cameras) generated by SLIC [1] and different vision foundation modes [50, 122, 111, 123]. The **horizontal axis** denotes the number of superpixels per image. The **vertical axis** denotes the frequency of occurrence.

| **Method** | Front Left | Front | Front Right |
|---|---|---|---|
| SLIC [1] | 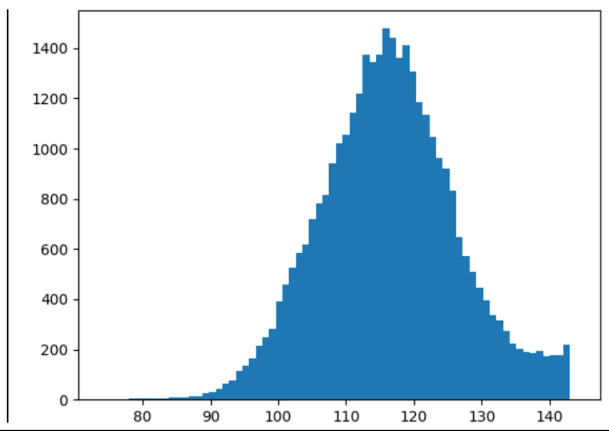 | 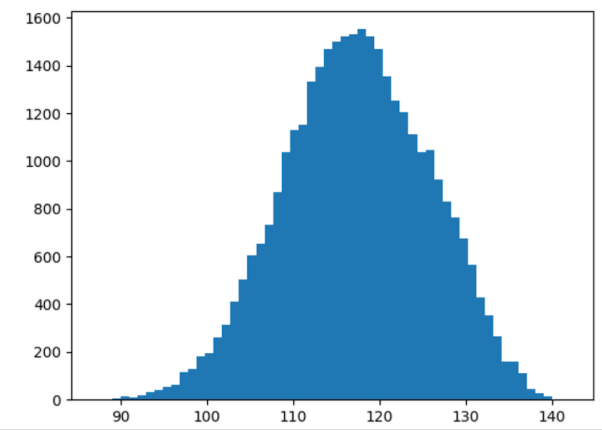 | 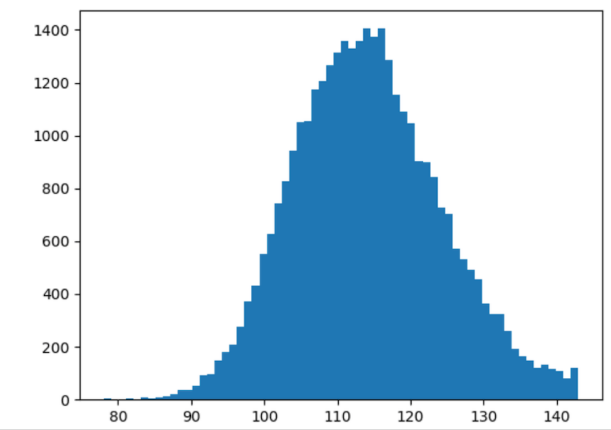 |
| SAM [50] | 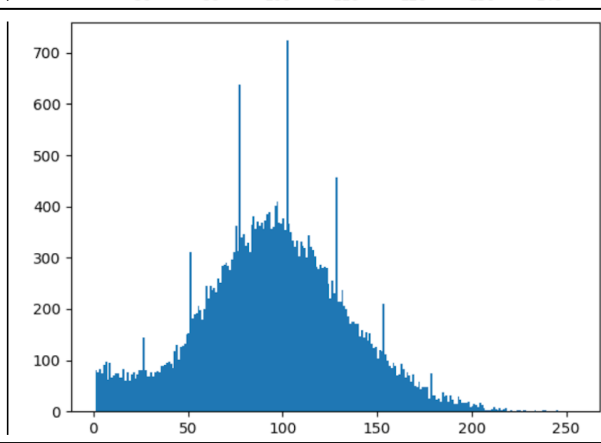 | 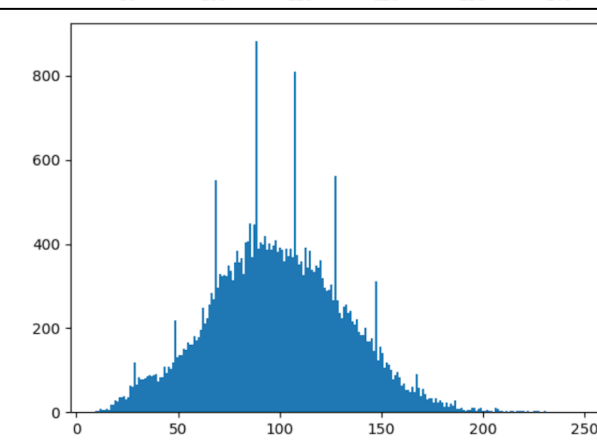 | 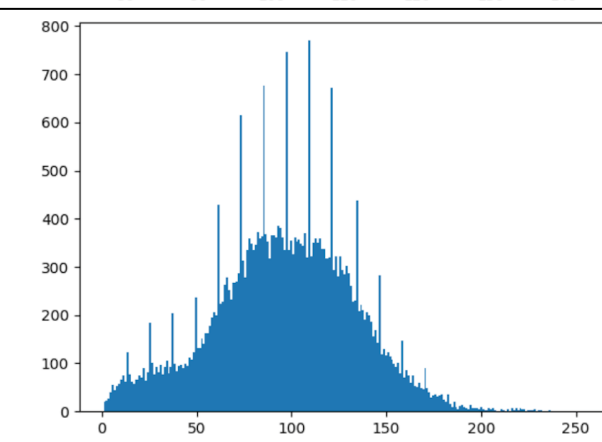 |
| X-Decoder [122] | 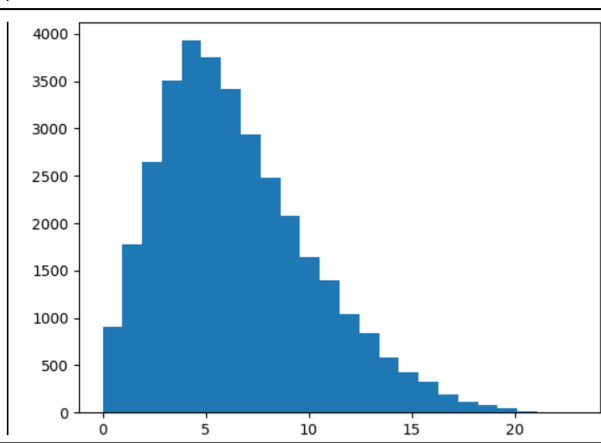 | 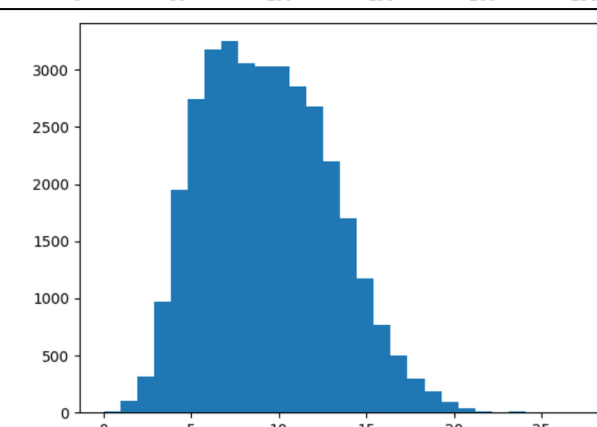 | 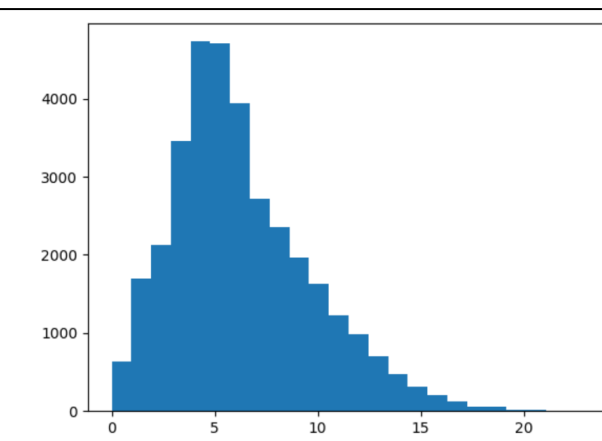 |
| OpenSeeD [111] | 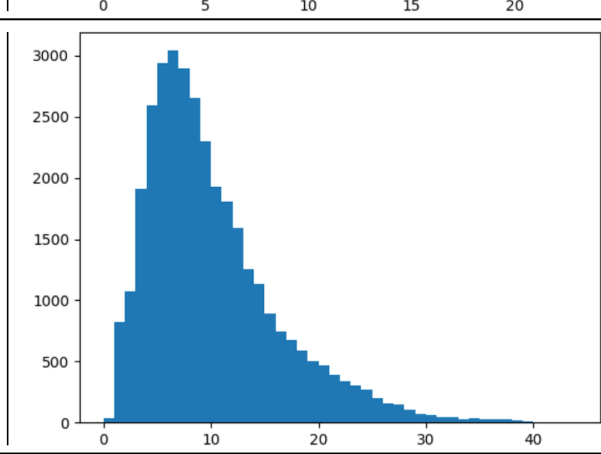 | 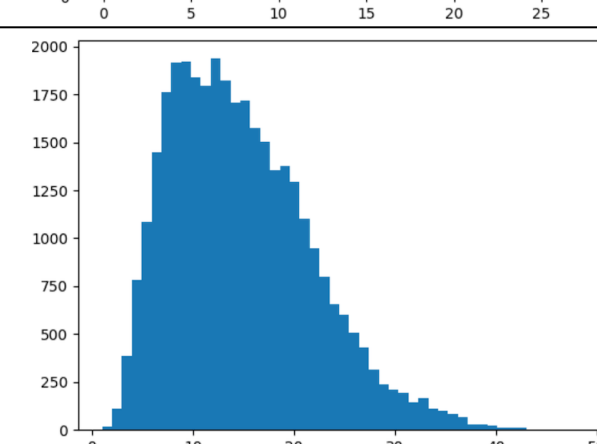 | 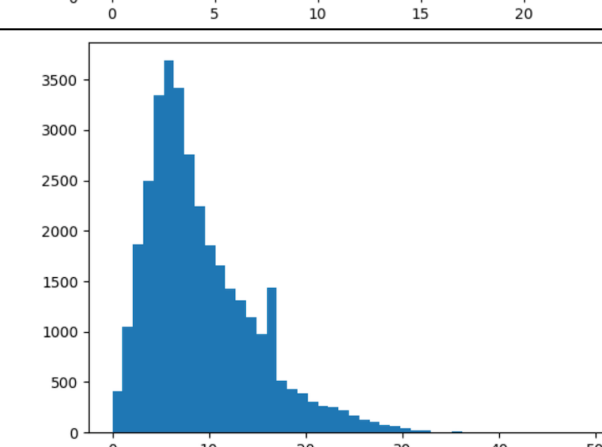 |
| SEEM [123] | 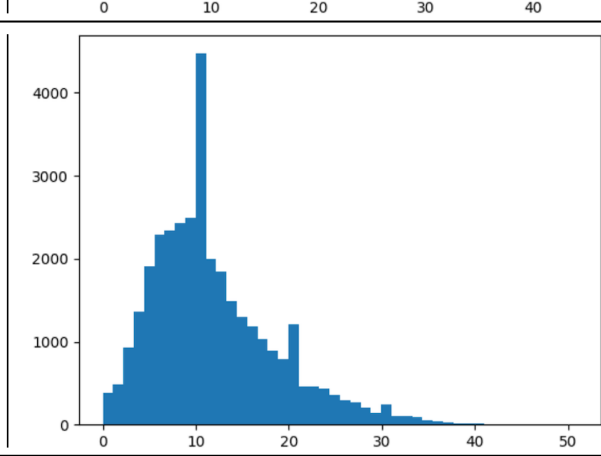 | 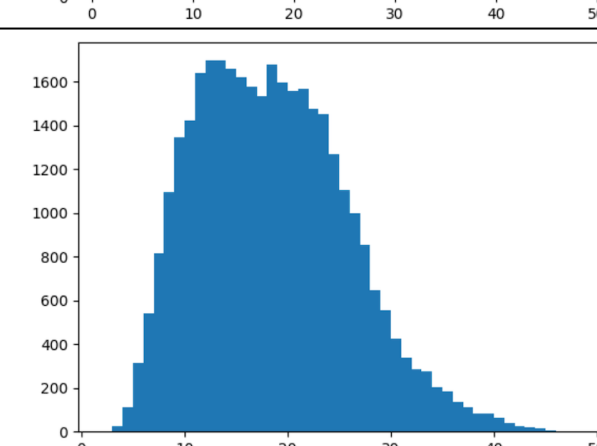 | 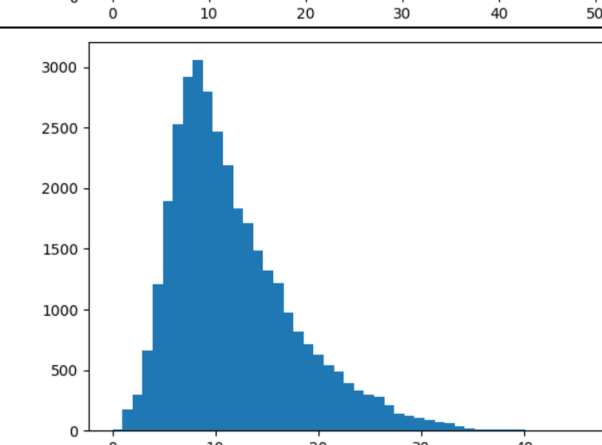 |

pertaining methods under out-of-distribution scenarios. More details of this dataset can be found at https://github.com/ldkong1205/Robo3D.

### A.2 Vision Foundation Models

In this work, we conduct comprehensive experiments on analyzing the effects brought by different vision foundation models (VFMs), compared to the traditional SLIC [1] method. Some statistical analyses of these different visual partition methods are shown in Table B and Table C.

- **SLIC** [1] (traditional method): The SLIC model, which stands for 'simple linear iterative clustering', is a popular choice for visual partitions of RGB images. It adapts a k-means clustering approach to generate superpixels, in an efficient manner, and offers good unsupervised partition abilities for many downstream tasks. The pursuit of adherence to boundaries and computational and memory efficiency allows SLIC to perform well on different image collections. In this work, we follow SLidR [85] and use SLIC to generate

Table C: The **statistics of superpixels** (for back-view cameras) generated by SLIC [1] and different vision foundation modes [50, 122, 111, 123]. The **horizontal axis** denotes the number of superpixels per image. The **vertical axis** denotes the frequency of occurrence.

| **Method** | Back Left | Back | Back Right |
|---|---|---|---|
| SLIC [1] | 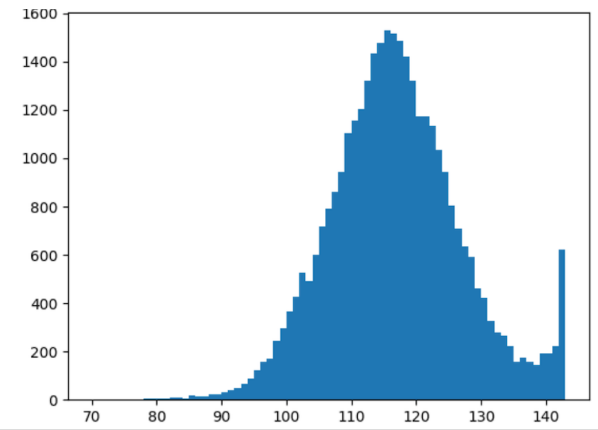 | 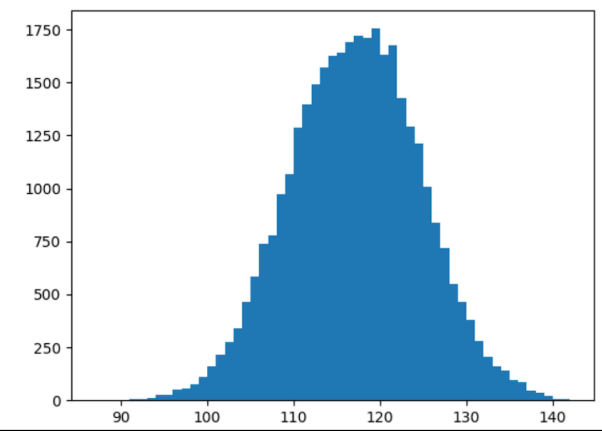 | 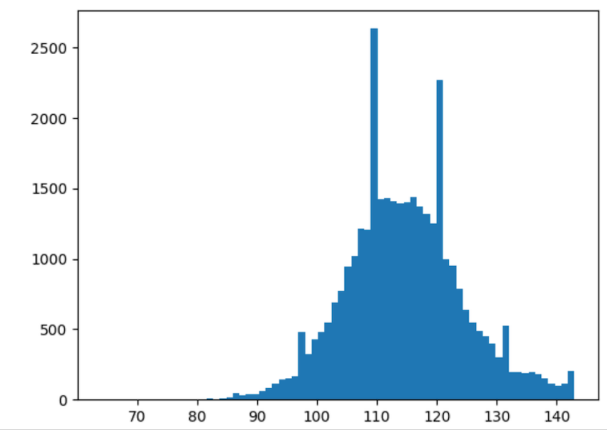 |
| SAM [50] | 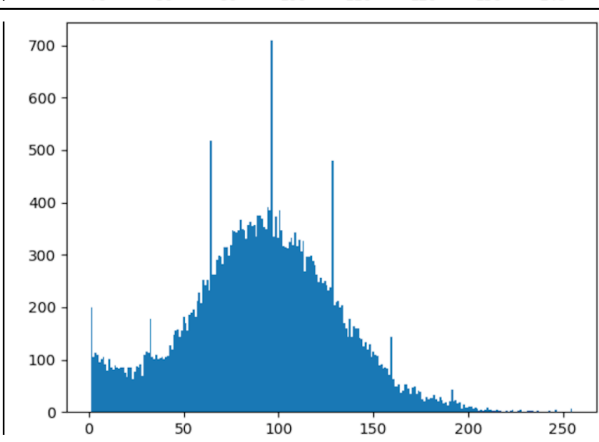 | 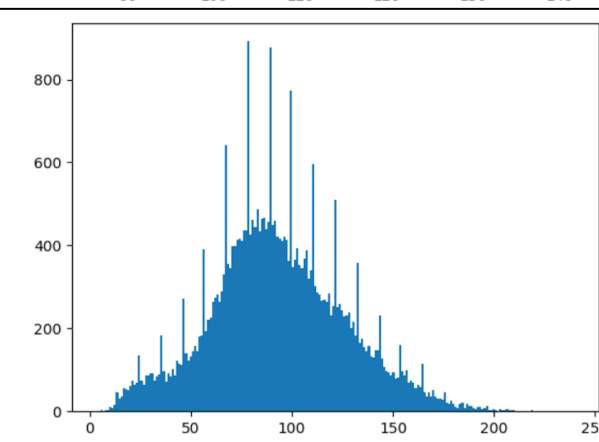 | 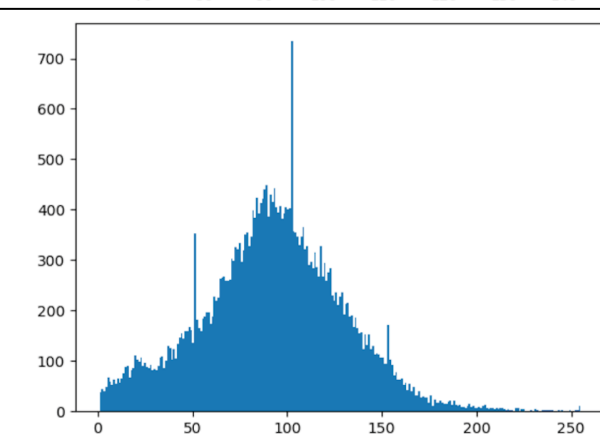 |
| X-Decoder [122] | 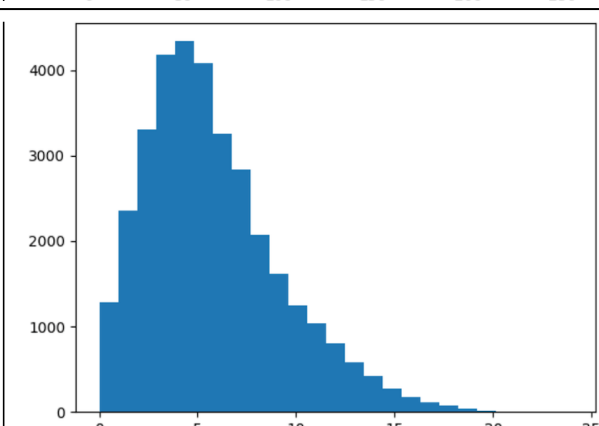 | 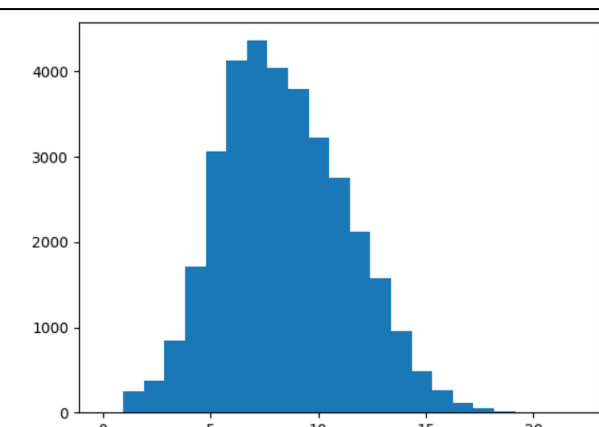 | 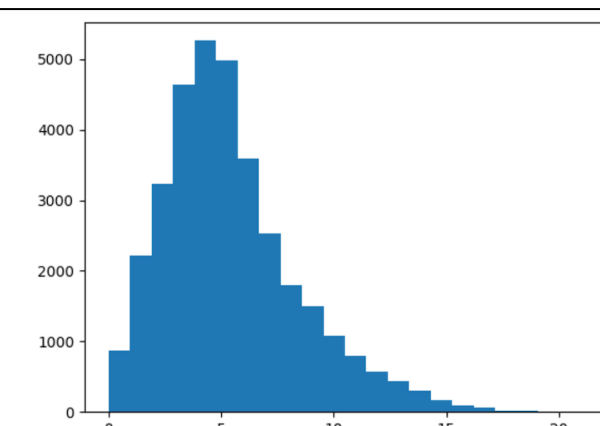 |
| OpenSeeD [111] | 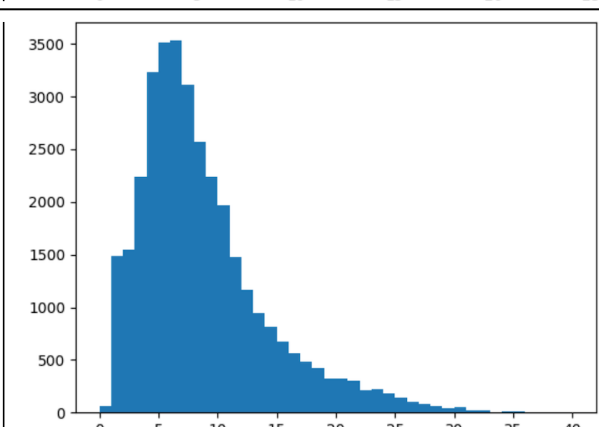 | 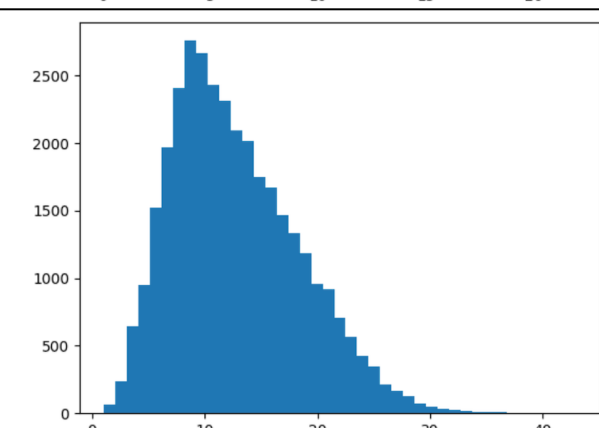 | 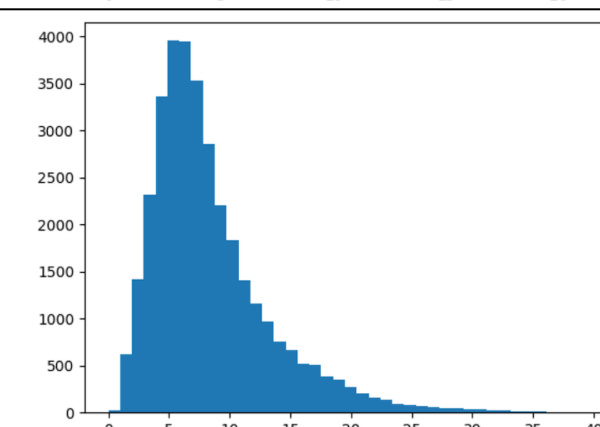 |
| SEEM [123] | 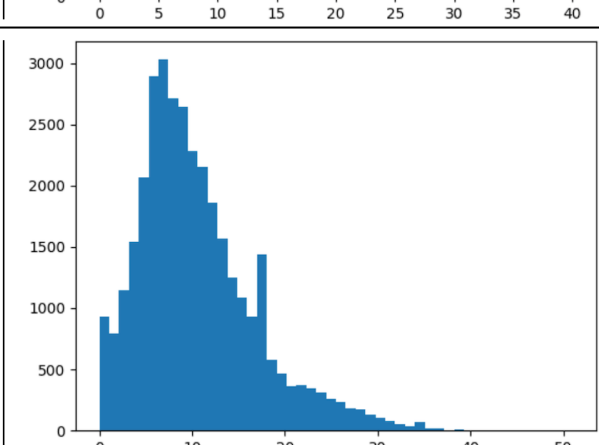 | 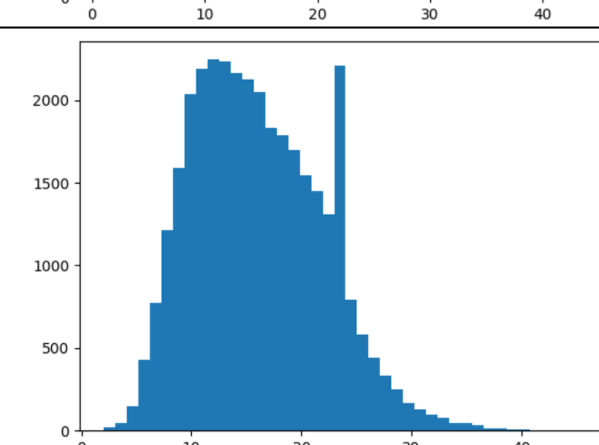 | 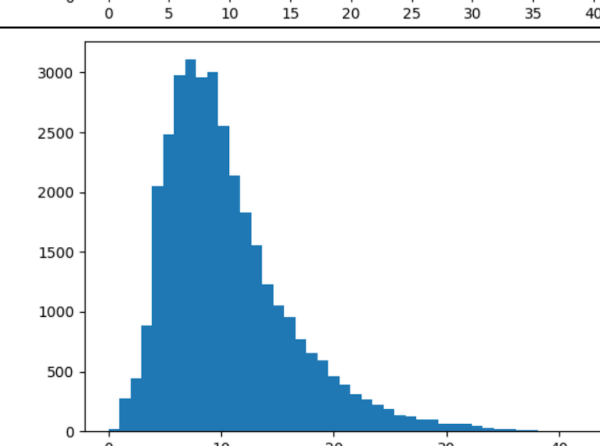 |

superpixels, with a fixed quota of 150 superpixels per image, and compare our framework with previous ones using SLIC superpixels. More details of this model can be found at https://github.com/valeoai/SLidR.

- **SAM** [50]: The Segment Anything Model (SAM) is a recent breakthrough towards zero-shot transferable visual understanding across a wide range of tasks. This model is trained on SA-1B, a large-scale dataset with over 1 billion masks on 11M licensed and privacy-respecting images. As a result, SAM is able to segment images, with either point, box, or mask prompts, across different domains and data distributions. In this work, we use a fixed SAM model with the ViT-H backbone (termed as `ViT-H-SAM` model) to generate superpixels. We use this pretrained model directly without any further fine-tuning. More details of this model can be found at https://github.com/facebookresearch/segment-anything.
- **X-Decoder** [122]: The X-Decoder model is a generalized decoding framework that can predict pixel-level segmentation and language tokens seamlessly. This model is pretrained on three types of data, including panoptic segmentation, image-text pairs, and referring segmentation. For the panoptic segmentation task, the model is trained on COCO2017,

which includes around 104k images for model training. In this work, we use a fixed X-Decoder model termed `BestSeg-Tiny` to generate superpixels. We use this pretrained model directly without any further fine-tuning. More details of this model can be found at https://github.com/microsoft/X-Decoder.

- **OpenSeeD** [111]: The OpenSeeD model is designed for open-vocabulary segmentation and detection, which jointly learns from different segmentation and detection datasets. This model consists of an image encoder, a text encoder, and a decoder with foreground, background, and conditioned mask decoding capability. The model is trained on COCO2017 and Objects365, under the tasks of panoptic segmentation and object detection, respectively. In this work, we use a fixed OpenSeeD model termed `openseed-swint-lang` to generate superpixels. We use this pretrained model directly without any further fine-tuning. More details of this model can be found at https://github.com/IDEA-Research/OpenSeeD.
- **SEEM** [123]: The SEEM model contributes a new universal interactive interface for image segmentation, where 'SEEM' stands for 'segment everything everywhere with multi-modal prompts all at once'. The newly designed prompting scheme can encode various user intents into prompts in a joint visual-semantic space, which possesses properties of versatility, compositionality, interactivity, and semantic awareness. As such, this model is able to generalize well to different image datasets, under a zero-shot transfer manner. Similar to X-Decoder, SEEM is trained on COCO2017, with a total number of 107k images used during model training. In this work, we use a fixed SEEM model termed `SEEM-oq101` to generate superpixels. We use this pretrained model directly without any further fine-tuning. More details of this model can be found at https://github.com/UX-Decoder/Segment-Everything-Everywhere-All-At-Once.

The quality of superpixels directly affects the performance of self-supervised representation learning. Different from the previous paradigm [85, 66], our ***Seal*** framework resorts to the recent VFMs for generating superpixels. Compared to the traditional SLIC method, these VFMs are able to generate semantically-aware superpixels that represent coherent semantic meanings of objects and backgrounds around the ego-vehicle.

As been verified in our experiments, these semantic superpixels have the ability to ease the "over-segment" problem in current self-supervised learning frameworks [85, 66] and further improve the performance for both linear probing and downstream fine-tuning.

The histograms shown in Table B and Table C verify that the number of superpixels per image of VFMs is much smaller than that of SLIC. This brings two notable advantages: *i)* Since semantically similar objects and backgrounds are grouped together in semantic superpixels, the "self-conflict" problem in existing approaches is largely mitigated, which directly boosts the quality of representation learning. *ii)* Since the embedding length $D$ of the superpixel embedding features $\mathbf{Q} \in \mathbb{R}^{M \times D}$ and superpoint embedding features $\mathbf{K} \in \mathbb{R}^{M \times D}$ directly relates to computation overhead, a reduction (*e.g.*, around 150 superpixels per image in SLIC [1] and around 25 superpixels per image in X-Decoder [122], OpenSeeD [111], and SEEM [123]) on $D$ would allow us to train the segmentation models in a more efficient manner.

Some typical examples of our generated superpixels and their corresponding superpoints are shown in Fig. A, Fig. B, Fig. C, and Fig. D. As will be shown in Section B, our use of semantic superpixels generated by VFMs brings not only performance gains but also a much faster convergence rate during the model pretraining stage.

### A.3 Implementation Detail

#### A.3.1 Data Split

For **model pertaining**, we follow the SLidR protocol [85] in data splitting. Specifically, the nuScenes [26] dataset consists of 700 training scenes in total, 100 of which are kept aside, which constitute the SLidR mini-val split. All models are pretrained using all the scans from the 600 remaining training scenes. The 100 scans in the mini-val split are used to find the best possible hyperparameters. The trained models are then validated on the official nuScenes validation set, without any kind of test-time augmentation or model ensemble. This is to ensure a fair comparison with previous works and also in line with the practical requirements.

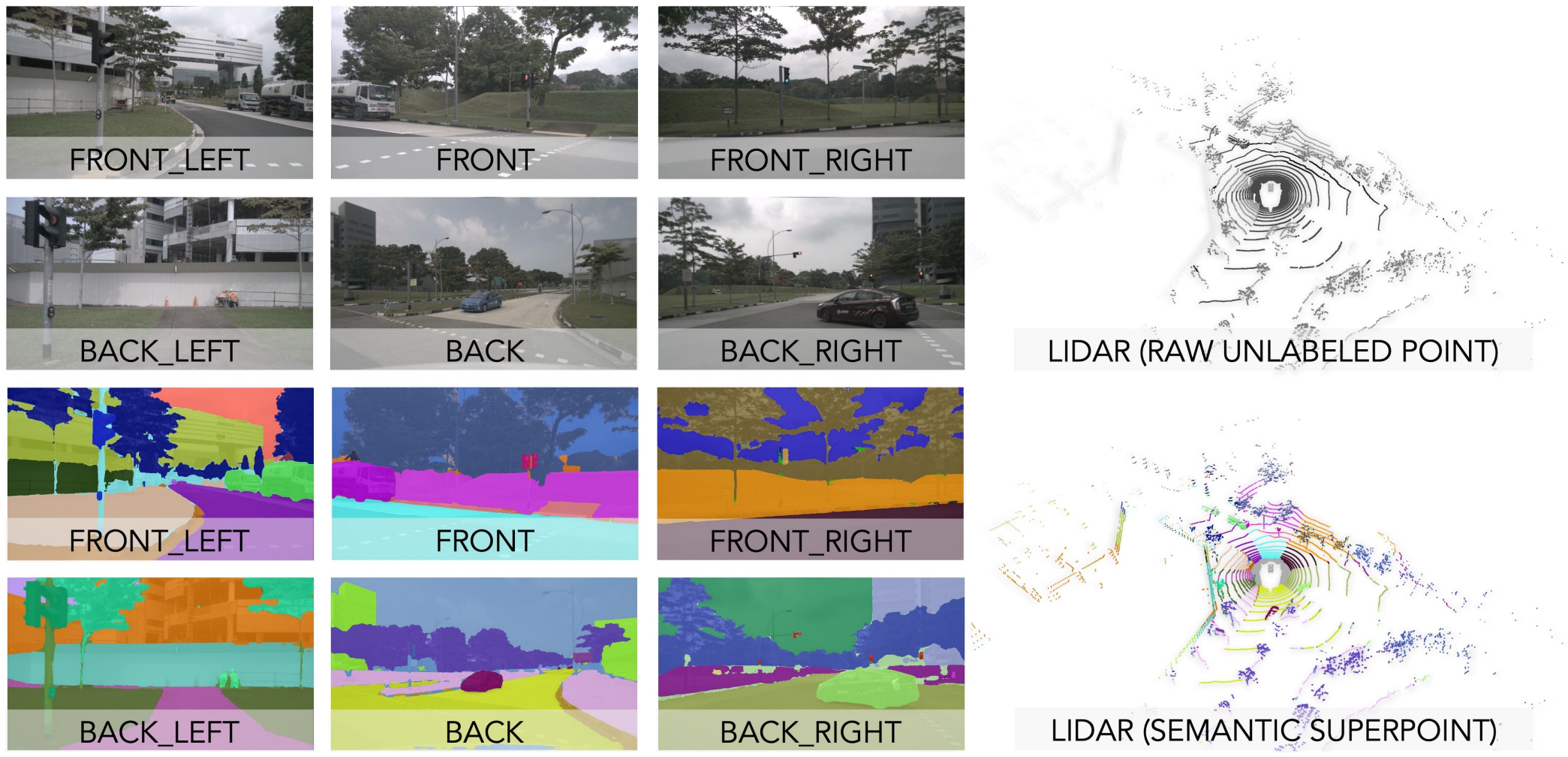

Figure A: Illustration of the **semantic superpixel-to-superpoint transformation** in the proposed ***Seal*** framework. **[Row 1 & 2]** The raw data captured by the multi-view camera and LiDAR sensors. **[Row 3 & 4]** The semantic superpixel on the camera images and superpoint formed by projecting superpixel into the point cloud via camera-LiDAR correspondence. The superpixels are generated using SEEM [123]. Each color represents one distinct segment. Best viewed in color.

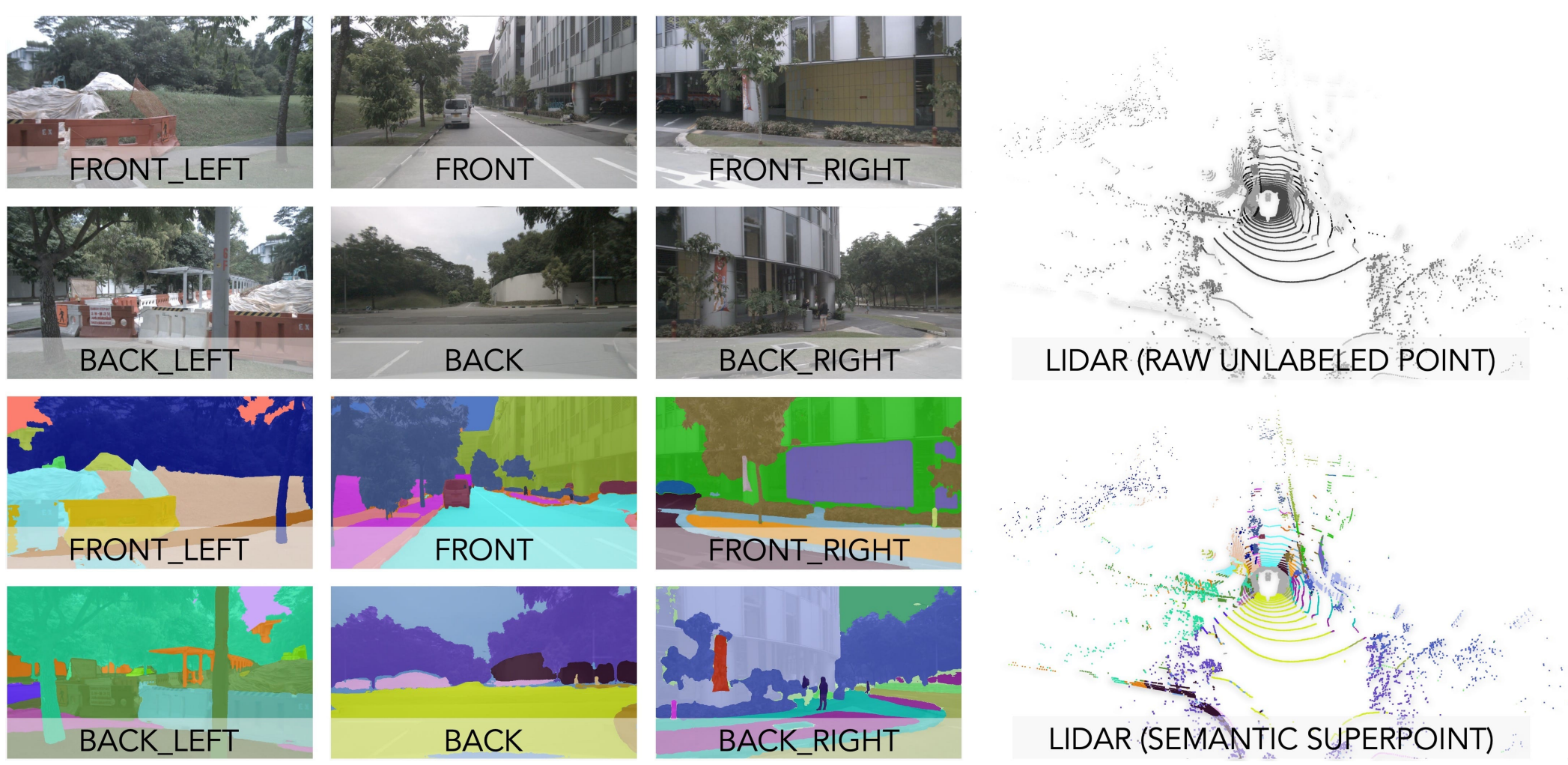

Figure B: Illustration of the **semantic superpixel-to-superpoint transformation** in the proposed ***Seal*** framework. **[Row 1 & 2]** The raw data captured by the multi-view camera and LiDAR sensors. **[Row 3 & 4]** The semantic superpixel on the camera images and superpoint formed by projecting superpixel into the point cloud via camera-LiDAR correspondence. The superpixels are generated using SEEM [123]. Each color represents one distinct segment. Best viewed in color.

For **linear probing**, the pretrained 3D network $F_{\theta_p}$ is frozen with a trainable point-wise linear classification head which is trained for 50 epochs on an A100 GPU with a learning rate of 0.05, and batch size is 16 on the nuScenes train set for all methods.

For **downstream fine-tuning** tasks, we stick with the common practice in SLidR [85] whenever possible. The detailed data split strategies are summarized as follows.

- For fine-tuning on **nuScenes** [26], we follow the SLidR protocol to split the train set of nuScenes to generate 1%, 5%, 10%, 25%, and 100% annotated scans for the training subset.
- For fine-tuning on **DAPS-3D** [51], we take sequences '38-18_7_72_90' as the training set and '38-18_7_72_90', '42-48_10_78_90', and '44-18_11_15_32' as the validation set.

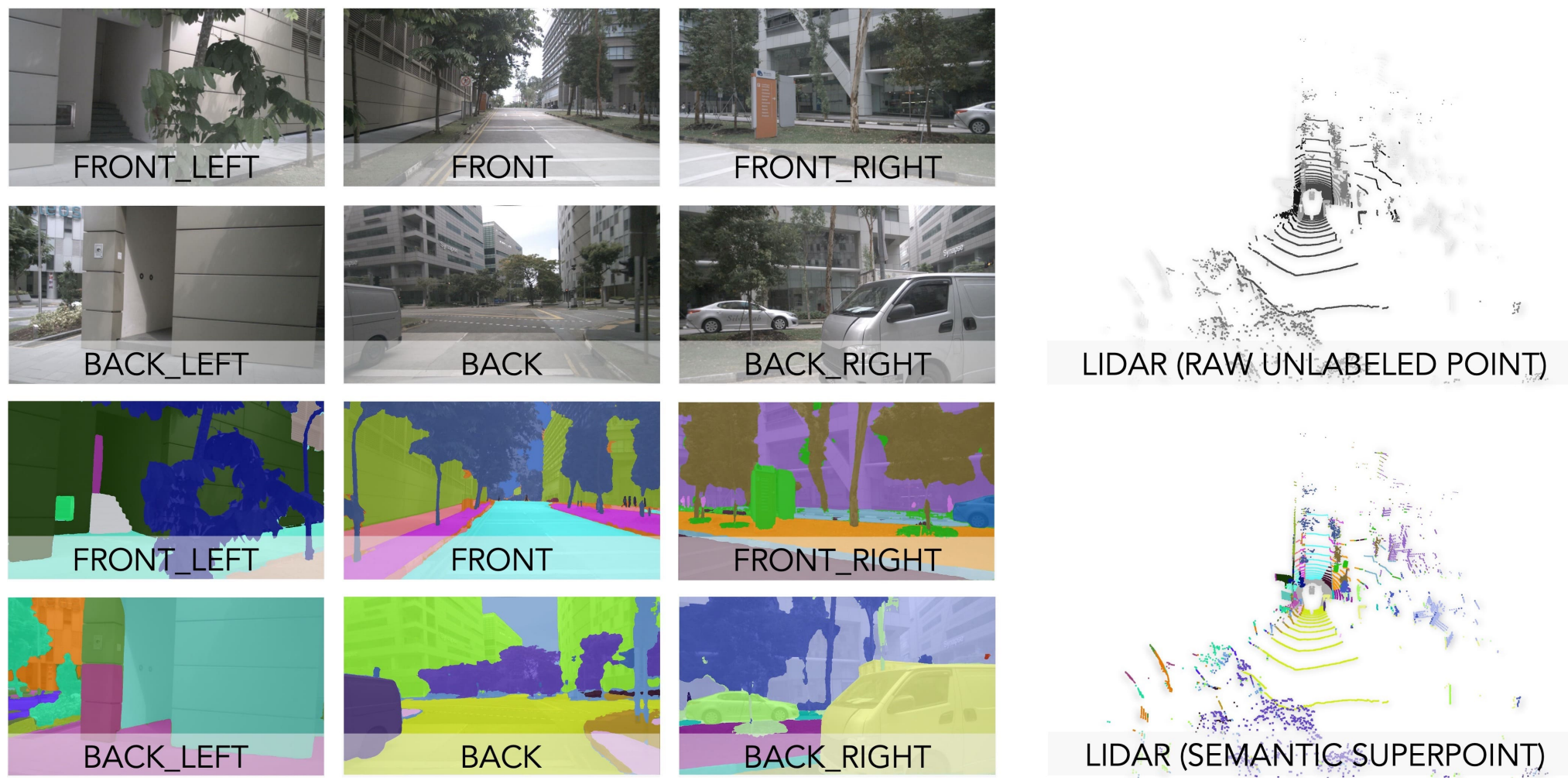

Figure C: Illustration of the **semantic superpixel-to-superpoint transformation** in the proposed ***Seal*** framework. **[Row 1 & 2]** The raw data captured by the multi-view camera and LiDAR sensors. **[Row 3 & 4]** The semantic superpixel on the camera images and superpoint formed by projecting superpixel into the point cloud via camera-LiDAR correspondence. The superpixels are generated using SEEM [123]. Each color represents one distinct segment. Best viewed in color.

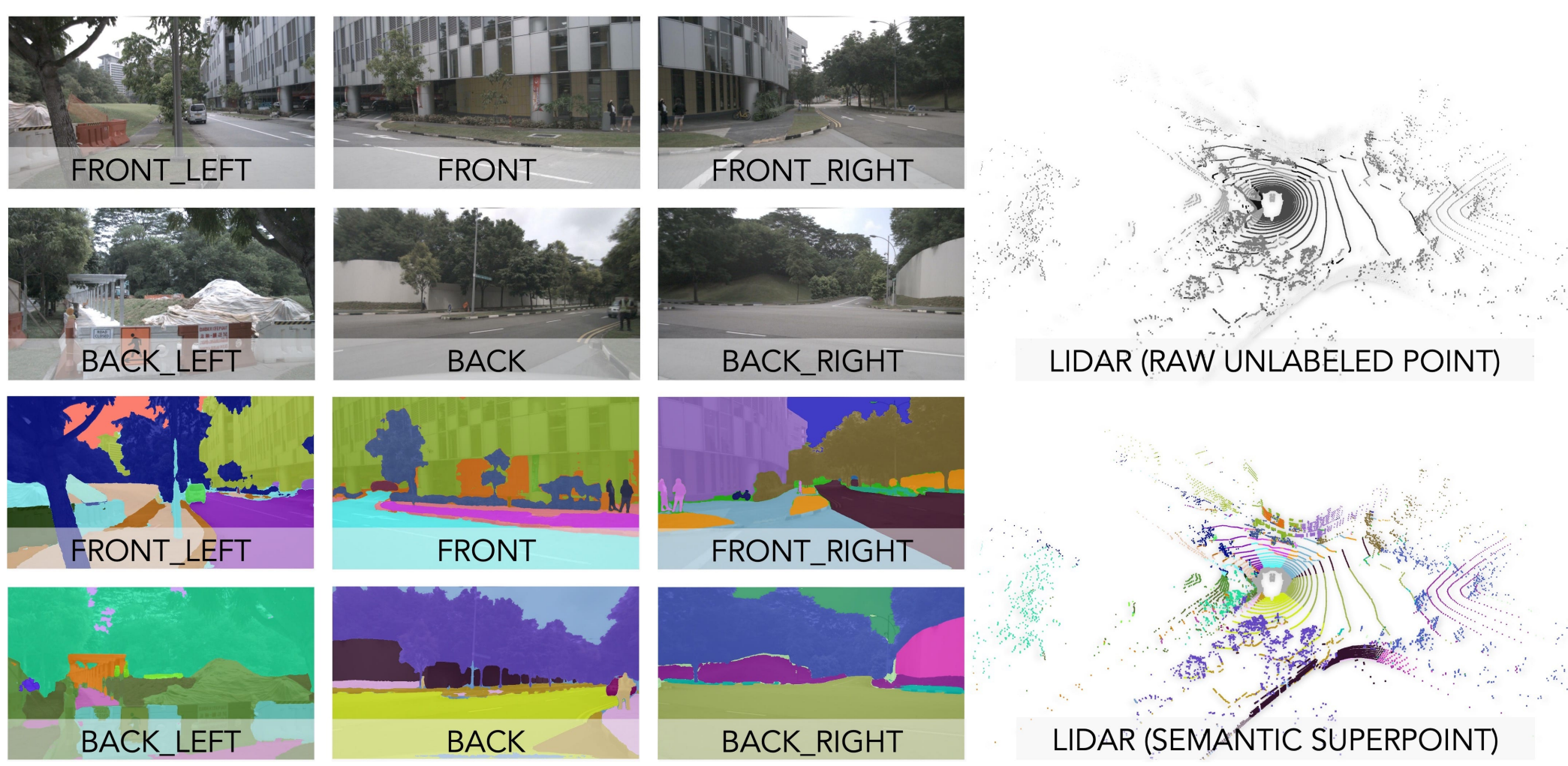

Figure D: Illustration of the **semantic superpixel-to-superpoint transformation** in the proposed ***Seal*** framework. **[Row 1 & 2]** The raw data captured by the multi-view camera and LiDAR sensors. **[Row 3 & 4]** The semantic superpixel on the camera images and superpoint formed by projecting superpixel into the point cloud via camera-LiDAR correspondence. The superpixels are generated using SEEM [123]. Each color represents one distinct segment. Best viewed in color.

- For fine-tuning on **SynLiDAR** [100], we use the sub-set which is a uniformly downsampled collection from the whole dataset.
- For fine-tuning on **SemanticPOSS** [75], we use sequences 00 and 01 as *half* of the annotated training scans and use sequences 00 to 05, except 02 for validation to create *full* of the annotated training samples.
- For fine-tuning on **SemanticKITTI** [3], **Waymo Open** [88], **ScribbleKITTI** [95], **RELLIS-3D** [49], **SemanticSTF** [102], and **Synth4D** [82], we follow the SLidR protocol to create 1%, 10%, *half*, or *full* split of the annotated training scans, *e.g.*, one scan is taken every 100 frame from the training set to get 1% of the labeled training samples. Notably, the point

cloud segmentation performance in terms of IoU is reported on the official validation sets for all the above-mentioned datasets.

### A.3.2 Experimental Setup

In our experiments, we fine-tune the entire 3D network on the semantic segmentation task using a linear combination of the cross-entropy loss and the Lovász-Softmax loss [4] as training objectives on a single A100 GPU. For the few-shot semantic segmentation tasks, the 3D networks are fine-tuned for 100 epochs with a batch size of 10 for the SemanticKITTI [3], Waymo Open [88], ScribbleKITTI [95], RELLIS-3D [49], SemanticSTF [102], SemanticPOSS [75], DAPS-3D [51], SynLiDAR [100], and Synth4D [82] datasets.

For the nuScenes [26] dataset, we fine-tune the 3D network for 100 epochs with a batch size of 16 while training on the $1\%$ annotated scans. The 3D network train on the other portions of nuScenes is fine-tuned for 50 epochs with a batch size of 16. We adopt different learning rates on the 3D backbone $F_{\theta_p}$ and the classification head, except for the case that $F_{\theta_p}$ is randomly initialized. The learning rate of $F_{\theta_p}$ is set as $0.05$ and the learning rate of the classification head is set as $2.0$, respectively, for all the above-mentioned datasets except nuScenes. On the nuScenes dataset, the learning rate of $F_{\theta_p}$ is set as $0.02$.

We train our framework using the SGD optimizer with a momentum of $0.9$, a weight decay of $0.0001$, and a dampening ratio of $0.1$. The cosine annealing learning rate strategy is adopted which decreases the learning rate from its initial value to zero at the end of the training.

### A.3.3 Data Augmentation

For the **model pretraining**, we apply two sets of data augmentations on the point cloud and the multi-view image, respectively, and update the point-pixel correspondences after each augmentation by following SLidR [85].

- Regarding the point cloud, we adopt a random rotation around the $z$-axis and flip the $x$-axis or $y$-axis with a $50\%$ probability. Besides, we randomly drop the cuboid where the length of each side covers no more than $10\%$ range of point coordinates on the corresponding axis, and the cuboid center is located on a randomly chosen point in the point cloud. We also ensure that the dropped cuboid retains at least $1024$ pairs of points and pixels; otherwise, we select another new cuboid instead. For the temporal frame, we apply the same data augmentation to it.

- Regarding the multi-view image, we apply a horizontal flip with a $50\%$ probability and a cropped resize which reshapes the image to $416 \times 224$. Before resizing, the random crop fills at least $30\%$ of the available image space with a random aspect ratio between $14:9$ and $17:9$. If this random cropping does not preserve at least $1024$ or $75\%$ of the pixel-point pairs, a different crop is chosen.

For the **downstream fine-tuning** tasks, we apply a random rotation around the $z$-axis and flip the $x$-axis or $y$-axis with a $50\%$ probability for all the points in the point cloud. If the results are mentioned with LaserMix [55] augmentation, we augment the point clouds via the LaserMix before the above data augmentations. We report both results in the main paper so that we can fairly compare the point cloud segmentation performance with previous works.

### A.3.4 Model Configuration

For the **model pretraining**, the 3D backbone $F_{\theta_p}$ is a Minkowski U-Net [19] with $3 \times 3 \times 3$ kernels; while the 2D image encoder $G_{\theta_i}$ is a ResNet-50 [33] initialized with 2D self-supervised pretrained model of MoCoV2 [12]. These configurations are kept the same as SLidR [85]. The channel dimension of $G_{\theta_i}$ head and $F_{\theta_p}$ head is set to 64. For the **linear probing** task, we adopt a linear classification head, as mentioned in previous sections. For the **downstream fine-tuning** tasks, the same 3D backbone $F_{\theta_p}$, *i.e.*, the Minkowski U-Net [19] with $3 \times 3 \times 3$ kernels, is used.

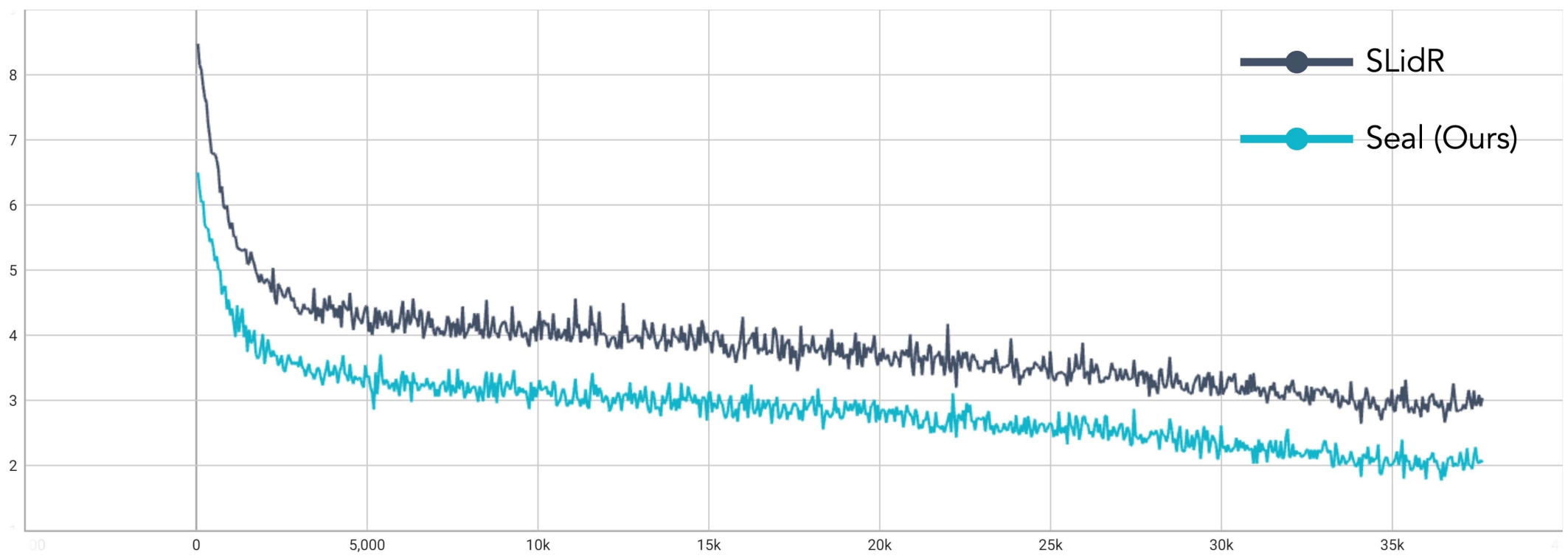

Figure E: The convergence rate comparison between SLidR [85] and the proposed ***Seal*** framework.

# B Additional Quantitative Result

## B.1 Self-Supervised Learning

We report the complete results (*i.e.*, the class-wise IoU scores) for the **linear probing** and **downstream fine-tuning** tasks shown in the main paper. We report the official results from previous works whenever possible. We also report our reproduced results for random initialization, PPKT [65], and SLidR [85]. Specifically, the complete results on the nuScenes [26], SemanticKITTI [3], Waymo Open [88], and Synth4D [82] datasets are shown in Table D, Table E, Table F, and Table G, respectively. We observe that ***Seal*** constantly outperforms prior methods for most semantic classes on all datasets.

We also compare the convergence rate of ***Seal*** with SLidR [85] in Fig. E. As can be seen, our framework is able to converge faster with the use of semantic superpixels, where these higher qualitative contrastive samples gradually form a much more coherent optimization landscape.

## B.2 Robustness Probing

We report the complete results of the robustness evaluation experiments in the main paper, including the per-corruption CE scores and the per-corruption RR scores. Specifically, the complete results on the nuScenes-C [53] dataset are shown in Table H and Table I. We observe that ***Seal*** is superior to previous methods in terms of both CE and RR metrics.

# C Additional Qualitative Result

## C.1 Cosine Similarity

We provide more examples for the cosine similarity study in Fig. F, Fig. G, and Fig. H.

## C.2 Downstream Tasks

We provide more qualitative results for downstream fine-tuning tasks in Fig. I, Fig. J, and Fig. K.

Table D: The **per-class IoU scores** of different pretraining methods pretrained on *nuScenes* [26] and linear probed or fine-tuned on different proportions (1%, 5%, 10%, 25%, and Full) of the *nuScenes* [26] data. Symbol ¶ denotes our reproduced results and the remaining are reported scores. All IoU scores are given in percentage (%). The **best** mIoU score is highlighted in **bold**.

| **Method** | mIoU | barrier | bicycle | bus | car | con. veh. | motor | ped | traf. cone | trailer | truck | driv. surf. | other flat | sidewalk | terrain | manmade | veg |
|---|---|---|---|---|---|---|---|---|---|---|---|---|---|---|---|---|---|
| **Linear Probing** | | | | | | | | | | | | | | | | | |
| Random ¶ | 8.4 | 0.5 | 0.0 | 0.0 | 3.9 | 0.0 | 0.0 | 0.0 | 6.4 | 0.0 | 3.9 | 60.4 | 0.0 | 0.1 | 16.2 | 30.6 | 12.2 |
| PointContrast | 21.9 | - | - | - | - | - | - | - | - | - | - | - | - | - | - | - | - |
| DepthContrast | 22.1 | - | - | - | - | - | - | - | - | - | - | - | - | - | - | - | - |
| PPKT | 35.9 | - | - | - | - | - | - | - | - | - | - | - | - | - | - | - | - |
| SLidR ¶ | 39.2 | 44.2 | 0.0 | 30.8 | 60.2 | 15.1 | 22.4 | 47.2 | 27.7 | 16.3 | 34.3 | 80.6 | 21.8 | 35.2 | 48.1 | 71.0 | 71.9 |
| ST-SLidR | 40.5 | - | - | - | - | - | - | - | - | - | - | - | - | - | - | - | - |
| **Seal (Ours)** | **45.0** | 54.7 | 5.9 | 3.6 | 61.7 | 18.9 | 28.8 | 48.1 | 31.0 | 22.1 | 39.5 | 83.8 | 35.4 | 46.7 | 56.9 | 74.7 | 74.7 |
| **Fine-tuning (1%)** | | | | | | | | | | | | | | | | | |
| Random | 30.3 | 0.0 | 0.0 | 8.1 | 65.0 | 0.1 | 6.6 | 21.0 | 9.0 | 9.3 | 25.8 | 89.5 | 14.8 | 41.7 | 48.7 | 72.4 | 73.3 |
| PointContrast | 32.5 | 0.0 | 1.0 | 5.6 | 67.4 | 0.0 | 3.3 | 31.6 | 5.6 | 12.1 | 30.8 | 91.7 | 21.9 | 48.4 | 50.8 | 75.0 | 74.6 |
| DepthContrast | 31.7 | 0.0 | 0.6 | 6.5 | 64.7 | 0.2 | 5.1 | 29.0 | 9.5 | 12.1 | 29.9 | 90.3 | 17.8 | 44.4 | 49.5 | 73.5 | 74.0 |
| PPKT | 37.8 | 0.0 | 2.2 | 20.7 | 75.4 | 1.2 | 13.2 | 45.6 | 8.5 | 17.5 | 38.4 | 92.5 | 19.2 | 52.3 | 56.8 | 80.1 | 80.9 |
| SLidR | 38.8 | 0.0 | 1.8 | 15.4 | 73.1 | 1.9 | 19.9 | 47.2 | 17.1 | 14.5 | 34.5 | 92.0 | 27.1 | 53.6 | 61.0 | 79.8 | 82.3 |
| ST-SLidR | 40.8 | 0.0 | 2.7 | 16.0 | 74.5 | 3.2 | 25.4 | 50.9 | 20.0 | 17.7 | 40.2 | 92.0 | 30.7 | 54.2 | 61.1 | 80.5 | 82.9 |
| **Seal (Ours)** | **45.8** | 0.0 | 9.4 | 32.6 | 77.5 | 10.4 | 28.0 | 53.0 | 25.0 | 30.9 | 49.7 | 94.0 | 33.7 | 60.1 | 59.6 | 83.9 | 83.4 |
| **Fine-tuning (5%)** | | | | | | | | | | | | | | | | | |
| Random | 47.8 | - | - | - | - | - | - | - | - | - | - | - | - | - | - | - | - |
| Random ¶ | 44.5 | 50.1 | 2.9 | 57.3 | 70.3 | 1.1 | 6.1 | 39.1 | 18.3 | 17.3 | 44.8 | 92.3 | 38.6 | 54.9 | 61.1 | 80.3 | 77.9 |
| PPKT ¶ | 52.7 | 56.1 | 8.2 | 65.3 | 79.0 | 9.1 | 15.5 | 54.3 | 34.5 | 26.7 | 58.6 | 93.2 | 44.1 | 63.1 | 64.8 | 85.1 | 83.6 |
| SLidR | 52.5 | - | - | - | - | - | - | - | - | - | - | - | - | - | - | - | - |
| ST-SLidR | 54.7 | - | - | - | - | - | - | - | - | - | - | - | - | - | - | - | - |
| **Seal (Ours)** | **55.6** | 61.0 | 7.4 | 70.4 | 82.4 | 11.9 | 30.4 | 59.2 | 34.0 | 33.6 | 61.1 | 94.7 | 46.2 | 63.4 | 63.9 | 85.7 | 84.9 |
| **Fine-tuning (10%)** | | | | | | | | | | | | | | | | | |
| Random | 56.2 | - | - | - | - | - | - | - | - | - | - | - | - | - | - | - | - |
| Random ¶ | 53.2 | 56.8 | 5.2 | 66.3 | 74.5 | 5.7 | 36.1 | 49.5 | 38.2 | 29.2 | 54.4 | 94.0 | 47.7 | 61.4 | 67.6 | 82.8 | 81.1 |
| PPKT ¶ | 60.3 | 64.0 | 12.0 | 67.8 | 77.6 | 16.0 | 56.8 | 63.3 | 49.8 | 28.3 | 56.3 | 94.1 | 62.7 | 66.4 | 68.7 | 85.9 | 85.4 |
| SLidR | 59.8 | - | - | - | - | - | - | - | - | - | - | - | - | - | - | - | - |
| ST-SLidR | 60.8 | - | - | - | - | - | - | - | - | - | - | - | - | - | - | - | - |
| **Seal (Ours)** | **63.0** | 64.9 | 15.1 | 78.9 | 83.5 | 22.5 | 61.0 | 63.0 | 51.5 | 37.4 | 65.2 | 95.2 | 59.4 | 67.7 | 69.6 | 86.2 | 86.4 |
| **Fine-tuning (25%)** | | | | | | | | | | | | | | | | | |
| Random | 65.5 | - | - | - | - | - | - | - | - | - | - | - | - | - | - | - | - |
| Random ¶ | 63.0 | 64.9 | 15.1 | 78.9 | 83.5 | 22.5 | 61.0 | 63.0 | 51.5 | 37.4 | 65.2 | 95.2 | 59.4 | 67.7 | 69.6 | 86.2 | 86.4 |
| PPKT ¶ | 67.1 | 63.7 | 12.1 | 87.4 | 85.2 | 42.0 | 61.2 | 69.6 | 54.7 | 50.1 | 74.9 | 95.8 | 64.6 | 69.9 | 70.4 | 87.4 | 85.1 |
| SLidR | 66.9 | - | - | - | - | - | - | - | - | - | - | - | - | - | - | - | - |
| ST-SLidR | 67.7 | - | - | - | - | - | - | - | - | - | - | - | - | - | - | - | - |
| **Seal (Ours)** | **68.4** | 67.4 | 15.5 | 90.5 | 85.1 | 40.0 | 62.3 | 68.1 | 58.3 | 54.5 | 76.0 | 95.8 | 65.6 | 69.8 | 71.0 | 87.9 | 86.7 |
| **Fine-tuning (Full)** | | | | | | | | | | | | | | | | | |
| Random | 74.7 | - | - | - | - | - | - | - | - | - | - | - | - | - | - | - | |
| Random ¶ | 74.9 | 77.4 | 34.4 | 90.4 | 86.5 | 51.1 | 78.4 | 77.0 | 64.9 | 67.8 | 77.7 | 96.6 | 72.2 | 74.3 | 73.2 | 89.6 | 86.7 |
| PPKT ¶ | 74.5 | 75.4 | 37.4 | 91.7 | 86.1 | 45.3 | 77.3 | 75.6 | 65.1 | 65.0 | 79.9 | 96.6 | 72.1 | 74.6 | 73.4 | 89.2 | 86.6 |
| SLidR | 74.8 | - | - | - | - | - | - | - | - | - | - | - | - | - | - | - | - |
| ST-SLidR | 75.1 | - | - | - | - | - | - | - | - | - | - | - | - | - | - | - | - |
| **Seal (Ours)** | **75.6** | 76.7 | 26.0 | 93.4 | 86.1 | 55.8 | 82.7 | 77.3 | 66.0 | 67.9 | 83.9 | 96.5 | 73.6 | 74.3 | 73.2 | 89.3 | 87.1 |

Table E: The **per-class IoU scores** of different pretraining methods pretrained on *nuScenes* [26] and fine-tuned on 1% of the *SemanticKITTI* [3] data. Symbol ¶ denotes our reproduced results and the remaining are reported scores. All IoU scores are given in percentage (%). The **best** mIoU score is highlighted in **bold**.

| **Method** | mIoU | car | bicycle | motorcycle | truck | other-vehicle | person | bicyclist | motorcyclist | road | parking | sidewalk | other-ground | building | fence | vegetation | trunk | trunk | trunk | trunk |
|---|---|---|---|---|---|---|---|---|---|---|---|---|---|---|---|---|---|---|---|---|
| Random | 39.5 | 91.2 | 0.0 | 9.4 | 8.0 | 10.7 | 21.2 | 0.0 | 0.0 | 89.4 | 21.4 | 73.0 | 1.1 | 85.3 | 41.1 | 84.9 | 50.1 | 71.4 | 55.4 | 37.6 |
| PPKT | 43.9 | 91.3 | 1.9 | 11.2 | 23.1 | 12.1 | 27.4 | 37.3 | 0.0 | 91.3 | 27.0 | 74.6 | 0.3 | 86.5 | 38.2 | 85.3 | 58.2 | 71.6 | 57.7 | 40.1 |
| SLidR | 44.6 | 92.2 | 3.0 | 17.0 | 22.4 | 14.3 | 36.0 | 22.1 | 0.0 | 91.3 | 30.0 | 74.7 | 0.2 | 87.7 | 41.2 | 85.0 | 58.5 | 70.4 | 58.3 | 42.4 |
| ST-SLidR | 44.7 | - | - | - | - | - | - | - | - | - | - | - | - | - | - | - | - | - | - | - |
| **Seal** | **46.6** | 92.3 | 14.9 | 18.7 | 16.1 | 23.7 | 43.0 | 34.4 | 0.0 | 91.3 | 27.2 | 75.3 | 0.7 | 85.7 | 38.8 | 85.1 | 61.9 | 71.3 | 57.7 | 47.7 |

Table F: The **per-class IoU scores** of different pretraining methods pretrained on *nuScenes* [26] and fine-tuned on 1% of the *Waymo Open* [88] data. Symbol ¶ denotes our reproduced results and the remaining are reported scores. All IoU scores are given in percentage (%). The **best** mIoU score is highlighted in **bold**.

| **Method** | mIoU | car | truck | bus | other vehicle | motorcyclist | bicyclist | pedestrian | sign | traffic light | pole | construction | bicycle | motorcycle | building | vegetation | tree trunk | curb | road | lane marker | other ground | walkable | sidewalk |
|---|---|---|---|---|---|---|---|---|---|---|---|---|---|---|---|---|---|---|---|---|---|---|---|
| Random | 39.4 | - | - | - | - | - | - | - | - | - | - | - | - | - | - | - | - | - | - | - | - | - | - |
| PPKT ¶ | 47.6 | 92.3 | 48.8 | 34.8 | 6.9 | 0.0 | 22.0 | 73.0 | 58.3 | 17.6 | 55.8 | 20.4 | 25.5 | 7.8 | 91.5 | 87.1 | 53.8 | 55.1 | 89.1 | 38.6 | 34.7 | 71.4 | 62.6 |
| SLidR | 47.1 | - | - | - | - | - | - | - | - | - | - | - | - | - | - | - | - | - | - | - | - | - | - |
| ST-SLidR | 44.9 | - | - | - | - | - | - | - | - | - | - | - | - | - | - | - | - | - | - | - | - | - | - |
| **Seal** | **49.3** | 92.5 | 52.6 | 33.5 | 3.7 | 0.0 | 31.9 | 73.1 | 61.0 | 23.5 | 57.0 | 31.4 | 20.1 | 12.4 | 91.4 | 87.1 | 53.2 | 57.2 | 89.5 | 38.8 | 34.9 | 72.1 | 68.7 |

Table G: The **per-class IoU scores** of different pretraining methods pretrained on *nuScenes* [26] and fine-tuned on 1% of the *Synth4D* [82] data. Symbol ¶ denotes our reproduced results and the remaining are reported scores. All IoU scores are given in percentage (%). The **best** mIoU score is highlighted in **bold**.

| **Method** | mIoU | building | fences | other | pedestrain | pole | roadlines | road | sidewalk | vegetation | vehicle | wall | traffic sign | sky | ground | bridge | rail track | guardrail | traffic light | static | dynamic | water | terrain |
|---|---|---|---|---|---|---|---|---|---|---|---|---|---|---|---|---|---|---|---|---|---|---|---|
| Random | 20.2 | 33.2 | 13.8 | 0.0 | 16.3 | 13.9 | 0.0 | 88.6 | 51.0 | 48.9 | 95.7 | 27.6 | 0.0 | 0.0 | 0.0 | 0.0 | 0.1 | 19.4 | 0.0 | 0.2 | 0.0 | 0.0 | 36.2 |
| PPKT ¶ | 61.1 | 84.4 | 67.1 | 62.9 | 77.4 | 75.0 | 2.2 | 92.0 | 76.3 | 92.5 | 99.2 | 73.4 | 67.3 | 0.0 | 62.2 | 39.6 | 83.5 | 66.8 | 0.0 | 63.6 | 55.3 | 37.4 | 66.0 |
| SLidR ¶ | 63.1 | 83.7 | 66.3 | 64.9 | 77.4 | 76.4 | 6.5 | 92.8 | 78.7 | 92.5 | 99.0 | 72.5 | 64.2 | 0.0 | 74.3 | 48.9 | 85.3 | 67.1 | 0.0 | 67.1 | 60.0 | 47.1 | 63.6 |
| **Seal** | **64.5** | 84.8 | 70.5 | 64.8 | 80.3 | 76.3 | 9.3 | 92.9 | 79.8 | 92.7 | 98.9 | 73.0 | 60.7 | 0.0 | 75.2 | 55.3 | 84.6 | 67.0 | 0.0 | 68.2 | 60.7 | 53.2 | 70.9 |

Table H: The **Corruption Error (CE) scores** of different pretraining methods pretrained on *nuScenes* [26] and probed under the eight out-of-distribution corruptions in the *nuScenes-C* dataset from the Robo3D benchmark [53]. All CE scores are given in percentage (%). The **best** CE score for each corruption type is highlighted in **bold**.

| | **Initial** | **Backbone** | mCE | Fog | Wet Ground | Snow | Motion Blur | Beam Missing | Crosstalk | Incomplete Echo | Cross-Sensor |
|---|---|---|---|---|---|---|---|---|---|---|---|
| **LP** | PPKT | MinkUNet$_{18}$ | 183.44 | 149.59 | 247.53 | 120.50 | 266.03 | 213.54 | 109.62 | 199.03 | 161.65 |
| | SLidR | MinkUNet$_{18}$ | 179.38 | 140.47 | 237.29 | **112.93** | 276.44 | 210.65 | **107.86** | 189.27 | 160.16 |
| | **Seal (Ours)** | MinkUNet$_{18}$ | **166.18** | **135.18** | **219.36** | 117.47 | **234.01** | **189.70** | 108.54 | **172.16** | **153.03** |
| **Full** | Random | PolarNet | 115.09 | 90.10 | 115.33 | 58.98 | 208.19 | 121.07 | 80.67 | 128.17 | 118.23 |
| | Random | FIDNet | 122.42 | 75.93 | 122.58 | 68.78 | 192.03 | 164.84 | 57.95 | 141.66 | 155.56 |
| | Random | CENet | 112.79 | 71.16 | 115.48 | 64.31 | 156.67 | 159.03 | **53.27** | 129.08 | 153.35 |
| | Random | WaffleIron | 106.73 | 94.76 | 99.92 | 84.51 | 152.35 | 110.65 | 91.09 | 106.41 | 114.15 |
| | Random | Cylinder3D | 105.56 | 83.22 | 111.08 | 69.74 | 165.28 | 113.95 | 74.42 | 110.67 | 116.15 |
| | Random | SPVCNN$_{18}$ | 106.65 | 88.42 | 105.56 | 98.78 | 156.48 | 110.11 | 86.04 | 104.26 | 103.55 |
| | Random | SPVCNN$_{34}$ | 97.45 | 95.21 | 99.50 | 97.32 | **95.34** | **98.73** | 97.92 | **96.88** | **98.74** |
| | Random | MinkUNet$_{18}$ | 112.20 | 79.90 | 112.50 | 74.64 | 181.47 | 120.76 | 93.22 | 111.58 | 123.53 |
| | PPKT | MinkUNet$_{18}$ | 105.64 | 77.63 | 104.22 | 68.60 | 160.95 | 114.81 | 86.71 | 108.96 | 123.20 |
| | SLidR | MinkUNet$_{18}$ | 106.08 | 74.61 | 106.13 | 73.75 | 165.09 | 118.02 | 79.08 | 107.38 | 124.57 |
| | **Seal (Ours)** | MinkUNet$_{18}$ | **92.63** | **58.97** | **98.47** | **56.63** | 127.25 | 108.20 | 57.97 | 110.95 | 122.63 |

Table I: The **Resilience Rate (RR) scores** of different pretraining methods pretrained on *nuScenes* [26] and probed under the eight out-of-distribution corruptions in the *nuScenes-C* dataset from the Robo3D benchmark [53]. All RR scores are given in percentage (%). The **best** RR score for each corruption type is highlighted in **bold**.

| | **Initial** | **Backbone** | mRR | Fog | Wet Ground | Snow | Motion Blur | Beam Missing | Crosstalk | Incomplete Echo | Cross-Sensor |
|---|---|---|---|---|---|---|---|---|---|---|---|
| **LP** | PPKT | MinkUNet$_{18}$ | **78.15** | 85.38 | **98.66** | 78.33 | 81.36 | **91.42** | **54.37** | 78.02 | **57.69** |
| | SLidR | MinkUNet$_{18}$ | 77.18 | **89.90** | 98.17 | **84.12** | 68.14 | 86.93 | 53.63 | 81.29 | 55.26 |
| | **Seal (Ours)** | MinkUNet$_{18}$ | 75.38 | 83.05 | 95.15 | 66.59 | **83.94** | 89.70 | 45.18 | **83.94** | 55.48 |
| **Full** | Random | PolarNet | 76.34 | 81.59 | 97.95 | **90.82** | 62.49 | 86.75 | 57.12 | 75.16 | 58.86 |
| | Random | FIDNet | 73.33 | 90.78 | 95.29 | 82.61 | 68.51 | 67.44 | **80.48** | 68.31 | 33.20 |
| | Random | CENet | 76.04 | 91.44 | 95.35 | 84.12 | 79.57 | 68.19 | 83.09 | 72.75 | 33.82 |
| | Random | WaffleIron | 72.78 | 73.71 | 97.19 | 65.19 | 78.16 | 85.70 | 43.54 | 80.86 | 57.85 |
| | Random | Cylinder3D | 78.08 | 83.52 | 96.57 | 79.41 | 76.18 | 87.23 | 61.68 | 81.55 | 58.51 |
| | Random | SPVCNN$_{18}$ | 74.70 | 79.31 | 97.39 | 55.22 | 78.44 | 87.85 | 49.50 | 83.72 | 66.14 |
| | Random | SPVCNN$_{34}$ | 75.10 | 72.95 | 96.70 | 54.79 | **97.47** | **90.04** | 36.71 | **84.84** | **67.35** |
| | Random | MinkUNet$_{18}$ | 72.57 | 84.33 | 94.63 | 74.31 | 69.26 | 83.06 | 42.27 | 79.88 | 52.79 |
| | PPKT | MinkUNet$_{18}$ | 76.06 | 85.90 | 97.71 | 79.28 | 76.72 | 85.72 | 48.77 | 81.31 | 53.10 |
| | SLidR | MinkUNet$_{18}$ | 75.99 | 87.46 | 96.68 | 74.89 | 74.97 | 84.06 | 56.08 | 81.78 | 52.01 |
| | **Seal (Ours)** | MinkUNet$_{18}$ | **83.08** | **96.11** | **98.29** | 87.59 | 87.49 | 87.25 | 75.98 | 79.19 | 52.71 |

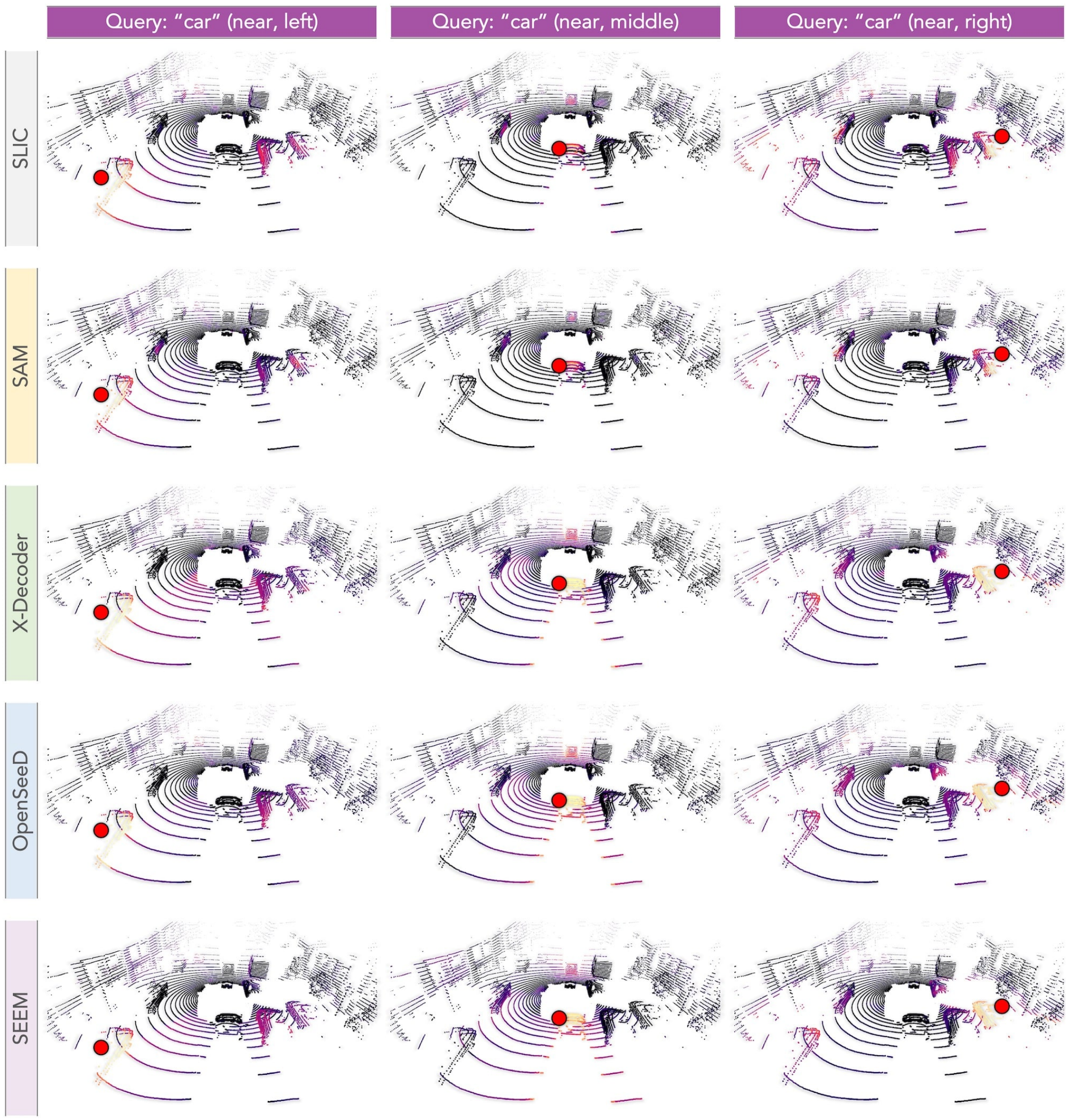

Figure F: The **cosine similarity** between the query point (denoted as the **red dot**) and the feature learned with SLIC [1] and different VFMs [50, 122, 111, 123]. The color goes from **violet** to **yellow** denoting **low** and **high** similarity scores, respectively. Best viewed in color.

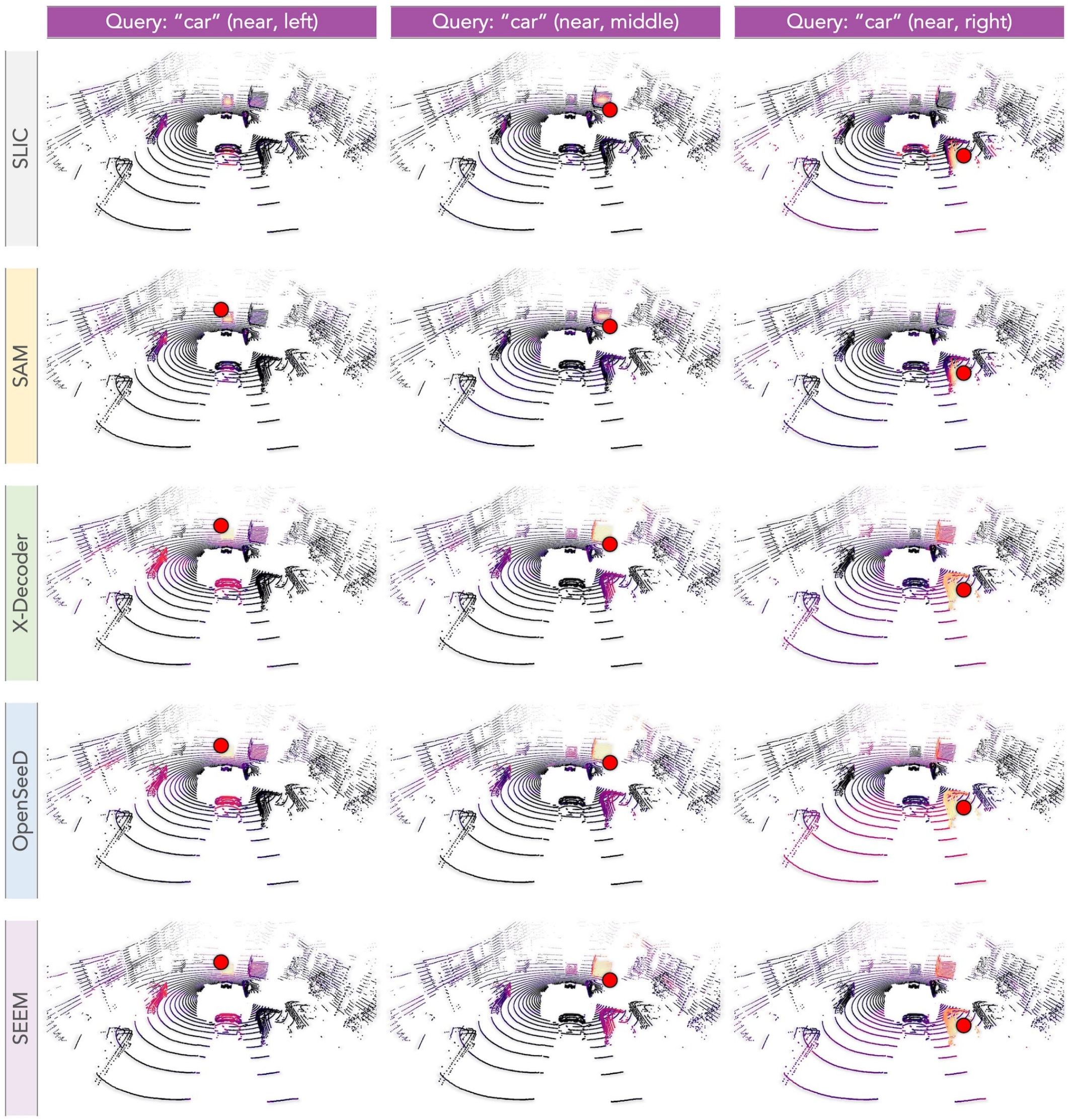

Figure G: The **cosine similarity** between the query point (denoted as the **red dot**) and the feature learned with SLIC [1] and different VFMs [50, 122, 111, 123]. The color goes from **violet** to **yellow** denoting **low** and **high** similarity scores, respectively. Best viewed in color.

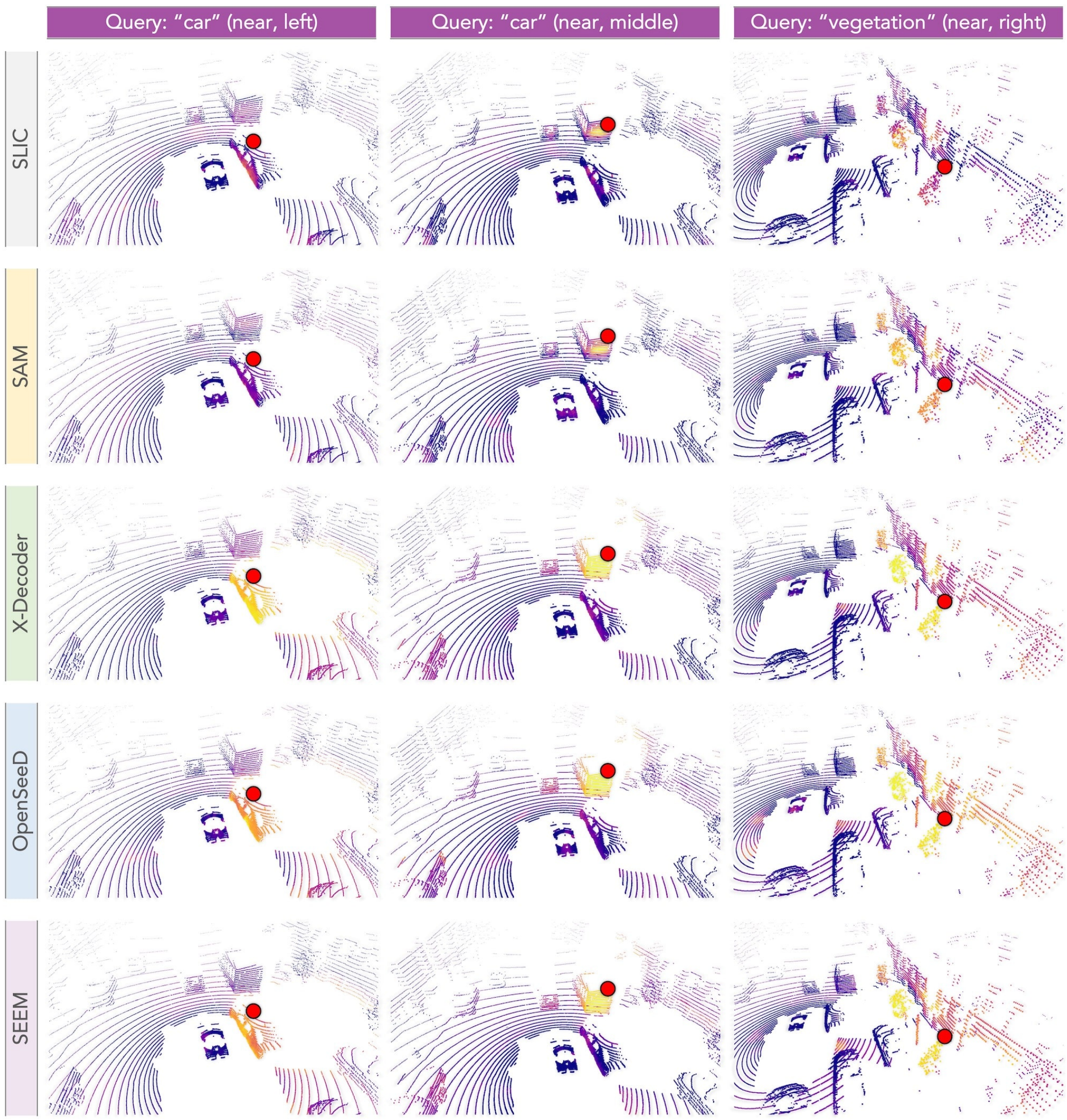

Figure H: The **cosine similarity** between the query point (denoted as the **red dot**) and the feature learned with SLIC [1] and different VFMs [50, 122, 111, 123]. The color goes from **violet** to **yellow** denoting **low** and **high** similarity scores, respectively. Best viewed in color.

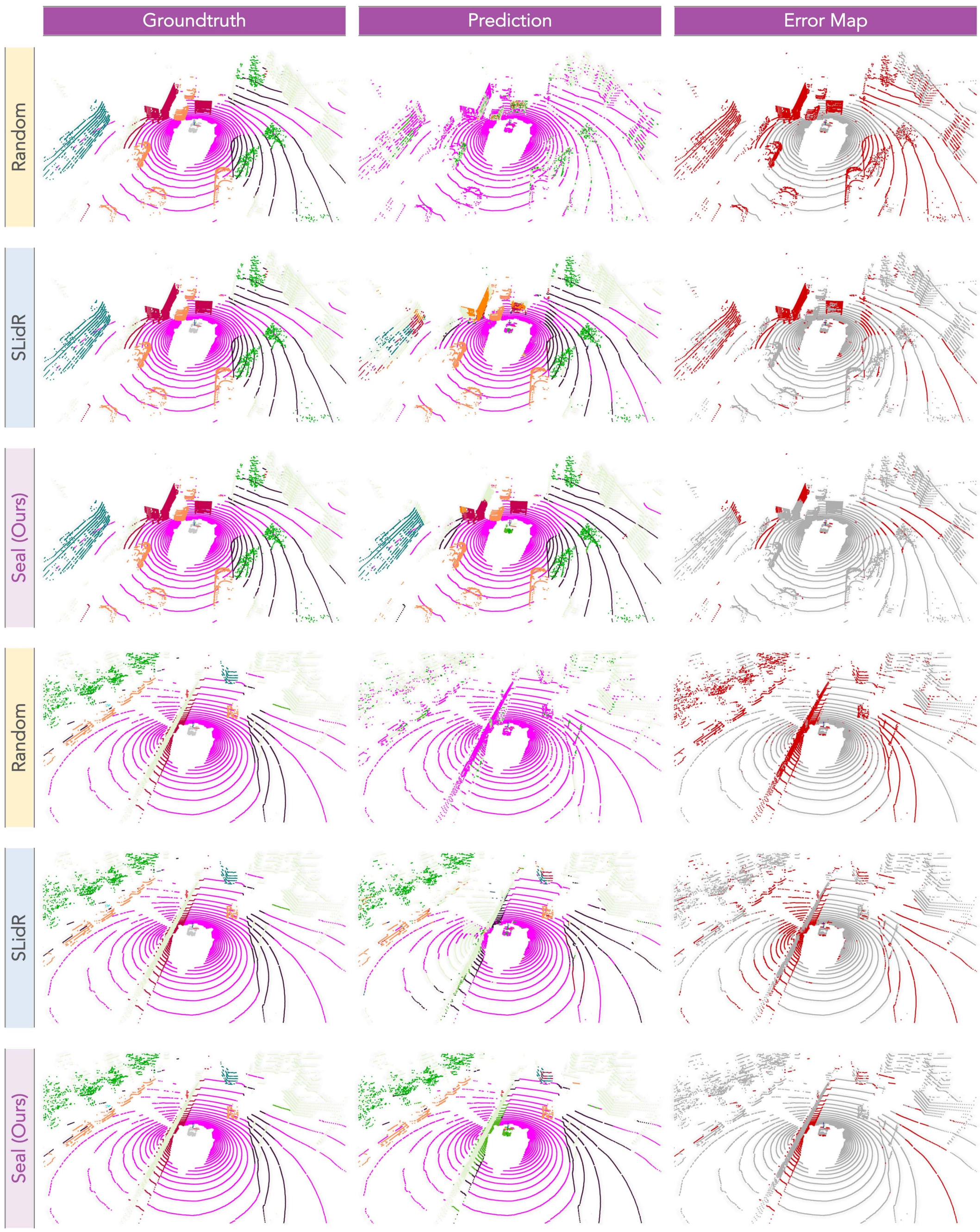

Figure I: The **qualitative results** of different point cloud pretraining approaches pretrained on the raw data of *nuScenes* [26] and fine-tuned with $1\%$ labeled data. To highlight the differences, the **correct** / **incorrect** predictions are painted in **gray** / **red**, respectively. Best viewed in color.

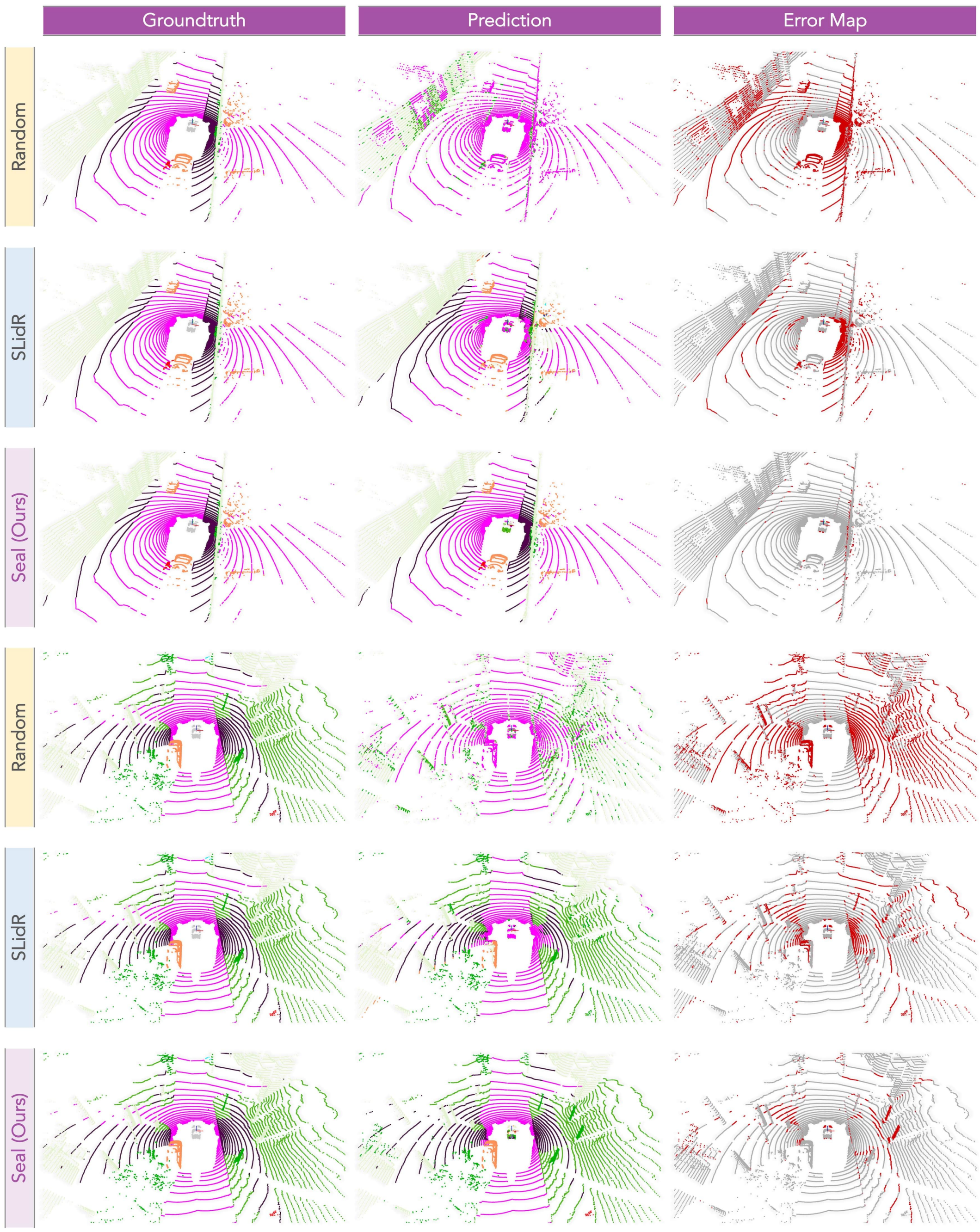

Figure J: The **qualitative results** of different point cloud pretraining approaches pretrained on the raw data of *nuScenes* [26] and fine-tuned with 1% labeled data. To highlight the differences, the **correct** / **incorrect** predictions are painted in **gray** / **red**, respectively. Best viewed in color.

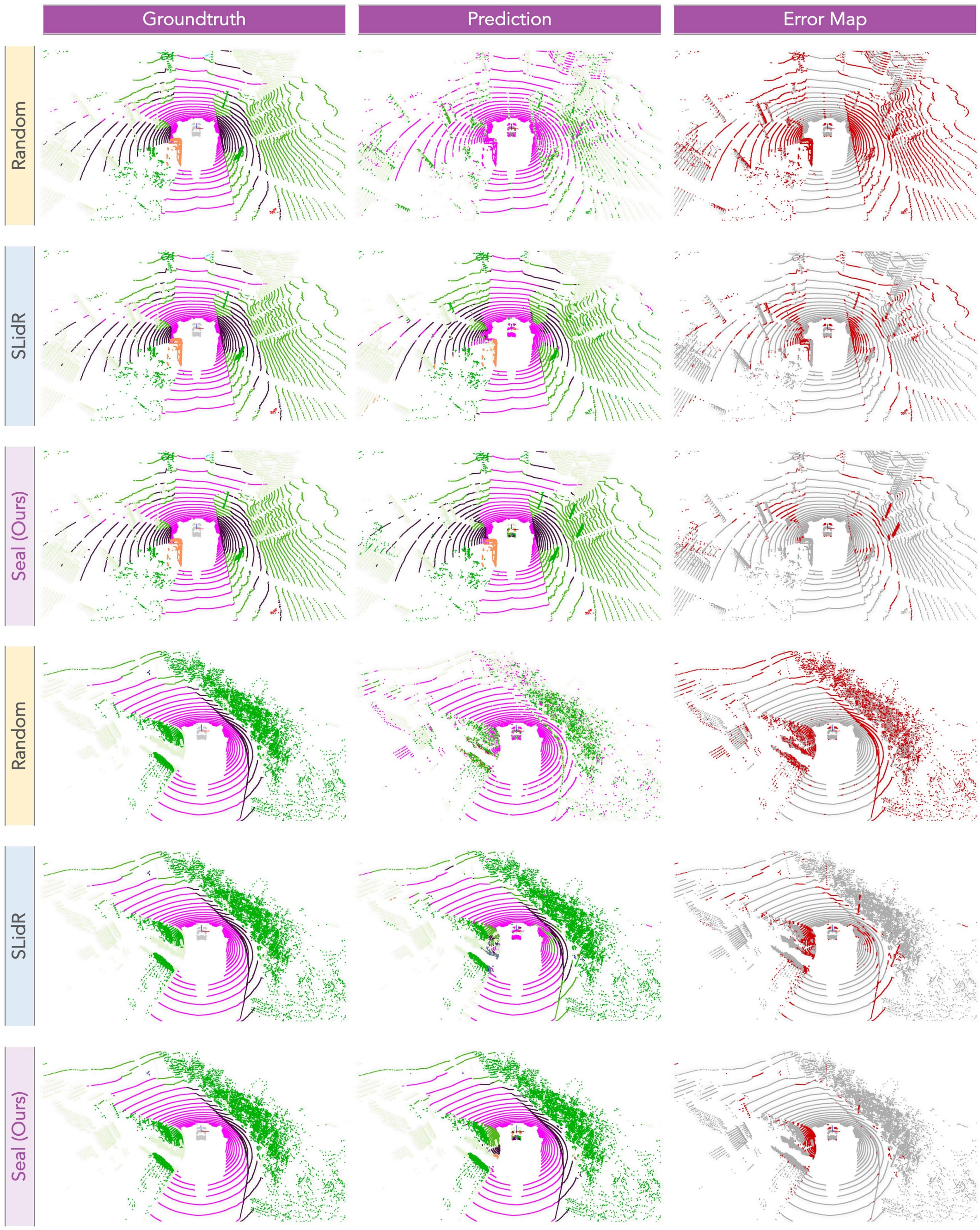

Figure K: The **qualitative results** of different point cloud pretraining approaches pretrained on the raw data of *nuScenes* [26] and fine-tuned with $1\%$ labeled data. To highlight the differences, the **correct** / **incorrect** predictions are painted in **gray** / **red**, respectively. Best viewed in color.

## D Public Resources Used

We acknowledge the use of the following public resources, during the course of this work:

- nuScenes[5] . . . CC BY-NC-SA 4.0
- nuScenes-devkit[6] . . . Apache License 2.0
- SemanticKITTI[7] . . . CC BY-NC-SA 4.0
- SemanticKITTI-API[8] . . . MIT License
- Waymo Open Dataset[9] . . . Waymo Dataset License
- ScribbleKITTI[10] . . . Unknown
- RELLIS-3D[11] . . . CC BY-NC-SA 3.0
- SemanticPOSS[12] . . . Unknown
- SemanticSTF[13] . . . CC BY-NC-SA 4.0
- SynLiDAR[14] . . . MIT License
- Synth4D[15] . . . GNU General Public License 3.0
- DAPS-3D[16] . . . MIT License
- nuScenes-C[17] . . . CC BY-NC-SA 4.0
- MinkowskiEngine[18] . . . MIT License
- SLidR[19] . . . Apache License 2.0
- spvnas[20] . . . MIT License
- Cylinder3D[21] . . . Apache License 2.0
- LaserMix[22] . . . CC BY-NC-SA 4.0
- mean-teacher[23] . . . Attribution-NonCommercial 4.0 International
- PyTorch-Lightning[24] . . . Apache License 2.0
- mmdetection3d[25] . . . Apache License 2.0

[5] https://www.nuscenes.org/nuscenes.
[6] https://github.com/nutonomy/nuscenes-devkit.
[7] http://semantic-kitti.org.
[8] https://github.com/PRBonn/semantic-kitti-api.
[9] https://waymo.com/open.
[10] https://github.com/ouenal/scribblekitti.
[11] http://www.unmannedlab.org/research/RELLIS-3D.
[12] http://www.poss.pku.edu.cn/semanticposs.html.
[13] https://github.com/xiaoaoran/SemanticSTF.
[14] https://github.com/xiaoaoran/SynLiDAR.
[15] https://github.com/saltoricristiano/gipso-sfouda.
[16] https://github.com/subake/DAPS3D.
[17] https://github.com/ldkong1205/Robo3D.
[18] https://github.com/NVIDIA/MinkowskiEngine.
[19] https://github.com/valeoai/SLidR.
[20] https://github.com/mit-han-lab/spvnas.
[21] https://github.com/xinge008/Cylinder3D.
[22] https://github.com/ldkong1205/LaserMix.
[23] https://github.com/CuriousAI/mean-teacher.
[24] https://github.com/Lightning-AI/lightning.
[25] https://github.com/open-mmlab/mmdetection3d.