# OpenReview forum: "Segment Any Point Cloud Sequences by Distilling Vision Foundation Models"
_NeurIPS.cc/2023/Conference — NeurIPS 2023 spotlight_

### Official Review · Reviewer_7qT5 · 2023-07-03

**Soundness:** 3 good
**Presentation:** 3 good
**Contribution:** 2 fair
**Rating:** 6
**Confidence:** 5

**Summary:**

This study represents the pioneering effort to utilize 2D vision foundation models for self-supervised representation learning on large-scale 3D point clouds. They introduce Seal, an innovative framework specifically designed to extract informative features from sequences of automotive point clouds. With its emphasis on scalability, consistency, and generalizability, Seal effectively captures semantic-aware spatial and temporal consistency, enabling the generation of highly valuable insights. Finally, they clearly demonstrate their superiority over previous state-of-the-art (SoTA) methods in both linear probing and fine-tuning for various downstream tasks.

**Strengths:**

1. The paper is well-written and easily comprehensible.
2. To my knowledge, this is the first attempt to utilize large-scale vision models for aiding 3D point cloud segmentation. The authors have also put in tremendous effort, completing this work within just a short span of one to two months.
3. Conducted extensive experiments and achieved satisfactory results on various segmentation datasets.

**Weaknesses:**

1. Strictly speaking, the approach in this paper is not considered unsupervised pretraining as it utilizes large models that rely on additional data. Most methods in Table 1, however, are unsupervised pretraining. Nevertheless, this distinction is not crucial, as the primary concern is to improve model performance, which is achieved through effective methods.
2. I believe that the approach presented in the paper has not fully distilled the knowledge of large 2D models. When using the complete dataset for both pretraining and fine-tuning, the information gain from distilling large models is not significant. Why do I say this? In the method employed in this paper, the large models primarily provide semantic supervision signals. If full semantic labels are given during fine-tuning, the information gain is limited. Although Table 2 shows significant improvements for some full dataset cases, this is mainly due to the use of semantic features in distillation rather than simple semantic pseudo-labels. The advantage of distilling large 2D models is that I can use more image-lidar pairs during pretraining. I believe that if additional image-lidar pairs are used, such as on the nuscenes dataset, there will be further improvements. I also hope to see similar experiments, such as pretraining with ten times the number of image-lidar pairs compared to nuscenes.
3. The method proposed in this paper is expected to be very general, so it should yield better results for other 3D perception tasks (e.g., 3D object detection).

**Questions:**

The questions are all in the weaknesses section.

**Limitations:**

Limitations are all in the weaknesses section.

---

> ### Author Rebuttal · Authors · 2023-08-09
>
> We thank Reviewer 7qT5 for devoting time to this review and providing valuable comments.
>
> ---
> > ***Q1:** "The approach in this paper is not considered unsupervised pretraining as it utilizes large models that rely on additional data."*
>
> **A:** Thanks for the comment. We agree with the reviewer that this approach is not strictly unsupervised, since the vision foundation models (VFMs) used are trained with image annotations. We also believe that this distinction is not crucial as the use of VFMs is becoming prevailing and can be considered as, to a certain extent, *"off-the-shelf"*. To avoid any potential misunderstanding, we will revise the tone of the elaboration and try to avoid using *"unsupervised"* in the revised manuscript.
>
> ---
> > ***Q2:** "I believe that if additional image-lidar pairs are used, there will be further improvements. I hope to see similar experiments, such as pretraining with ten times the number of image-lidar pairs compared to nuScenes."*
>
> **A:** Thanks for asking this insightful question. Our response to this question is as follows:
> - We agree with the reviewer that if sufficient semantic labels are given during the fine-tuning, the gain from model pretraining would become limited. In fact, this is also the case for many 2D pretraining tasks. Nevertheless, as verified in the experiments, the use of pretraining still exhibits advantages over random initialization, such as higher final performance, faster convergence rate, and better out-of-training-distribution robustness.
> - The improvements over some full dataset cases in Table 2 are mainly credited to the use of the large-scale pretraining dataset, i.e., nuScenes, compared to these small-scale downstream datasets (SemanticPOSS, SemanticSTF, and DAPS-3D).
> - An experiment on pretraining with much more image-LiDAR pairs is indeed desirable. Since nuScenes is already one of the largest LiDAR datasets, we seek the following two resources to consolidate the above observation:
>   - The *sweeps* data of nuScenes, which contains unannotated image-LiDAR pairs around nine times larger (around a total of 260k) than the used keyframes.
>   - The *Waymo Open* dataset, consisting of around a total of 150k image-LiDAR pairs from 64-beam LiDAR and five camera sensors. The point clouds are denser than that of nuScenes.
> - Due to the time limit, we are only able to include these results in the revised manuscript. We will update our progress (if any) during the Author-Reviewer discussion section.
>
> ---
> > ***Q3:** "The method proposed in this paper is expected to be very general, so it should yield better results for other 3D perception tasks (e.g., 3D object detection)."*
>
> **A:** Thanks for your suggestion. Our response to this question is as follows:
> - We agree with the reviewer that the pretrained weights should be general for related 3D perception tasks as well. We consolidate this statement by extending Seal (and existing LiDAR representation approaches [R1,R2]) to 3D panoptic segmentation -- a task that requires predicting both semantics and instance identities of LiDAR points.
> - The results of different pretraining approaches pretrained on nuScenes (the same configuration as LiDAR semantic segmentation in the paper) and fine-tuned on the Panoptic nuScenes dataset [R3] under 5%, 10%, and 20% splits are as follows:
> |Method|PQ (5%)|mIoU (5%)|PQ (10%)|mIoU (10%)|PQ (20%)|mIoU (20%)|
> |-|:-:|:-:|:-:|:-:|:-:|:-:
> |Random|35.60|45.56|44.10|54.80|51.29|62.29
> |PPKT|47.73|53.16|53.63|58.95|56.25|64.42
> |SLidR|44.68|52.92|51.44|60.00|55.64|65.67
> |Seal|49.20|55.33|55.01|62.74|57.73|66.94
> - We use the pretrained MinkUNet as the semantic extractor in the 3D panoptic segmentation pipeline. The instance extractor consists of an instance head, followed by a clustering step. This step leverages the semantic predictions to filter out *stuff* points, retaining only those belonging to *thing* instances, such as ‘pedestrian’, ‘car’, and ‘bicyclist’. Subsequently, the remaining points are subjected to clustering to identify the instances based on the features from the instance head. Similar to [R4], the mean shift clustering is employed for this purpose, where the bandwidth is set to 2.5, and the minimum number of points per cluster is set to 20.
> - Following the conventional reporting of PQ (panoptic quality) and mIoU, we observe from the above table that Seal also shows superiority over existing LiDAR pretraining approaches in the 3D panoptic segmentation task. The results verify that the pretrained weights can be general across different 3D perception tasks.
> - To further solidify this claim, we seek another extension to the 3D object detection task. Due to the time limit, we will include the results in the revised manuscript.
>
> ---
> **References:**
> - [R1] Y.-C. Liu, et al. "Learning from 2D: Contrastive Pixel-to-Point Knowledge Transfer for 3D Pretraining," arXiv, 2021.
> - [R2] C. Sautier, et al. "Image-to-LiDAR Self-Supervised Distillation for Autonomous Driving Data," CVPR, 2022.
> - [R3] W. K. Fong, et al. "Panoptic nuScenes: A Large-Scale Benchmark for LiDAR Panoptic Segmentation and Tracking," RA-L, 2022.
> - [R4] F. Hong, et al. "LiDAR-based Panoptic Segmentation via Dynamic Shifting Network," CVPR, 2021.

---

> > ### Author Response · Authors · 2023-08-18
> > **Looking forward to discussion**
> >
> > Dear Reviewer 7qT5,
> >
> > We sincerely thank you for devoting time to this review and providing valuable comments.
> >
> > ---
> > Based on the reviewers' comments, we have revised our manuscript to include the following changes:
> >
> > - We have supplemented more details on the definition and generation of semantic superpixel and superpoint.
> > - We have added some remarks on the choice and motivation behind each design.
> > - We conducted a study on the effect of possible misalignment between the LiDAR and camera sensors.
> > - We have extended the Seal framework to other 3D perception tasks.
> > - We have polished the elaboration and clarified some typos and misunderstandings in the main submission.
> >
> > ---
> > We will actively participate in the Author-Reviewer discussion session. Please don't hesitate to let us know of any additional comments on the manuscript or the changes.
> >
> > Best regards,
> >
> > The Authors

---

> > > ### Author Response · Authors · 2023-08-19
> > > **Authors' Response to Reviewer 7qT5**
> > >
> > > We thank Reviewer 7qT5 for devoting time to this review and providing valuable comments.
> > >
> > > ---
> > > Regarding Q2 (copied below) in the previous comment:
> > > > *"I believe that if additional image-lidar pairs are used, there will be further improvements. I hope to see similar experiments, such as pretraining with ten times the number of image-lidar pairs compared to nuScenes."*
> > >
> > > **A:** Thanks for the suggestion. We now present the follow-up updates for this question.
> > > - We believe pretraining with much more image-LiDAR pairs is indeed desirable. We use the *sweeps data* of nuScenes for this kind of pretraining, which contains unannotated image-LiDAR pairs around nine times larger (a total of ~ 260k frames) than the used keyframe data (~ 29k frames).
> > > - We use the baseline SLidR for pretraining in this update and will include more results for Seal in the revision. The pretraining results on nuScenes are shown in the following table:
> > > | Method | LP | 1% | 5% | 10% | 25% | Full |
> > > |:-|:-:|:-:|:-:|:-:|:-:|:-:|
> > > | Random | 8.10 | 30.30 | 47.84 | 56.15 | 65.48 | 74.66 |
> > > | PointContrast | 21.90 | 32.50 | - | - | - | - |
> > > | DepthContrast | 22.10 | 31.70 | - | - | - | - |
> > > | PPKT | 35.90 | 37.80 | 53.74 | 60.25 | 67.14 | 74.52 |
> > > | SLidR | 38.80 | 38.30 | 52.49 | 59.84 | 66.91 | 74.79 |
> > > | **SLidR (w/ sweeps)** | 39.57 | 39.40 | 53.21 | 60.69 | 67.44 | 75.09 |
> > > | ST-SLidR | 40.48 | 40.75 | 54.69 | 60.75 | 67.70 | 75.14 |
> > > | Seal | 44.95 | 45.84 | 55.64 | 62.97 | 68.41 | 75.60 |
> > >
> > > - We observe that using more image-LiDAR pairs during pretraining indeed improves the performance, for both linear probing and downstream fine-tuning tasks with different ratios.
> > > - The improvements are not linearly scaled up with the amount of data used for pretraining. We conjecture that this is because the *sweeps data* of nuScenes are less diverse, since they were collected sequentially.
> > > - Nevertheless, we believe that pretraining with more image-LiDAR pairs is promising and desirable; with the advent of datasets with larger scales, the proposed pretraining framework could become more powerful and can achieve better cross-dataset performance.
> > >
> > > ---
> > > Last but not least, we thank Reviewer 7qT5 again for the time and effort devoted to this review.

---

### Official Review · Reviewer_1xK9 · 2023-07-06

**Soundness:** 3 good
**Presentation:** 3 good
**Contribution:** 3 good
**Rating:** 7
**Confidence:** 4

**Summary:**

This paper proposed a novel framework named Seal. The Seal distills VFMs into point clouds, enabling achieves efficient segmentation of various automotive point cloud sequences without requiring extensive annotation during the pre-training stage. It exhibits excellent performance across multiple datasets.

**Strengths:**

The Seal distills the feature extraction capability of visual VFMs to point clouds. This paper promotes cross-modal representation learning through the proposed spatiotemporal consistency compared to previous works. The results obtained by The Seal on 11 different point cloud datasets demonstrate its effectiveness and superiority, highlighting its significant potential for 3D feature learning.

**Weaknesses:**

The innovative contributions of the paper can be summarized into two points compared to previous works: 1. Distilling the representation learning capability from VFMs into point cloud data processing. 2. Introducing semantic superpoint temporal consistency to promote cross-modal learning. However, it is noted that the level of innovation in the paper might not be particularly strong.

**Questions:**

1. I have some questions regarding the semantic superpoint temporal consistency. Firstly, according to Figure 4, the segmentation results provided by VFMs are at the instance level, while this paper aims to obtain a pretraining model for semantic segmentation tasks. According to Equation 2, this paper treats instances of the same object as positive samples and the rest as negative samples. Therefore, there is a high possibility that negative samples may include instances of the same class. This could potentially hurt the segmentation task. On the other hand, how is the correspondence between moving objects in point cloud data at different time steps obtained?
2. The paper mentions the ability to achieve good performance even in cases where camera and LiDAR calibration information is unavailable (on page 6 line 199). However, without approximate intrinsic and extrinsic parameters, establishing the 2D-3D spatial correspondence would be challenging. Therefore, I am curious about the performance in such scenarios. However, it appears that the paper does not provide specific experiments or results for these particular cases.

**Limitations:**

The statement in the limitation section that the assumption of obtaining calibrated and synchronized data between LiDAR and cameras is overly idealistic contradicts the description in the main text.

---

> ### Author Rebuttal · Authors · 2023-08-09
>
> We thank Reviewer 1xK9 for devoting time to this review and providing valuable comments.
>
> ---
> > ***Q1:** "(i) The results provided by VFMs are at the instance level, while this paper aims to obtain a pretraining model for semantic segmentation tasks. (ii) The negative samples may include instances of the same class, which could potentially hurt the segmentation task. (ii) how is the correspondence between moving objects in point cloud data at different time steps obtained?"*
>
> **A:** Thank you for the comment.
> - For question (i): The reason for performing instance-level temporal consistency has two aspects:
>   - Unlike 2D scenarios with stable shape and texture cues, 3D objects and backgrounds have spatial coherence and complex point arrangements. Treating each point separately and applying point-level temporal consistency regularization could result in fragmented or noisy representations that don't capture the object's inherent structure.
>   - The class distribution of LiDAR scenes is severely long-tailed, with a substantial portion of points belonging to static classes like 'driveable surface,' 'manmade,' and 'vegetation.' For instance, in the nuScenes training set, approximately ~90.6% of points correspond to *static* classes, leaving only around ~9.4% for *dynamic* (instance) classes. Adequate representations for static classes can be acquired through fine-tuning.
> - For question (ii): We agree that it is possible to include instances of the same class into negative samples.  In fact, it could be a common problem (i.e., negative samples), but our method is more reliable than the prior works. For one thing, the semantic superpixels generated by VFMs are semantically richer than the conventional method and also provide more complete instances. This reduces, to a certain extent, the possibility of "self-conflict". Besides, our method utilizes much fewer segments (\~30) in a frame than the prior work (\~300). We verify through experiments that our approach outperforms the prior art.
> - For question (iii): We use the following steps to associate the moving objects in different timestamps:
>   - *Coordinate transformation*. The point cloud $P^{t+n}$​ at timestamp $t+n$ is first transformed into the coordinate system of $P^t$ at $t$. This alignment is established using the sensor extrinsic and intrinsic matrices. The transformed $P^{t+n}$ is concatenated with $P^t$, creating a composite point cloud $P$.
>   - *RANSAC segmentation*. The composite point cloud $P$ is divided into ground and non-ground point groups using the RANSAC algorithm, enabling differentiation between static and moving points.
>   - *HDBSCAN clustering.* The non-ground points are then used for clustering. This provides distinct segments representing different instances in the composite point cloud $P$.
>   - *Instance labeling.* Each segment is assigned a unique instance ID, where the ground plane points are labeled as background.
>   - Consequently, we separate $P$ into $P^{t+n}$ and $P^t$ via reverse coordinate transformation. The correspondence of the same moving object across different timestamps can thus be extracted using the instance IDs.
>
> ---
> > ***Q2:** "I am curious about the performance in cases where camera and LiDAR calibration information is unavailable."*
>
> **A:** Thanks for the comment.
> - An accurate calibration could be important for establishing correct correspondences between LiDAR and cameras. In most cases, those sensors should be well-calibrated for an autonomous car. It is unusual if the calibration is completely unknown, but it is possible to be imprecise due to a lack of maintenance. Hence, we conduct the following experiments to validate the robustness of our method.
>
> -  For each point coordinate $(x_i, y_i, z_i)$ in a LiDAR point cloud, the corresponding pixel $(u_i, v_i)$ can be found by:
> $$[u_i, v_i, 1]^T = \frac{1}{z_i}  \cdot \Gamma_K\cdot  \Gamma \cdot [x_i, y_i, z_i, 1]^T,$$
> where $\Gamma \in \mathbb{R}^{4\times4}$ is the camera extrinsic matrix that consists of a rotation matrix and a translation matrix, and $\Gamma_K\in \mathbb{R}^{3\times4}$ is the camera intrinsic matrix. To simulate the misalignment between LiDAR and cameras, we add random noises to the camera extrinsic matrix $\Gamma$.
>
> - **[Table 1]** Add $\pm$1% random noises to $\Gamma$:
> |Method|1%|5%|10%|25%
> |-|-|-|-|-
> |Random|30.30|47.84|56.15|65.48
> |PPKT|34.94|51.11|58.54|65.01
> |SLidR|37.92|53.08|59.89|66.90
> |Seal|45.23|55.71|62.62|68.13
> - **[Table 2]** Add $\pm$5% random noises to $\Gamma$:
> |Method|1%|5%|10%|25%
> |-|-|-|-|-
> |Random|30.30|47.84|56.15|65.48
> |PPKT|33.69|51.40|58.00|64.11
> |SLidR|38.00|52.36|60.01|64.10
> |Seal|45.66|55.42|62.77|68.01
> - **[Table 3]** Add $\pm$10% random noises to $\Gamma$:
> |Method|1%|5%|10%|25%
> |-|-|-|-|-
> |Random|30.30|47.84|56.15|65.48
> |PPKT|33.35|50.98|57.84|63.52
> |SLidR|37.30|51.11|58.50|64.50
> |Seal|44.80|54.45|61.80|68.29
> - **[Table 4]** Performance w/o calibration noises (from Table 1 in the paper):
> |Method|1%|5%|10%|25%
> |-|-|-|-|-
> |Random|30.30|47.84|56.15|65.48
> |PPKT|37.80|53.74|60.25|67.14
> |SLidR|38.30|52.49|59.84|66.91
> |Seal|45.84|55.64|62.97|68.41
> - From the above results we observe:
>   - The possible calibration errors in-between LiDAR and camera sensors will cause performance degradation for different pretraining approaches, i.e., Tables 1 to 3 compared to that of Table 4.
>   - The performance degradation for PPKT is especially prominent; we conjecture that this is because the point-wise consistency regularization of PPKT relies heavily on the calibration accuracy and encounters problems under misalignment.
>   - Both SLidR and Seal exhibit certain robustness; we believe the superpixel-level consistency is less sensitive to calibration perturbations. It is worth mentioning that Seal can maintain good performance under calibration error, since i) our VFM-assisted representation learning tends to be more robust; ii) we enforce also superpoint temporal consistency which not relies on the 2D-3D correspondence.

---

> > ### Author Response · Authors · 2023-08-18
> > **Looking forward to discussion**
> >
> > Dear Reviewer 1xK9,
> >
> > We sincerely thank you for devoting time to this review and providing valuable comments.
> >
> > ---
> > Based on the reviewers' comments, we have revised our manuscript to include the following changes:
> > - We have supplemented more details on the definition and generation of semantic superpixel and superpoint.
> > - We have added some remarks on the choice and motivation behind each design.
> > - We conducted a study on the effect of possible misalignment between the LiDAR and camera sensors.
> > - We have extended the Seal framework to other 3D perception tasks.
> > - We have polished the elaboration and clarified some typos and misunderstandings in the main submission.
> >
> > ---
> > We will actively participate in the Author-Reviewer discussion session. Please don't hesitate to let us know of any additional comments on the manuscript or the changes.
> >
> > Best regards,
> >
> > The Authors

---

### Official Review · Reviewer_3rsk · 2023-07-06

**Soundness:** 4 excellent
**Presentation:** 4 excellent
**Contribution:** 4 excellent
**Rating:** 8
**Confidence:** 4

**Summary:**

This paper introduces a novel framework, called *Seal*, that leverages VFMs for self-supervised representation learning on automotive point cloud sequences. The main idea is to leverage the 2D-3D correspondence between LiDAR and camera sensors and construct high-quality contrastive samples for cross-modal representation learning. The paper also proposes a superpoint temporal consistency regularization to enforce geometric stability across different timestamps. The paper evaluates the proposed framework on 11 different point cloud datasets and shows that it outperforms previous state-of-the-art methods in both linear probing and few-shot fine-tuning settings.

**Strengths:**

**Originality:**
 - the **first** work utilizing 2D VFMs for self-supervised representation learning on large-scale 3D point clouds
 - a novel VFM-assisted contrastive learning objective that transfers the knowledge from the pretrained 2D to the 3D network at the semantic superpixel level
 - a superpoint temporal consistency regularization

**Quality:**
 - well-written and organized, with clear motivation, problem formulation, related work, methodology, experiments, conclusion and discussion & limitations
 - extensive experiments on 11 different point cloud datasets and sufficient ablation studies
 - sufficient mathematical formulations and derivations to support the proposed methods
 - well-written and sufficient Supplementary Material with demo visualizations

**Clarity:**
 - clear and easy to follow
 - uses consistent notations and symbols throughout the text and equations
 - helpful figures and tables to illustrate the main ideas and results

**Significance:**
 - significant for both research and practice in 3D perception and representation learning
 - addresses an important problem of segmenting diverse automotive point clouds in a scalable, consistent, and generalizable manner

**Weaknesses:**

The datasets conducted on are all **outdoor** point cloud dataset for autonomous driving. Unless the authors conduct more experiments on **indoor** point cloud datasets, they'd better just state that "Segment Any **Automotive** Point Cloud Sequences ..." in their title.

Minior Issue:
- Undefined variable $L^{tmp}$ in Equ. (3)

**Questions:**

In Table 3 (Full), it seems *Seal* performs not well in `Cross`, `Echo` and `Sensor` (especially `Sensor`, 39.85 vs 49.21). Could you please provide some possible reasons about that?

**Limitations:**

Authors should be rewarded that the limitations are explicitly mentioned in *Discussion & Limitation*.

---

> ### Author Rebuttal · Authors · 2023-08-09
>
> We thank Reviewer 3rsk for devoting time to this review and providing valuable comments.
>
> ---
> > ***Q1:** "Unless the authors conduct more experiments on indoor point cloud datasets, they'd better just state that 'Segment Any Automotive Point Cloud Sequences …' in their title."*
>
> **A:** Thanks for your suggestion. We will seek a more proper title description in the revision. In this work, we mainly targeted segmenting point cloud *sequences*, where such sequences are often in the form of *automotive point clouds*. Nevertheless, the proposed spatial and temporal consistency regularization objectives tend not constrained by current use cases. We seek further extension of this framework to other point cloud datasets as well, such as those from indoor scenes.
>
> ---
> > ***Q2:** "Undefined variable $L^{tmp}$  in Equ. (3)."*
>
> **A:** We thank the reviewer for pointing this typo out. We have addressed this issue in the revised paper. Here the symbol $\mathcal{L}^{tmp}$ means to denote the temporal consistency loss, which is used to encourage consistency between the mean representation of a segment across different timestamps.
>
> ---
> > ***Q3:** "In Table 3, it seems Seal performs not well in Cross, Echo, and Sensor. Can you provide some possible reasons for that?"*
>
> **A:** Thanks for your question. We would like to highlight that the performance discrepancy of Seal under the *'Crosstalk'*, *'Incomplete Echo'*, and *'Cross-Sensor'* scenarios in Table 3 is mainly attributed to the use of different LiDAR segmentation backbones.
> - When comparing different pretraining approaches with *the same LiDAR segmentation backbone*, i.e., MinkUNet for the last four rows in Table 3, Seal exhibits superiority over all corruption types except *‘Incomplete Echo’*, which is slightly lower than that of SLidR (59.87 vs. 61.16).
> - When comparing among backbones of different modalities, e.g., range view (CENet [R2]), raw points (WaffleIron [R3]), and point-voxel fusion (SPVCNN [R4]), the robustness against different corruption types becomes different. As been validated in Robo3D [R1], different LiDAR modalities often show diverse robustness effects. on nuScenes, the sparse voxel-based backbones like MinkUNet show sub-par performance under point cloud density changes, such as *'Crosstalk'* (extra noisy points within the mid-range areas in between two or multiple LiDAR sensors), *'Incomplete Echo'* (point miss detection for instances with dark colors), and *'Cross-Sensor'* (point clouds captured by sensors of different configurations).
> - To further solidify this observation, we will explore the robustness of Seal and other pretraining approaches (e.g. PPKT and SLidR) using LiDAR segmentation backbones from other modalities. Due to the time limit, we are only able to supplement this result in the updated paper.
>
> ---
> **References:**
> - [R1] L. Kong, et al. “Robo3D: Towards Robust and Reliable 3D Perception against Corruptions.” arXiv, 2023.
> - [R2] H.-X. Cheng, et al. “CENet: Toward Concise and Efficient LiDAR Semantic Segmentation for Autonomous Driving.” ICME, 2022.
> - [R3] G. Puy, A. Boulch, and R. Marlet. “Using A Waffle Iron for Automotive Point Cloud Semantic Segmentation.” arXiv, 2023.
> - [R4] H. Tang, et al. “Searching Efficient 3D Architectures with Sparse Point-Voxel Convolution.” ECCV, 2020.

---

> > ### Author Response · Authors · 2023-08-18
> > **Looking forward to discussion**
> >
> > Dear Reviewer 3rsk,
> >
> > We sincerely thank you for devoting time to this review and providing valuable comments.
> >
> > ---
> > Based on the reviewers' comments, we have revised our manuscript to include the following changes:
> > - We have supplemented more details on the definition and generation of semantic superpixel and superpoint.
> > - We have added some remarks on the choice and motivation behind each design.
> > - We conducted a study on the effect of possible misalignment between the LiDAR and camera sensors.
> > - We have extended the Seal framework to other 3D perception tasks.
> > - We have polished the elaboration and clarified some typos and misunderstandings in the main submission.
> >
> > ---
> > We will actively participate in the Author-Reviewer discussion session. Please don't hesitate to let us know of any additional comments on the manuscript or the changes.
> >
> > Best regards,
> >
> > The Authors

---

> > > ### Comment · Reviewer_3rsk · 2023-08-18
> > >
> > > Thanks for the rebuttal and revised manuscript. The authors have addressed all my concerns and questions.

---

> > > > ### Author Response · Authors · 2023-08-19
> > > > **Authors' Response to Reviewer 3rsk**
> > > >
> > > > We sincerely thank Reviewer 3rsk for the positive feedback provided and the time devoted to this review.
> > > >
> > > > Best regards,
> > > >
> > > > The Authors

---

### Official Review · Reviewer_EPuy · 2023-07-07

**Soundness:** 3 good
**Presentation:** 3 good
**Contribution:** 2 fair
**Rating:** 6
**Confidence:** 5

**Summary:**

The manuscript presents Seal, a framework that leverages VFMs (Visual Foundation Models) to segment diverse point cloud sequences in autonomous driving scenarios. The proposed approach employs VFMs to initially segment superpixels in 2D camera images and subsequently projects them to 3D superpoints. Then, it incorporates two specific pre-training schemes: spatial contrastive learning, which involves training cross-modally from pre-trained 2D features to 3D representations, and temporal consistency regularization, which ensures the consistency of 3D point features across different timestamps. Extensive experimentation validates the effectiveness and robustness of the proposed framework.

**Strengths:**

1. The manuscript exhibits clear and concise writing, making it easy to comprehend. The mathematical explanations and figures are presented with high clarity, facilitating understanding.
2. Through extensive experiments, the study demonstrates the effectiveness and robustness of the proposed method, establishing its reliability.
3. The main contribution of the manuscript is the development of the framework that can serve as a valuable resource for future reference and practical implementation.

**Weaknesses:**

Novelty: The novelty of the manuscript may be somewhat limited. While the authors claim that the main novelty lies in leveraging VFM models for superpixel set segmentation and introducing two specific pretraining schemes, it is worth noting that the use of VFM models for segmentation is already a widely discussed topic and is short of novelty. Additionally, the proposed pretraining schemes closely follow common cross-modal contrastive learning approaches. This may give the impression that the paper is riding on the popularity of VFM models without truly exploring their potential for feature-level distillation.

**Questions:**

1. What is the parameterization of a superpixel? If a superpixel is selected as a representative pixel from an area with the same class in VFM outputs, how is the pixel specifically chosen?
2. What is the motivation behind enforcing the convergence of 3D point features to a single mean representation? In the 2D scenario, it is common for features at different pixels of the same object to be distinct, as they capture diverse parts and information about the object. Shouldn't this principle apply to the 3D case?

**Limitations:**

Besides the limitations the authors discuss in the manuscript, I've seen no additional specific limitations.

---

> ### Author Rebuttal · Authors · 2023-08-09
>
> We thank Reviewer EPuy for devoting time to this review and providing valuable comments.
>
> ---
> > ***Q1:** "The use of VFM models for segmentation is widely discussed. The proposed pretraining schemes closely follow common cross-modal contrastive learning approaches. The paper is riding on the popularity of VFM models without truly exploring their potential for feature-level distillation."*
>
> **A:** We thank the reviewer for the comment. Our response to this question is as follows:
> - Using vision foundation models (VFMs) for segmentation is among one of the hottest research topics in the computer vision community. Several recent attempts have been made to leverage VFMs (e.g. SAM [R1]) for tackling different vision tasks, such as remote sensing [R2], open-set object detection [R3], tracking [R4], image tagging [R5], medical imaging [R6], and many others.
> - It is worth noting that, different from all the above pursuits, we present the first study on utilizing  VFMs for self-supervised representation learning on large-scale 3D point clouds. We believe this work could enlighten follow-up research on designing more robust and scalable LiDAR segmentation frameworks.
> - Seal adopts the camera-to-LiDAR contrastive objective [R7] during representation learning. The knowledge from VFMs is leveraged, in the form of semantic superpixels and superpoints, to help better enforce cross-modal consistency. We agree with the reviewer that exploring feature-level distillation is also a promising solution. However, there are several limitations to conducting such a distillation using existing VFMs:
>   - *Computational overhead*. Current VFMs are often constrained by large model sizes and are struggled with high latency. Conducting feature-level distillation would require significantly more computing resources compared to the original LiDAR segmentation task.
>   - *Domain gap and representational differences*. VFMs are primarily trained for specific tasks; the features learned might not be directly transferable to the LiDAR modality due to the inherent differences in sensor data and environmental characteristics.
>   - *Loss of modality-specific information*. When using VFMs' features for distillation, modality-specific information present in LiDAR data might be lost. VFMs' features are optimized for vision-related tasks and might not capture the intricate details and characteristics unique to LiDAR data.
>
> ---
> > ***Q2:** "What is the parameterization of a superpixel? If a superpixel is selected as a representative pixel from an area with the same class in VFM outputs, how is the pixel specifically chosen?"*
>
> **A:** Thanks for asking. In our framework, a superpixel denotes a group of semantically cohesive pixels segmented by VFMs, which is similar to the concept of a semantic mask. That is to say, a superpixel is not just one representative pixel from an area but a group of pixels that tend to belong to the same semantic class. Some examples of superpixels are shown in the first row of Figure 1, where each color represents a distinct superpixel generated by either SLIC or VFMs.
>
> ---
> > ***Q3:** "What is the motivation behind enforcing the convergence of 3D point features to a single mean representation? In the 2D scenario, it is common for features at different pixels of the same object to be distinct, as they capture diverse parts and information about the object. Shouldn't this principle apply to the 3D case?"*
>
> **A:** We thank the reviewer for asking this insightful question. Our response is as follows:
> - The motivation for enforcing the convergence of 3D point features to a single mean representation is rooted in the distinct nature of LiDAR data. Unlike 2D scenarios, where pixel features at different locations of the same object can encapsulate various object parts and details, the complexities of LiDAR point clouds pose unique challenges:
>   - Different from 2D scenarios which contain relatively stable shape and texture cues, 3D objects and backgrounds are characterized by their spatial coherence and intricate arrangements of points. Treating each individual point as an independent entity might lead to fragmented or noisy representations that fail to capture the object's underlying structure.
>   - Here we enforce the convergence of 3D point features to a single mean representation to synthesize a cohesive depiction of the objects and backgrounds, while mitigating noise and variability intrinsic to the raw LiDAR data. This could facilitate, to a certain extent, a more stable and informative 3D representation.
>   - We validate through experiments that this single mean representation exhibits better performance than the point-wise contrastive approaches, such as PointContrast [R8] and PPKT [R9], under the linear probing, few-shot fine-tuning, and robustness evaluations (i.e. Tables 1 to 3 in the main text).
> - What is more, the per-point contrastive learning objectives, which are more popular for indoor scenes captured by RGB-D cameras, are computationally more expensive than the mean representation for handling outdoor LiDAR scenes; the automotive point clouds are both larger in scale and richer in diversity than the indoor scenes.
>
> ---
> **References:**
> - [R1] Segment anything. ICCV, 2023.
> - [R2] RSPrompter: Learning to Prompt for Remote Sensing Instance Segmentation based on Visual Foundation Model. arXiv, 2023.
> - [R3] Grounding DINO: Marrying DINO with Grounded Pre-Training for Open-Set Object Detection. arXiv, 2023.
> - [R4] Segment and Track Anything. arXiv, 2023.
> - [R5] Recognize Anything: A Strong Image Tagging Model. arXiv, 2023.
> - [R6] Segment Anything in Medical Images. arXiv, 2023.
> - [R7] Image-to-LiDAR Self-Supervised Distillation for Autonomous Driving Data. CVPR, 2022.
> - [R8] PointContrast: Unsupervised Pre-training for 3D Point Cloud Understanding. ECCV, 2020.
> - [R9] Learning from 2D: Contrastive Pixel-to-Point Knowledge Transfer for 3D Pretraining. arXiv, 2021.

---

> > ### Author Response · Authors · 2023-08-18
> > **Looking forward to discussion**
> >
> > Dear Reviewer EPuy,
> >
> > We sincerely thank you for devoting time to this review and providing valuable comments.
> >
> > ---
> > Based on the reviewers' comments, we have revised our manuscript to include the following changes:
> > - We have supplemented more details on the definition and generation of semantic superpixel and superpoint.
> > - We have added some remarks on the choice and motivation behind each design.
> > - We conducted a study on the effect of possible misalignment between the LiDAR and camera sensors.
> > - We have extended the Seal framework to other 3D perception tasks.
> > - We have polished the elaboration and clarified some typos and misunderstandings in the main submission.
> >
> > ---
> > We will actively participate in the Author-Reviewer discussion session. Please don't hesitate to let us know of any additional comments on the manuscript or the changes.
> >
> > Best regards,
> >
> > The Authors

---

### Author Response · Authors · 2023-08-10
**General Response**

We sincerely thank all the reviewers for the time and effort devoted to this review.

---
We are glad to see that the reviewers are acknowledging this work:
- *"is significant for both research and practice in 3D perception and representation learning"* (Reviewer 3rsk);
- *"can serve as a valuable resource for future reference and practical implementation"* (Reviewer EPuy);
- *"demonstrates the effectiveness and superiority, highlighting its significant potential for 3D feature learning"* (Reviewer 1xK9);
- *"conducted extensive experiments and achieved satisfactory results on various segmentation datasets"* (Reviewer 7qT5).

---
We would like to re-emphasize the novelty and main contributions of this work:
- To the best of our knowledge, this study represents the first attempt at utilizing 2D vision foundation models for self-supervised representation learning on large-scale 3D point clouds.
- The introduce Seal framework is scalable, consistent, and generalizable. This approach is designed to capture semantic-aware spatial and temporal consistency, enabling the extraction of informative features from automotive point cloud sequences.
- Our approach demonstrates clear superiority over previous state-of-the-art methods in both linear probing and fine-tuning for downstream tasks across 11 different point cloud datasets with diverse data configurations.

---
We have revised our manuscript to include the following changes according to the reviewers’ insightful comments:
- We have supplemented more details on the definition and generation of semantic superpixel and superpoint.
- We have added some remarks on the choice and motivation behind each design.
- We conducted a study on the effect of possible misalignment between the LiDAR and camera sensors.
- We have extended the Seal framework to other 3D perception tasks.
- We have added experiments on image-LiDAR pre-training using sweeps data in nuScenes.
- We have polished the elaboration and clarified some typos and misunderstandings in the main submission.

---
Last but not least, we thank the PCs, ACs, and all the reviewers again for devoting time and effort to this review.

---

### Decision · Program_Chairs · 2023-09-21

**Decision:**

Accept (spotlight)

**Comment:**

The paper received 4 detailed reviews, and the authors did a serious job in addressing all the reviewer comments and suggestions. This includes running new experiments (e.g. performance analysis and generalization to other 3D perception tasks). Overall, the general view of the reviewers was quite positive and leaning to accept. The rebuttal addressed the majority of their concerns.